# SARS-CoV-2 is associated with changes in brain structure in UK Biobank

Gwenaëlle Douaud[1✉], Soojin Lee[1], Fidel Alfaro-Almagro[1], Christoph Arthofer[1], Chaoyue Wang[1], Paul McCarthy[1], Frederik Lange[1], Jesper L. R. Andersson[1], Ludovica Griffanti[1,2], Eugene Duff[1,3], Saad Jbabdi[1], Bernd Taschler[1], Peter Keating[4], Anderson M. Winkler[5], Rory Collins[6], Paul M. Matthews[7], Naomi Allen[6], Karla L. Miller[1], Thomas E. Nichols[8] & Stephen M. Smith[1]

There is strong evidence of brain-related abnormalities in COVID-19[1–13]. However, it remains unknown whether the impact of SARS-CoV-2 infection can be detected in milder cases, and whether this can reveal possible mechanisms contributing to brain pathology. Here we investigated brain changes in 785 participants of UK Biobank (aged 51–81 years) who were imaged twice using magnetic resonance imaging, including 401 cases who tested positive for infection with SARS-CoV-2 between their two scans—with 141 days on average separating their diagnosis and the second scan—as well as 384 controls. The availability of pre-infection imaging data reduces the likelihood of pre-existing risk factors being misinterpreted as disease effects. We identified significant longitudinal effects when comparing the two groups, including (1) a greater reduction in grey matter thickness and tissue contrast in the orbitofrontal cortex and parahippocampal gyrus; (2) greater changes in markers of tissue damage in regions that are functionally connected to the primary olfactory cortex; and (3) a greater reduction in global brain size in the SARS-CoV-2 cases. The participants who were infected with SARS-CoV-2 also showed on average a greater cognitive decline between the two time points. Importantly, these imaging and cognitive longitudinal effects were still observed after excluding the 15 patients who had been hospitalised. These mainly limbic brain imaging results may be the in vivo hallmarks of a degenerative spread of the disease through olfactory pathways, of neuroinflammatory events, or of the loss of sensory input due to anosmia. Whether this deleterious effect can be partially reversed, or whether these effects will persist in the long term, remains to be investigated with additional follow-up.

The global pandemic of SARS-CoV-2 has now claimed millions of lives across the world. There has been an increased focus by the scientific and medical community on the effects of mild-to-moderate COVID-19 in the longer term. There is strong evidence for brain-related pathologies, some of which could be a consequence of viral neurotropism[1,2,14] or virus-induced neuroinflammation[3–5,15], including the following: neurological and cognitive deficits demonstrated by patients[6,7], with an incidence of neurological symptoms in more than 80% of the severe cases[8], radiological and post mortem tissue analyses demonstrating the impact of COVID-19 on the brain[9,10], and the possible presence of the coronavirus in the central nervous system[11–13].

In particular, one consistent clinical feature, which can appear before the onset of respiratory symptoms, is the disturbance in olfaction and gustation in patients with COVID-19[16,17]. In a recent study, 100% of the patients in the subacute stage of the disease were displaying signs of gustatory impairment (hypogeusia), and 86%, signs of either hyposmia

or anosmia[18]. Such loss of sensory olfactory inputs to the brain could lead to a loss of grey matter in olfactory-related brain regions[19]. Olfactory cells—whether neuronal or supporting—concentrated in the olfactory epithelium are also particularly vulnerable to coronavirus invasion, and this seems to be also the case specifically with SARS-CoV-2[17,20–22]. Within the olfactory system, direct neuronal connections from and to the olfactory bulb encompass regions of the piriform cortex (the primary olfactory cortex), parahippocampal gyrus, entorhinal cortex and orbitofrontal areas[23,24].

Most brain imaging studies of COVID-19 to date have focussed on acute cases and radiological reports of single cases or case series based on computed tomography (CT), positron emission tomography (PET) or magnetic resonance imaging (MRI) scans, revealing a broad array of gross cerebral abnormalities, including white matter hyperintensities, hypoperfusion and signs of ischaemic events spread throughout the brain, but found more consistently in the cerebrum[9]. Of the few larger

[1]FMRIB Centre, Wellcome Centre for Integrative Neuroimaging (WIN), Nuffield Department of Clinical Neurosciences, University of Oxford, Oxford, UK. [2]OHBA, Wellcome Centre for Integrative Neuroimaging (WIN), Department of Psychiatry, University of Oxford, Oxford, UK. [3]Department of Paediatrics, University of Oxford, Oxford, UK. [4]Ear Institute, University College London, London, UK. [5]National Institute of Mental Health, National Institutes of Health, Bethesda, MD, USA. [6]Nuffield Department of Population Health, University of Oxford, Oxford, UK. [7]UK Dementia Research Institute and Department of Brain Sciences, Imperial College, London, UK. [8]Big Data Institute, University of Oxford, Oxford, UK. ✉e-mail: gwenaelle.douaud@ndcn.ox.ac.uk

studies focussing on cerebrovascular damage using CT or MRI, some have either found no clear marker of abnormalities in the majority of their patients, or importantly no spatially consistent pattern for the distribution of white matter hyperintensities or microhaemorrhages, except perhaps in the middle or posterior cerebral artery territories and the basal ganglia[9]. Imaging cohort studies of COVID-19, quantitatively comparing data across participants through automated preprocessing and co-alignment of images, are much rarer. For example, a recent PET cohort study focussing on correlates of cognitive impairment demonstrated, in 29 patients with COVID-19 at a subacute stage, the involvement of fronto-parietal areas revealed as fluorodeoxyglucose ($^{18}$F-FDG) hypometabolism[18]. Another glucose PET study has shown bilateral hypometabolism in the bilateral orbital gyrus rectus and the right medial temporal lobe[25]. One multiorgan imaging study[26] (and its brain-focussed follow-up[27]) in over 50 previously hospitalised patients with COVID-19 suggested modest abnormalities in T2* of the left and right thalami compared with matched controls. However, it remains unknown whether any of these abnormalities predates the infection by SARS-CoV-2. These effects could be associated with a pre-existing increased brain vulnerability to the deleterious effects of COVID-19 and/or a higher probability to show more pronounced symptoms, rather than being a consequence of the COVID-19 disease process.

UK Biobank offers a unique resource to elucidate these questions. With the data from this large, multimodal brain imaging study, we used for the first time a longitudinal design whereby participants had been already scanned as part of UK Biobank before being infected by SARS-CoV-2. They were then imaged again, on average 38 months later, after some had either medical and public health records of COVID-19, or had tested positive for SARS-CoV-2 twice using rapid antibody tests. Those participants were then matched with control individuals who had undergone the same longitudinal imaging protocol but had tested negative using the rapid antibody test or had no medical record of COVID-19. In total, 401 participants with SARS-CoV-2 infection with usable imaging data at both time points were included in this study, as well as 384 control individuals, matched for age, sex, ethnicity and time elapsed between the two scans. These large numbers may enable us to detect subtle, but consistent spatially distributed sites of damage associated with the infection, therefore underlining in vivo the possible spreading pathways of the effects of the disease within the brain (whether such effects relate to the invasion of the virus itself[11,14,20], inflammatory reactions[3,4,15], possible anterograde degeneration starting with the olfactory neurons in the nose, or through sensory deprivation[19,28,29]). The longitudinal aspect of the study aims to help to tease apart which of the observed effects between the first and second scans are probably related to the infection, rather than due to pre-existing risk factors between the two groups.

Our general approach in this study was therefore as follows: (1) use brain imaging data from 785 participants who visited the UK Biobank imaging centres for two scanning sessions, on average 3 years apart, with 401 of these having been infected with SARS-CoV-2 in between their two scans; (2) estimate—from each participant's multimodal brain imaging data—hundreds of distinct brain imaging-derived phenotypes (IDPs), each IDP being a measure of one aspect of brain structure or function; (3) model confounding effects, and estimate the longitudinal change in IDPs between the two scans; and (4) identify significant SARS-CoV-2 versus control group differences in these longitudinal effects, correcting for multiple comparisons across IDPs. We did this for both a focussed set of a priori-defined IDPs, testing the hypothesis that the olfactory system is particularly vulnerable in COVID-19, as well as an exploratory set of analyses considering a much larger set of IDPs covering the entire brain. In both cases, we identified significant effects associated with SARS-CoV-2 infection primarily relating to greater atrophy and increased tissue damage in cortical areas directly connected to the primary olfactory cortex, as well as to changes in global measures of brain and cerebrospinal fluid volume.

## Table 1 | Main demographics of the participants

|  | SARS-CoV-2 | Control | $P_{uncorr}$ |
|---|---|---|---|
| Number of participants | 401 | 384 | – |
| Age at scan 1 (mean±s.d. (range)) | 58.9±7.0 (46.9–80.2) | 60.2±7.4 (47.1–79.8) | 0.15 |
| Age at scan 2 (mean±s.d. (range)) | 62.1±6.7 (51.3–81.4) | 63.3±7.1 (51.3–81.3) | 0.08 |
| Sex (male/female) | 172 (42.9%)/229 (57.1%) | 164 (42.7%)/220 (57.3%) | 0.96 |
| Ethnicity (white/non-white[a]) | 388 (96.8%)/13 (3.2%) | 373 (97.1%)/11 (2.9%) | 0.76 |
| Years between scans 1 and 2 (mean±s.d. (range)) | 3.2±1.6 (1.0–7.0) | 3.2±1.6 (1.0–6.9) | 0.98 |
| Systolic blood pressure (mmHg) | 130.3±17.3 | 132.1±17.6 | 0.16 |
| Diastolic blood pressure (mmHg) | 78.7±10.6 | 79.0±10.2 | 0.63 |
| Diagnosed diabetes | 18 (4.5%) | 16 (4.2%) | 0.82 |
| Weight (kg) | 76.4±15.8 | 75.2±14.4 | 0.65 |
| Waist/hip ratio | 0.87±0.09 | 0.86±0.09 | 0.37 |
| BMI (kg m$^{-2}$) | 26.7±4.4 | 26.6±4.3 | 0.61 |
| Alcohol-intake frequency (a.u.) | 3.1±1.3 | 3.0±1.4 | 1.00 |
| Tobacco smoking | 0.61 ± 0.92 | 0.65 ± 0.89 | 0.87 |
| Townsend deprivation index | −1.5±2.9 | −1.6±2.9 | 0.65 |

We used the 'last observation carried forward' imputation method (see the 'Baseline group comparisons' section in the Methods). Nonparametric tests were used whenever a variable for each group was not normally distributed (Lilliefors $P < 0.05$). Two-sample Kolmogorov–Smirnov tests were used for age at scan 1 or scan 2, years between scan 1 and scan 2, alcohol-intake frequency and tobacco smoking; $\chi^2$ tests were used for sex, ethnicity and diagnosed diabetes; and Mann–Whitney $U$-tests were used for the systolic and diastolic blood pressures, weight, waist/hip ratio, BMI and Townsend deprivation index. a.u., arbitrary units.
[a]The white/non-white distinction was made as numbers were too low to allow for a finer distinction.

## Participants

UK Biobank has been releasing data from the COVID-19 re-imaging study on a rolling basis. As of 31 May 2021, 449 adult participants met the re-imaging study inclusion criteria (see the 'Study design' section of the Methods) and were identified as having been infected with SARS-CoV-2 based on either their primary care (GP) data, hospital records, results of their diagnostic antigen tests identified through record linkage to the Public Health datasets in England, Wales and Scotland, or two concordant antibody-based home lateral flow kit positive results. Of these 449 adult participants who had tested positive for SARS-CoV-2, a total of 401 had usable brain scans at both time points (Tables 1 and 2). For the 351 individuals for whom we had a diagnosis date based on their medical records or antigen tests, the time between diagnosis (a proxy for infection) and their second imaging scan was on average 141 days (Table 2 and Supplementary Fig. 1).

In total, 384 adult control participants met the inclusion criteria (see the 'Study design' section of the Methods) and had usable brain scans at both time points (Table 1). SARS-CoV-2 positive or negative status was identified using UK Biobank Showcase variable 41000.

Despite the original matched pairing of the SARS-CoV-2 cases and controls, their age distributions were slightly—albeit not statistically significantly—different, due to different patterns of missing/usable data (Extended Data Fig. 1). Note that the control group is on average slightly (albeit not significantly) older than the SARS-CoV-2-positive group, which would be expected to make any change between the two time points more difficult to detect in the group comparisons, rather than easier. Histograms of interval of time between the two scans in the two groups are shown in Extended Data Fig. 2.

## Table 2 | Main clinical information for SARS-CoV-2 cases

| | n or mean±s.d. (range) |
|---|---|
| **Total number of positive cases** | 401 |
| **Origin of diagnosis** | |
| GP | 11 |
| Hospital | 2 |
| Diagnostic antigen test from Public Health records | 338 |
| Antibody home-based lateral flow kits | 50 |
| **Number of infected participants with available information on date of diagnoses** | 351 |
| **Days of SARS-CoV-2 infection before scan 2** | 141±79 (35–407) |
| **Total number of hospitalised patients** | 15 |
| COVID-19 as primary cause | 11 |
| COVID-19 as secondary cause | 4 |
| Days of hospitalisation | 11.1±11.0 (1–40) |
| Critical care unit | 2 |
| Invasive ventilation | 1 |
| Continuous positive airway pressure | 1 |
| Non-invasive ventilation | 1 |
| Unspecified oxygen therapy | 1 |

Of the 401 participants in our SARS-CoV-2-positive group in our main analyses, 50 were identified as cases on the basis of two different antibody home-based lateral flow kits and do not have a date of diagnosis in their primary care or hospital records.

## Table 3 | Comparison between hospitalised versus non-hospitalised SARS-CoV-2 cases

| | Hospitalised | Non-hospitalised | $P_{uncorr}$ |
|---|---|---|---|
| Number of participants | 15 | 386 | – |
| Age at scan 1 (mean±s.d. (range)) | 65.4±8.9 (51.6–80.2) | 58.7±6.8 (46.9–77.0) | 0.0028 |
| Age at scan 2 (mean±s.d. (range)) | 68.1±8.4 (54.9–81.4) | 61.9±6.5 (51.3–80.0) | 0.0058 |
| Sex (male/female) | 10 (66.7%)/5 (33.3%) | 162 (42.0%)/224 (58.0%) | 0.058 |
| Ethnicity (white/non-white[a]) | 15 (100%)/0 (0%) | 373 (96.6%)/13 (3.4%) | 0.47 |
| Years between scan 1 and 2 (mean±s.d. (range)) | 2.7±1.4 (1.0–5.8) | 3.2±1.6 (1.1–7.0) | 0.50 |
| Systolic blood pressure (mmHg) | 140.6±16.6 | 129.9±17.2 | 0.022 |
| Diastolic blood pressure (mmHg) | 85.0±10.5 | 78.4±10.5 | 0.028 |
| Diagnosed diabetes | 4 (26.7%) | 14 (3.6%) | <0.001 |
| Weight (kg) | 85.9±12.0 | 76.0±15.8 | 0.0072 |
| Waist/hip ratio | 0.94±0.07 | 0.87±0.09 | 0.0015 |
| BMI (kg m$^{-2}$) | 29.3±3.7 | 26.6±4.4 | 0.0076 |
| Alcohol-intake frequency | 3.1±1.7 | 3.1±1.3 | 1.00 |
| Tobacco smoking | 0.80±1.0 | 0.60±0.91 | 0.75 |
| Townsend deprivation index | −2.1±2.6 | −1.5±2.9 | 0.42 |

Details of the statistical procedures are provided in Table 1.

[a]The white/non-white distinction was made as numbers were too low to allow for a finer distinction.

The two groups showed no statistical differences across all 6,301 non-imaging phenotypes after false-discovery rate (FDR) or family-wise error (FWE) correction for multiple comparisons (lowest $P_{FWE}$ = 0.12, and no uncorrected P values survived FDR correction). However, owing to the stringent correction for multiple comparisons that this analysis imposes, we investigated further whether subtle patterns of baseline differences could be observed using dimension reduction with principal component analysis on all 6,301 variables, and using a separate principal component analysis focussed on baseline cognition (Supplementary Analysis 1). We found no principal components that differed significantly between the two groups when examining all of the non-imaging variables. With respect to cognitive tests, although no single cognitive score was significantly different at baseline between control individuals and participants who were later infected (future cases), we identified two cognitive principal components that were different (Supplementary Analysis 1). These subtle baseline cognitive differences suggest that the future cases had slightly lower cognitive abilities compared with the control individuals. Importantly, none of these principal components—cognitive or otherwise—could statistically account for the longitudinal imaging results (see the 'Additional baseline investigations' section below).

Through hospital records, we identified 15 participants in the SARS-CoV-2-positive group who had been hospitalised with COVID-19, including 2 who received critical care (Tables 2 and 3). These hospitalised patients were on average older, had higher blood pressure and weight, and were more likely to have diabetes and be men, compared with non-hospitalised cases (Table 3).

## Hypothesis-driven results

The main case-versus-control analysis between the 401 SARS-CoV-2 cases and 384 controls (Model 1) on 297 olfactory-related cerebral IDPs yielded 68 significant results after FDR correction for multiple comparisons, including 6 that survived FWE correction (Table 4 and Fig. 1; a full list of the results is provided in Supplementary Table 1). Focussing on the top 10 most significant associations, 8 of these IDPs

covered similar brain regions that are functionally connected to the primary olfactory cortex (see the 'Hypothesis-driven approach' section in the Methods), showing overlap especially in the anterior cingulate cortex, orbitofrontal cortex and insula, as well as in the ventral striatum, amygdala, hippocampus and parahippocampal gyrus[30]. We found a greater longitudinal increase in diffusion indices for the SARS-CoV-2 group in these tailored IDPs defining the functional connections with the frontal and temporal piriform cortex, as well as the olfactory tubercle and anterior olfactory nucleus (Table 4, Fig. 1 and Supplementary Table 1). The other two of the top 10 IDPs encompassed the left lateral orbitofrontal cortex and parahippocampal gyrus, both showing a greater reduction in grey matter thickness or intensity contrast over time for cases compared with controls (Table 4, Fig. 1 and Supplementary Table 1). For those significant IDPs, the average differences in percentage change between the two groups were moderate, ranging from around 0.2% to about 2%, with the largest differences observed in the volume of the parahippocampal gyrus and entorhinal cortex (Supplementary Table 1). Scatter and box plots, as well as plots showing the percentage changes with age are available for the top 10 longitudinal IDPs (Supplementary Information, Longitudinal Plots).

As secondary analyses, we found that significant longitudinal differences remained in the same set of significant brain regions that survived FDR or FWE correction when removing from the SARS-CoV-2 group those patients who had been hospitalised with COVID-19 (Model 2, 47 IDPs significant after FDR correction, 3 of which were also significant after FWE correction; Supplementary Table 1). Although fewer results were significant for the comparison between the 15 hospitalised patients and 384 control individuals (Model 3, 4 results were significant after FDR correction; Supplementary Table 1), probably due to the large reduction in sample size for this model, this additional group comparison showed effects in the same regions of the parahippocampal gyrus, orbital cortex and superior insula. Finally, we found no significant differences between the 15 hospitalised patients and 386 non-hospitalised

**Table 4 | Top 10 of the 68 significant results for the hypothesis-driven olfactory approach**

| IDP | Main analysis (Model 1): all SARS-CoV-2 cases (*n*=401) versus control participants (*n*=384) | | | | | Model 2: non-hospitalised cases (*n*=386) versus control participants (*n*=384) | | | Model 3: hospitalised cases (*n*=15) versus control participants (*n*=384) | | | Model 4: hospitalised (*n*=15) versus non-hospitalised cases (*n*=386) | | |
|---|---|---|---|---|---|---|---|---|---|---|---|---|---|---|
| | % | s.e. | *Z* | $P_{uncorr}$ | $P_{FWE}$ | *Z* | $P_{uncorr}$ | $P_{FWE}$ | *Z* | $P_{uncorr}$ | $P_{FWE}$ | *Z* | $P_{uncorr}$ | $P_{FWE}$ |
| Temporal piriform cortex functional network: OD | **0.34** | **0.08** | **4.2** | **0.000023** | **0.0068** | *3.9* | *0.000081* | *0.0217* | 2.5 | 0.013985 | 0.9176 | 0.5 | 0.627678 | 1 |
| Olfactory tubercle functional network: ISOVF | **1.22** | **0.31** | **3.9** | **0.000102** | **0.028** | 3.4 | 0.000623 | 0.1319 | 2.8 | 0.004439 | 0.6576 | 1.4 | 0.155595 | 1 |
| Frontal piriform cortex functional network: MD | **0.39** | **0.1** | **3.8** | **0.000146** | **0.0386** | 3.4 | 0.000671 | 0.1417 | 2.4 | 0.017728 | 0.9532 | 1.3 | 0.202213 | 1 |
| Temporal piriform cortex functional network: MD | **0.38** | **0.1** | **3.8** | **0.00015** | **0.0396** | 3.3 | 0.000849 | 0.1717 | 2.5 | 0.011746 | 0.8913 | 1.4 | 0.172706 | 1 |
| Olfactory tubercle functional network: MD | **0.39** | **0.1** | **3.8** | **0.000171** | **0.0446** | 3.3 | 0.000946 | 0.188 | 2.5 | 0.011856 | 0.8931 | 1.4 | 0.173673 | 1 |
| Lateral orbitofrontal cortex L: thickness (DKT atlas) | **−0.76** | **0.2** | **−3.8** | **0.000172** | **0.0449** | *−2.9* | *0.003178* | *0.48* | −3.1 | 0.001957 | 0.4541 | −1.9 | 0.061739 | 0.9999 |
| Temporal piriform cortex functional network: ISOVF | 1.12 | 0.3 | 3.7 | 0.000222 | 0.0564 | 3.3 | 0.001068 | 0.2088 | 2.6 | 0.009838 | 0.8562 | 1.3 | 0.18374 | 1 |
| Anterior olfactory nucleus functional network: MD | 0.42 | 0.11 | 3.7 | 0.000242 | 0.0621 | 3.5 | 0.000559 | 0.1213 | 1.9 | 0.060313 | 0.9993 | 0.8 | 0.432506 | 1 |
| Parahippocampal gyrus L: intensity contrast (Desikan) | −0.92 | 0.25 | −3.7 | 0.000276 | 0.0685 | −2.7 | 0.006201 | 0.704 | −3.5 | *0.000507* | *0.212* | −2.0 | 0.042678 | 0.998 |
| Anterior olfactory nucleus functional network: ISOVF | 1.27 | 0.35 | 3.6 | 0.000286 | 0.0701 | 3.3 | 0.000985 | 0.1957 | 2.4 | 0.016069 | 0.9399 | 1.1 | 0.268503 | 1 |

The top 10 significant results, all surviving FDR correction, based on 297 IDPs, ranked on the basis of their uncorrected *P* values for our main analysis (Model 1), showing where the 401 SARS-CoV-2 infected participants and 384 controls differed over time. Associations with a total of 68 IDPs in total survived FDR correction for multiple comparisons for Model 1 (a full list of results is provided in Supplementary Table 1). We report differences in longitudinal change (as a percentage of the mean baseline value) between the two groups, s.e. of these percentage changes for Model 1, as well as uncorrected and FWE-corrected *P* values. The results in italics are significant after FDR correction for multiple comparisons for each corresponding model. The results in bold are significant after FWE correction for multiple comparisons for each corresponding model. Note: the *Z*-statistics reflect the statistical strength of the longitudinal group-difference modelling, and are not raw data effect sizes. All significant results involved either grey matter thickness, grey-white intensity contrast or proxy measures of tissue damage. ISOVF, isotropic volume fraction; MD, mean diffusivity; OD, orientation dispersion OD; L, left.

SARS-COV-2 cases, probably due to the large reduction in sample size, but effect sizes and direction of these effects suggested stronger detrimental effects for the hospitalised cases in the orbitofrontal, insula, parahippocampal and frontal piriform cortex functionally connected brain regions (all |*Z*| ≥ 3, Model 4; Supplementary Table 1).

Across the three models comparing SARS-CoV-2 cases with controls (Models 1–3), the top 4 longitudinal differences were found in the functionally connected regions of the temporal piriform cortex (diffusion index: orientation dispersion) and of the olfactory tubercle (diffusion index: isotropic volume fraction), as well as in the parahippocampal gyrus (intensity contrast) and lateral orbitofrontal cortex (thickness) (largest combined |*Z*| across Models 1–3; Fig. 1). For these results across Models 1–3, the percentage of participants infected with SARS-CoV-2 who showed a greater longitudinal change than the median value in the control individuals was 56% for the regions connected to the temporal piriform cortex, 62% for the regions connected to the olfactory tubercle, 57% for the left parahippocampal gyrus and 60% for the left orbitofrontal cortex.

Although significant IDPs related to grey matter thickness were found, using our main case-versus-control analysis (Model 1), to be bilateral for both the anterior parahippocampal gyrus (perirhinal cortex) and entorhinal cortex, 10 out of the 11 remaining significant IDP were left-lateralised (Supplementary Table 1). Thus, we directly investigated left–right differences in the group with SARS-CoV-2 only for those significant IDPs, and found that the participants with an infection did not have a significantly greater reduction in grey matter thickness in the left hemisphere compared with in the right hemisphere (lowest $P_{uncorr}$ = 0.30).

Of the top 10 IDPs showing a longitudinal effect between first and second scans, none correlated significantly with the time interval between their infection and their second scan, in the participants who tested positive for SARS-CoV-2 for whom we had a date of diagnosis (*n* = 351; lowest $P_{uncorr}$ = 0.08).

## Exploratory results

In total, 2,047 IDPs passed the initial tests of reproducibility (Extended Data Fig. 3) and data completeness. The main analysis (Model 1) revealed 65 significant longitudinal differences between the cases and controls that passed FDR correction, including 5 that were significant after FWE correction (Table 5; a complete list of reproducible IDPs and results is provided in Supplementary Table 1). Extended Data Figs. 4 and 5 show the *QQ* plot relating to the FDR thresholding, and a summary figure of *Z*-statistics results for all 2,047 IDPs grouped into different IDP classes.

In particular, in this exploratory analysis covering the entire brain, 33 out of the 65 significant IDPs overlapped with the IDPs selected a priori

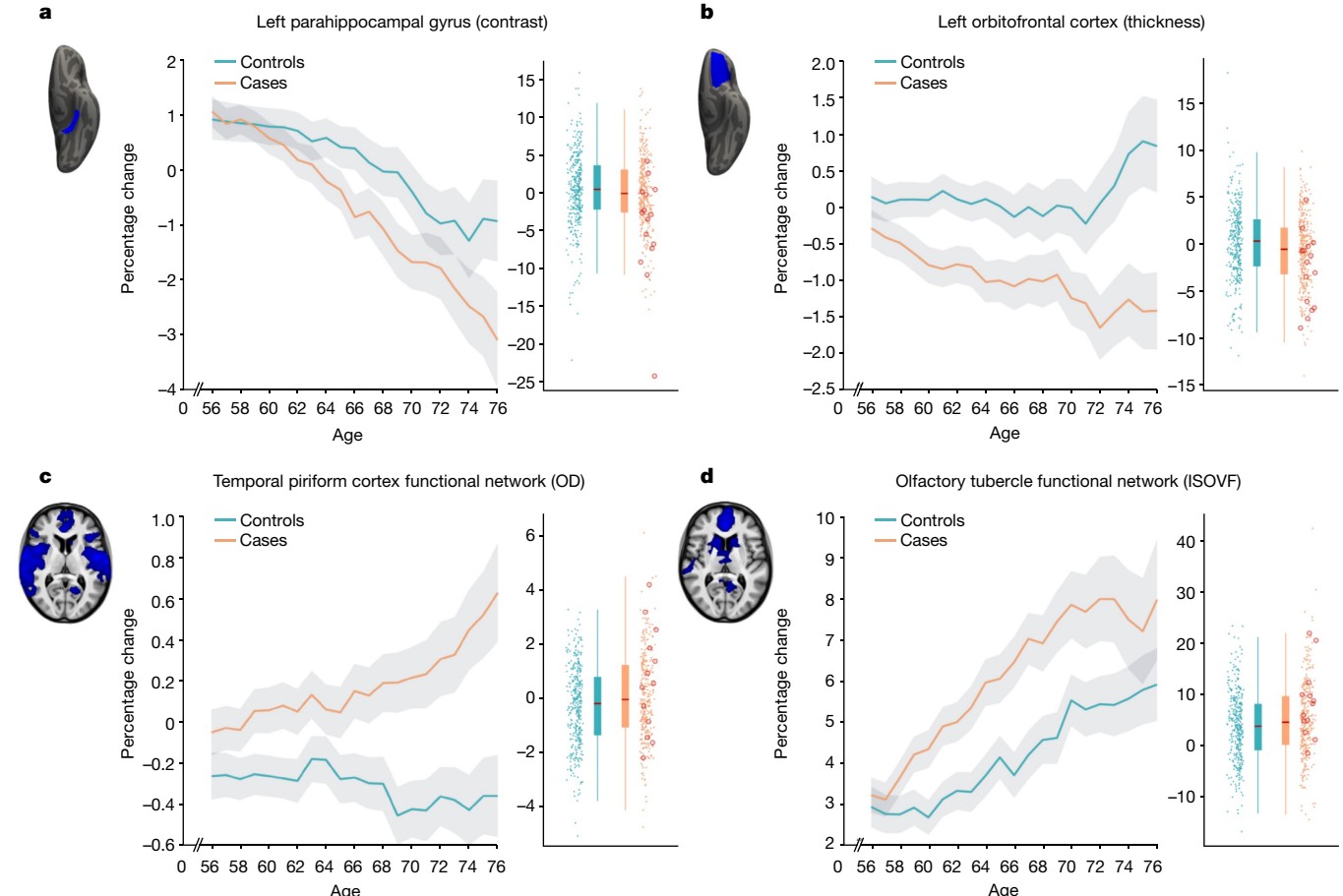

**Fig. 1 | The most significant longitudinal group comparison results from the hypothesis-driven approach. a–d**, The top four regions consistently showing longitudinal differences across the three models comparing SARS-CoV-2 cases and controls demonstrated either a significantly greater reduction in grey matter thickness and intensity contrast, or an increase in tissue damage (largest combined $|Z|$ across Models 1–3). All three models pointed to the involvement of the parahippocampal gyrus (**a**), whereas Models 1 and 2 also showed the significant involvement of the left orbitofrontal cortex (**b**) and of the functional connections of the primary olfactory cortex (**c**, **d**). For each region, the IDP's spatial region of interest is shown at the top left in blue, overlaid either on the FreeSurfer average inflated cortical surface, or the T1 template (left is shown on the right). For each IDP, the longitudinal percentage changes are shown with age for the two groups (control participants in blue, participants with infection in orange), obtained by normalising ΔIDP using the values for the corresponding IDPs across the 785 participants' scans as the baseline. These are created using a 10-year sliding window average, with s.e.m. values shown in grey. The counter-intuitive increase in thickness in the orbitofrontal cortex in older controls has been previously consistently reported in studies of ageing[57,58]. The difference in cortical thickness, intensity contrast or diffusion indices between the two time points is shown for the 384 controls (blue) and 401 infected participants (orange), enabling a visual comparison between the two groups in a binary manner (therefore underestimating the effects estimated when modulating with age; see the 'Main longitudinal model, deconfounding' section in the Methods). The 15 hospitalised patients are indicated (red circles). ISOVF, isotropic volume fraction; OD, orientation dispersion. All *y* axes represent the percentage change.

for our hypothesis-driven approach of the involvement of the olfactory system. Moreover, we found significant longitudinal effects in global measures of volume, such as the cerebrospinal fluid (CSF) volume normalised for head size and the ratio of the volume of the segmented brain to the estimated total intracranial volume generated by FreeSurfer, as well as in the volume of the left crus II of the cerebellum, the thickness of the left rostral anterior cingulate cortex and diffusion index in the superior fronto-occipital fasciculus (Table 5 and Supplementary Table 1; examples are provided in Extended Data Fig. 6). For those significant IDPs, average differences in the percentage change between the two groups were moderate, ranging from around 0.2% to about 2% (except for two diffusion measures in the fimbria at >6%, due to the very small size of these regions of interest), with the largest differences observed in the volume of the parahippocampal gyrus and caudal anterior cingulate cortex (Supplementary Table 1). Scatter and box plots, as well as plots showing the percentage longitudinal changes with age are available for the top 10 longitudinal IDPs (Supplementary Information, Longitudinal Plots).

For the secondary analyses, when comparing the non-hospitalised cases with the controls (Model 2), the same general pattern emerged, albeit with a reduced number of significant results: one olfactory-related region, the functionally connected areas to the temporal piriform cortex, showed a significant longitudinal difference between the two groups in a diffusion index, as well as one global volume measure (CSF normalised), and a diffusion index in the superior fronto-occipital fasciculus (Model 2; 4 after FDR correction, 1 after FWE correction; Supplementary Table 1). Despite the considerably limited degrees of freedom in Models 3 and 4, many results remained significant after correction for multiple comparison, particularly for IDPs of cortical thickness, with an emphasis on the anterior cingulate cortex for Model 3 (66 results after FDR correction, 3 results after FWE correction), and a wide distribution across prefrontal, parietal and temporal lobes for Model 4 (29 FDR-corrected results; Fig. 2).

As many of the top exploratory and hypothesis-driven results included IDPs of cortical thickness and of mean diffusivity, we further conducted an exploratory visualisation of the vertex-wise thickness, and voxel-wise

**Table 5 | Top 10 of 65 significant results for the exploratory approach**

| IDP | Main analysis (Model 1): all SARS-CoV-2 cases (*n*=401) versus control participants (*n*=384) | | | | | Model 2: non-hospitalised cases (*n*=386) versus control participants (*n*=384) | | | Model 3: hospitalised cases (*n*=15) versus control participants (*n*=384) | | | Model 4: hospitalised (*n*=15) versus non-hospitalised (*n*=386) cases | | |
|---|---|---|---|---|---|---|---|---|---|---|---|---|---|---|
| | % | s.e. | Z | $P_{uncorr}$ | $P_{FWE}$ | Z | $P_{uncorr}$ | $P_{FWE}$ | Z | $P_{uncorr}$ | $P_{FWE}$ | Z | $P_{uncorr}$ | $P_{FWE}$ |
| Ratio brain volume/estimated total intracranial volume | **−0.29** | **0.06** | **−4.6** | **0.000004** | **0.0083** | −3.2 | 0.001175 | 0.7836 | −4.5 | 0.000006 | 0.0708 | −3.4 | 0.000787 | 0.8043 |
| Normalised CSF: volume | **1.52** | **0.35** | **4.3** | **0.000016** | **0.0277** | 4.1 | 0.000047 | 0.0791 | 1.8 | 0.068896 | 1 | 0.5 | 0.620269 | 1 |
| Lateral ventricle R: volume | **1.7** | **0.4** | **4.3** | **0.000019** | **0.0329** | 3.7 | 0.000239 | 0.3009 | 2.7 | 0.006833 | 0.998 | 1.2 | 0.218988 | 1 |
| Temporal piriform cortex functional network: OD | **0.34** | **0.08** | **4.2** | **0.000023** | **0.0405** | 3.9 | 0.000081 | 0.1293 | 2.5 | 0.013985 | 0.9999 | 0.5 | 0.627678 | 1 |
| Superior fronto-occipital fasciculus: ICVF | **−0.79** | **0.19** | **−4.2** | **0.000025** | **0.0431** | **−4.3** | **0.000017** | **0.0297** | −1.1 | 0.278361 | 1 | 0.5 | 0.584195 | 1 |
| Brain volume without ventricles: surface model estimate | −0.3 | 0.07 | −4.1 | 0.000043 | 0.0685 | −2.9 | 0.003266 | 0.9776 | −4.0 | 0.00007 | 0.302 | −2.7 | 0.007248 | 0.9994 |
| Rostral anterior cingulate cortex L: thickness (Desikan) | −1.2 | 0.29 | −4.1 | 0.000043 | 0.069 | −2.9 | 0.003812 | 0.9877 | −4.3 | 0.00002 | 0.1483 | −2.3 | 0.021995 | 1 |
| Brain volume without ventricles | −0.3 | 0.07 | −4.1 | 0.000045 | 0.0712 | −2.9 | 0.003412 | 0.9813 | −4.0 | 0.000073 | 0.3086 | −2.7 | 0.007081 | 0.9992 |
| Supratentorial volume without ventricles | −0.32 | 0.08 | −4.0 | 0.000057 | 0.0901 | −2.9 | 0.003349 | 0.9799 | −3.8 | 0.000167 | 0.4577 | −2.5 | 0.012125 | 1 |
| Cerebellum crus II: volume | −0.78 | 0.19 | −4.0 | 0.000064 | 0.1 | −3.1 | 0.001986 | 0.9139 | −3.3 | 0.000932 | 0.8377 | −2.1 | 0.037117 | 1 |

The top 10 significant results show where the 401 infected participants and 384 control participants differed over time, ranked on the basis of their uncorrected *P* values for Model 1. Associations with a total of 65 IDPs remained significant after FDR correction for multiple comparisons for Model 1 (a full list of results is provided in Supplementary Table 1). The findings that remained significant after FDR correction for multiple comparisons for each model are shown in italics; those that remained significant after FWE correction are shown in bold. Note: the *Z*-statistics reflect the statistical strength of the longitudinal group-difference modelling, and are not raw data effect sizes. In addition to global measures relating to loss of brain volume (such as an increase of CSF volume), most of the top exploratory localised results implicate the primary connections of the olfactory system, as well as the rostral anterior cingulate cortex and crus II of the cerebellum, both also olfactory-related regions. ICVF, intracellular volume fraction; L, left; R, right.

mean diffusivity longitudinal differences between the cases and controls over the entire cortical surface and brain volume, respectively (Fig. 2). Grey matter thickness showed bilateral longitudinal differences in the parahippocampal gyrus, anterior cingulate cortex and temporal pole, as well as in the left orbitofrontal cortex, insula and supramarginal gyrus.

When visually comparing hospitalised and non-hospitalised cases, these longitudinal differences showed a similar pattern, especially in the parahippocampal gyrus, orbitofrontal and anterior cingulate cortex, but also markedly extending, particularly in the left hemisphere, to many fronto-parietal and temporal regions. Mean diffusivity differences in longitudinal effects between cases and controls were seen mainly in the orbitofrontal cortex, anterior cingulate cortex, as well as in the left insula and amygdala.

Although the results seen in IDPs of grey matter thickness seemed to indicate that the left hemisphere is more strongly associated with SARS-CoV-2 infection, direct (left–right) comparisons of all lateralised IDPs of thickness across the entire cortex showed no overall statistical difference between the two groups (lowest $P_{FWE}$ = 0.43, with no significant results after FDR correction).

## Cognitive results

Using the main model used to compare longitudinal imaging effects between SARS-CoV-2-positive participants and controls (Model 1), we explored differences between the two groups in ten scores from six cognitive tasks. These ten scores were selected using a data-driven approach based on out-of-sample participants who are the most likely to show cognitive impairment (Supplementary Analysis 2). After FDR correction, we found a significantly greater increase in the time taken to complete trails A (numeric) and B (alphanumeric) of the Trail Making Test in the group with SARS-CoV-2 infection (trail A: 7.8%, $P_{uncorr}$ = 0.0002, $P_{FWE}$ = 0.005; trail B: 12.2%, $P_{uncorr}$ = 0.00007, $P_{FWE}$ = 0.002; Fig. 3). These findings remained significant when excluding the 15 hospitalised cases (Model 2: trail A: 6.5%, $P_{uncorr}$ = 0.002, $P_{FWE}$ = 0.03; trail B: 12.5%, $P_{uncorr}$ = 0.00009, $P_{FWE}$ = 0.002).

In the SARS-CoV-2 group only, post hoc associations between the most significant cognitive score showing longitudinal effect using Model 1 (duration to complete trail B, as reported above) and the top 10 results from each of the hypothesis-driven and exploratory approaches revealed a significant longitudinal association with the volume of the mainly cognitive lobule crus II of the cerebellum (*r* = −0.19, $P_{FWE}$ = 0.020).

## Additional baseline investigations

When looking at binary baseline differences between controls and individuals who were later infected, none of the IDPs with significant longitudinal effects for either the hypothesis-driven or the exploratory approach demonstrated significant differences at baseline between the two groups

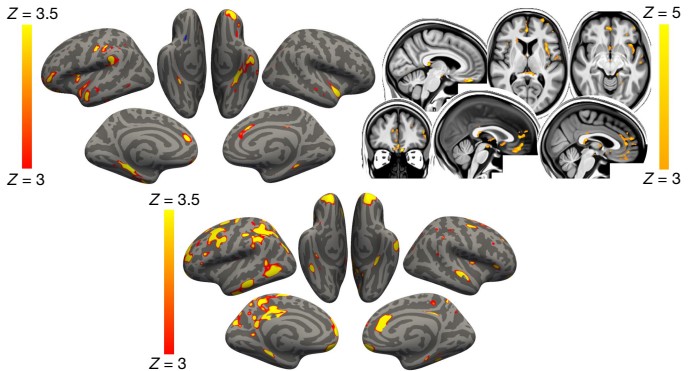

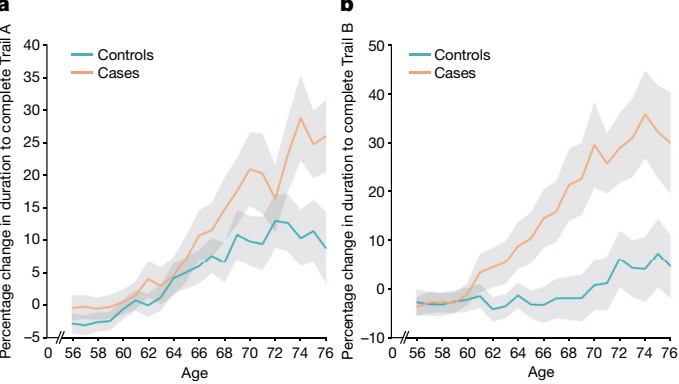

**Fig. 2 | Vertex-wise and voxel-wise longitudinal group differences in grey matter thickness and mean diffusivity changes.** Top, the main analysis (Model 1): the thresholded map ($|Z| > 3$) shows that the strongest, localised reductions in grey matter thickness in the 401 infected participants compared with the 384 controls are bilaterally in the parahippocampal gyrus, anterior cingulate cortex and temporal pole, as well as in the left orbitofrontal cortex, insula and supramarginal gyrus. Similarly, the strongest longitudinal differences in mean diffusivity ($|Z| > 3$, left is shown on the right) could be seen in the orbitofrontal cortex and anterior cingulate cortex, as well as in the left insula and amygdala (top). Bottom, secondary analysis (Model 4): the thresholded cortical thickness map ($|Z| > 3$) demonstrated longitudinal differences between the 15 hospitalised and 386 non-hospitalised SARS-CoV-2-positive cases in the orbitofrontal frontal cortex and parahippocampal gyrus bilaterally, right anterior cingulate cortex, as well as marked widespread differences in fronto-parietal and temporal areas, especially in the left hemisphere. We show the voxel-wise or vertex-wise longitudinal effects for illustrative purposes, avoiding any thresholding based on significance (as this would be statistically circular), similar to our previous analyses[59].

**Fig. 3 | Significant longitudinal differences in cognition. a, b,** The percentage longitudinal change for SARS-CoV-2-positive cases and controls in the duration to complete trails A (**a**) and B (**b**) of the UK Biobank Trail Making Test. The absolute baseline (used to convert longitudinal change into percentage change) was estimated across the 785 participants. These curves were created using a ten-year sliding window across cases and controls (s.e. values are shown in grey).

(lowest $P_{FWE} = 0.59$, none were significant after FDR correction; Supplementary Table 2). When applying age modulation in the two-group modelling of IDPs at baseline, a few of the IDPs demonstrated significant differences between the control and future SARS-CoV-2 groups, mainly for diffusion indices in the olfactory functional networks, as well as in the subcortical grey matter. As some IDPs cover spatially extended regions of the brain, we visually examined whether these baseline differences had any spatial overlap with our longitudinal results, but found none (Supplementary Fig. 2). The full list of binary and age-modulated results from group comparisons between the two groups at baseline is available in Supplementary Table 2 (and separately, at the second time point, in Supplementary Table 3). We also provide the scatter plots and box plots, as well as the percentage changes with age at baseline for the top 10 significant longitudinal IDPs from the hypothesis-driven and exploratory approaches (Supplementary Information, Baseline Plots).

Furthermore, none of the 10 preselected cognitive variables showed a significant difference at baseline between the group with SARS-CoV-2 and the control group (minimum $P_{uncorr} = 0.08$). With age modulation, only one cognitive score—time to complete pairs matching round—showed a trend difference at baseline ($P_{uncorr} < 0.05$, $P_{FWE} = 0.29$, not significant after FDR correction). This is a different cognitive score from the one showing longitudinal cognitive effects between the two groups—the UK Biobank Trail Making Test.

We also repeated the main analysis modelling for those top 10 IDPs that were found to show longitudinal differences between the group with SARS-CoV-2 and the control group, across both hypothesis-driven and exploratory approaches. For each of the 6,301 non-imaging variables available (see the 'Additional analyses—baseline group comparisons' section in the Methods), we included that variable as an additional confounder in the longitudinal analyses. On the basis of the regression $Z$-statistic values, the strength of the original associations was not reduced by more than 25% for any of the non-imaging variables.

We further carried out the same analyses, but using dimension reduction (principal component analysis) applied to these 6,301 non-imaging phenotypes ($d = 1$ to $d = 700$), and also focussing only on cognition, with 540 cognitive variables ($d = 10$). We found no substantial reduction in our longitudinal results with any of these principal components. In particular, for cognition in which two components were significantly different at baseline (PC1 and PC4; Supplementary Analysis 1), the strongest reduction in $Z$ was found for crus II of the cerebellum when adding PC1 to the model, with a decrease in $Z$ of only 5.7% (from $Z = 4$ to $Z = 3.77$), whereas the $Z$ values associated with all of the other IDPs were reduced by less than 5%. Adding PC4 to our main model reduced $Z$ by 0.4% at most.

## Longitudinal effects of pneumonia and influenza

To investigate whether pneumonia might have had an impact on our longitudinal findings, we assessed the age-modulated effects associated with pneumonia in an out-of-sample UK Biobank cohort that had been scanned twice. We identified 11 participants who contracted pneumonia not related to COVID-19 between the two scans, matched these to 261 controls and applied our main analysis (Model 1) to these two groups. This longitudinal investigation showed some significant group differences in IDPs, but with no overlap with those IDPs that we found for SARS-CoV-2 (all in the white matter; Supplementary Analysis 3). Overall, the correlation between the (unthresholded) $Z$-statistics of all of the IDPs from pneumonia and SARS-CoV-2 longitudinal group comparisons was very low ($r = 0.057$).

The sample size of cases who contracted influenza between the two scans in the out-of-sample UK Biobank cohort was unfortunately much smaller ($n = 5$, including $n = 3$ hospitalised cases), likely due to the low probability of influenza being recorded by a medical professional (GP or hospital). Nevertheless, for completeness, we also assessed longitudinally these two very small groups, compared with 127 matched controls. No result was significant for the 5 influenza cases, although a few IDPs showed significant longitudinal age-modulated effects, with just one IDP in the brainstem common to the SARS-CoV-2 findings (Supplementary Analysis 4). Correlation of $Z$-statistics between influenza and SARS-CoV-2 longitudinal group comparisons was again low ($r = 0.077$).

## Discussion

This is to our knowledge the first longitudinal imaging study of SARS-CoV-2 in which the participants were initially scanned before any

of them had been infected. Our longitudinal analyses revealed a significant, deleterious impact associated with SARS-CoV-2. This effect could be seen mainly in the limbic and olfactory cortical system, for example, with a change in diffusion measures—proxies for tissue damage—in regions that are functionally connected to the piriform cortex, olfactory tubercle and anterior olfactory nucleus, as well as a more pronounced reduction of grey matter thickness and contrast in the participants infected with SARS-CoV-2 in the left parahippocampal gyrus and lateral orbitofrontal cortex. Although the greater atrophy for the participants who tested positive for SARS-CoV-2 was localised to a few, mainly limbic, regions, the increase in CSF volume and decrease in whole-brain volume suggests an additional diffuse loss of grey matter superimposed onto the more regional effects observed in the olfactory-related areas. Note that these structural and microstructural longitudinal significant differences are modest in size—the strongest differences in changes observed between the SARS-CoV-2-positive and control groups, corresponding to around 2% of the mean baseline IDP value (Supplementary Table 1). This additional loss in the infected participants of 0.7% on average across the olfactory-related brain regions—and specifically ranging from 1.3% to 1.8% for the FreeSurfer volume of the parahippocampal/perirhinal and entorhinal cortex—can be helpfully compared with, for example, the longitudinal loss per year of around 0.2% (in middle age) to 0.3% (in older age) in hippocampal volume in community-dwelling individuals[31]. Our statistics also represent an average effect; not every infected participant will display longitudinal brain abnormalities. Comparing the few patients ($n = 15$) who had been hospitalised with COVID-19 against non-hospitalised cases showed a more widespread pattern of a greater reduction in grey matter thickness in the fronto-parietal and temporal regions (Fig. 2). Finally, significantly greater cognitive decline, which persisted even after excluding the hospitalised patients, was observed in the SARS-CoV-2-positive group between the two time points, and this decline was associated with greater atrophy of crus II—a cognitive lobule of the cerebellum.

Much has been made of the benefit of using a longitudinal design to estimate, for example, trajectories of brain ageing and cognitive decline[32,33]. The longitudinal nature of the UK Biobank COVID-19 re-imaging study, with the baseline scan acquired before infection by SARS-CoV-2 and the second scan after infection, reveals differences over time above and beyond any potential baseline differences, thereby helping to disentangle the direct or indirect contribution of the pathogenic process from pre-existing differences in the brain, or risk factors, of future patients with COVID-19. An illustrative example of the benefit of a longitudinal design is that, if looking solely at cross-sectional group comparisons at the second time point after infection (that is, the analysis that would, by necessity, be carried out in post hoc studies), the strongest effect is observed in the volume of the thalamus. However, this effect disappears when taking into account the baseline scans, as the thalamus of the participants who were later infected appears to already differ from the control participants years before infection. This highlights the difficulties in interpreting cross-sectional post-infection imaging differences as being necessarily the consequence of the infection itself. When looking at brain imaging baseline differences between the two groups across all IDPs, particularly in an age-modulated manner, we did find a few further significant baseline differences beyond the volume of the thalamus (Supplementary Table 2). These were principally using diffusion imaging, but also using grey matter volume in the subcortical structures. Importantly, none of these baseline imaging differences spatially overlapped with the regions that were found to be different longitudinally (Supplementary Fig. 2). However, as this study is observational, as opposed to a randomised interventional study, one cannot make claims of disease causality with absolute certainty, but interpretational ambiguities are greatly reduced compared with post hoc cross-sectional studies. The question remains as to whether the two groups are actually perfectly matched, as controls and cases could not be randomised a priori. Across the main risk factors, as well

as thousands of lifestyle, health data and environment variables available in UK Biobank, we did not identify any significant differences when looking at each variable in isolation (only a few variables showed some trends at $P_{uncorr} < 0.001$; Supplementary Table 4). This does not preclude the possibility of a subthreshold pattern of baseline differences making one group more at risk of being infected by SARS-CoV-2, and this risk perhaps interacting with the effects of the coronavirus. This motivated the use of principal component analyses, which revealed two significant components suggesting subtle lower cognitive abilities in the participants who were infected later on (Supplementary Analysis 1). Importantly, neither of these two cognitive components had any bearing on our longitudinal imaging results (reducing at most the strength of $Z$-statistics from $Z = 4$ to $Z = 3.77$ for crus II of the cerebellum, when added in as an extra confound to the longitudinal analysis). Whether any of these imaging and cognitive differences at baseline had a subsequent role in those patients being more likely to be infected by the coronavirus, or to develop symptoms from infection, needs further investigation.

Our cohort-based, quantitative imaging study, in contrast to the majority of single-case and case-series studies published so far, does not focus on gross abnormalities that could be observed at the single-participant level with the naked eye, such as microhaemorrhages or (sub)acute ischaemic infarctions[9]. However, it does rely on an anatomically consistent pattern of abnormalities caused by the disease process, a common spatial distribution of these pathological alterations across the infected participants, which could be uncovered by aligning all of the images together in a common space, followed by applying a pipeline of modality-specific image-processing algorithms. This automated, objective and quantitative processing of the images facilitates the detection of subtle changes that would not be visible at the individual level, but which point to a possible mechanism for the neurological effects of the coronavirus infection. Our hypothesis-driven analyses revealed a clear involvement of the olfactory cortex, which was also found in the exploratory analyses and the vertex-wise and voxel-wise maps of cortical thickness and mean diffusivity. Although no differences were seen in the olfactory bulbs or piriform cortex per se (both of which are located in a region above the sinuses that is prone to susceptibility distortions in the brain images, and both are difficult to segment in MRI data), we identified significant longitudinal differences in a network of regions that is functionally connected to the piriform cortex, mainly consisting of the anterior cingulate cortex and orbitofrontal cortex, as well as the ventral striatum, amygdala, hippocampus and parahippocampal gyrus[30]. Some of the most consistent abnormalities across hypothesis-driven and exploratory analyses, and all of the group comparisons, were revealed in the left parahippocampal gyrus (Table 4, Fig. 2 and Supplementary Table 1)—a limbic region of the brain that has a crucial, integrative role for the relative temporal order of events in episodic memory[34–36]. Importantly, it is directly connected to the piriform cortex and entorhinal cortex, which are both part of the primary olfactory cortex[24,37]. Similarly, the orbitofrontal cortex, which we also found was altered in the SARS-CoV-2-positive group, is often referred to as the secondary olfactory cortex, as it possesses direct connections to both the entorhinal and piriform cortex[37], as well as to the anterior olfactory nucleus[23,30]. In fact, in a recent functional connectivity study of the primary olfactory cortex, the orbitofrontal cortex was found to be connected to all four primary olfactory regions investigated (frontal and temporal piriform cortex, anterior olfactory nucleus and olfactory tubercle), possibly explaining why it is reliably activated even in basic and passive olfactory tasks[30]. Using the same olfactory connectivity maps, which overlap cortically in the orbitofrontal cortex, anterior cingulate cortex and insula, we found a more pronounced increase in diffusion metrics indicative of tissue damage in the SARS-CoV-2 group. The voxel-wise map of mean diffusivity revealed that these longitudinal differences were located in the orbitofrontal and anterior cingulate cortex, as well as in the insula and

the amygdala. The insula is not only directly connected to the primary olfactory cortex[23], but is also considered to be the primary gustatory cortex. 'Area G' (that is, the dorsal part of the insula at the junction with the frontal and parietal operculum), in turn, connects with the orbito-frontal cortex[38]. The vertex-wise and voxel-wise visualisation of both greater loss of grey matter and increase in mean diffusivity in the insula spatially correspond in particular to the area of consistent activation to all basic taste qualities[39]. Finally, the exploratory analysis revealed a more pronounced loss of grey matter in crus II, part of the cognitive and olfactory-related lobule VII of the cerebellum[40]. These results are consistent with previous post-infection PET findings showing, in more severe cases, FDG hypometabolism in the insula, orbitofrontal and anterior cingulate cortex, as well as lower grey matter volume in the insula and hippocampus[41,42].

Early neurological signs in COVID-19 include hyposmia and hypo-geusia, which appear to precede the onset of respiratory symptoms in the majority of affected patients[2,20,43]. Furthermore, a heavily debated hypothesis has been that an entry point of SARS-CoV-2 to the central nervous system is through the olfactory mucosa, or the olfactory bulb[2,11,20] (the coronavirus itself would not necessarily need to enter the central nervous system; anterograde degeneration from olfactory neurons might suffice to generate the pattern of abnormalities revealed in our longitudinal analyses). The predominance observed in other studies of hyposmic and anosmic symptoms—whether caused directly by loss of olfactory neurons or by perturbation of supporting cells of the olfactory epithelium[17,22]—could also, through repeated sensory deprivation, lead to a loss of grey matter in these olfactory-related brain regions. A highly focal reduction in grey matter in the orbito-frontal cortex and insula has been observed, for example, in patients with severe olfactory dysfunction in a cross-sectional study of chronic rhinosinusitis[29]. A more extensive study of congenital and acquired (post-infectious, chronic inflammation due to rhinosinusitis or idi-opathic) olfactory loss also demonstrated an association between grey matter volume and olfactory function in the orbitofrontal cortex[19]. It also showed that the duration of olfactory loss for those with acquired olfactory dysfunction, ranging from 0 to over 10 years, was related to a more pronounced loss of grey matter in the gyrus rectus and orbitofron-tal cortex[19]. By contrast, it has been reported in a longitudinal study that patients with idiopathic olfactory loss had higher grey matter volume after undergoing olfactory training in various brain regions including the orbitofrontal cortex and gyrus rectus[44]. This raises the interesting possibility that the pattern of longitudinal abnormalities observed here in the limbic, olfactory brain regions of SARS-CoV-2-positive participants, if they are indeed related to olfactory dysfunction, might be attenuated over time if the infected participants go on to recover their sense of smell and taste. For example, there is some very preliminary evidence, in a few previously hospitalised patients with COVID-19, that brain hypometabolism becomes less pronounced when followed up 6 months later, even if it does not entirely resolve[41,45]. In our milder cohort, structural (as opposed to functional) changes might take longer and require larger numbers to be detected. When we tested whether the time between infection and the second brain scan had any relationship—positive, indicative of recovery, or negative, indicative of an ongoing degenerative process—with the grey matter loss or increase in diffusivity in the significant IDPs, we found no signifi-cant effect. This result is also possibly due to the relatively small range in the interval between infection and second brain scan at the time of this study—between 1 and 13 months for those 351 infected participants for whom we had a diagnosis date and, particularly, less than 20% of these participants had been infected more than 6 months prior to their second scan (Supplementary Fig. 3). Another source of variability is that each individual in our cohort was infected between the months of March 2020 and April 2021, periods that saw various dominant strains of SARS-CoV-2. Of those 351 participants for whom we have a proxy date of infection, but no formal way of assessing the strain responsible for the

infection, a small minority of the participants were probably infected with the original strain, and a majority with the variants of concern present in the UK from October 2020 onwards (predominantly Alpha, but also Beta and Gamma), while presumably very few participants, if any, were infected with the Delta variant, which appeared in the UK in April 2021. As the second scans were acquired over a relatively short period in these positive participants (February–May 2021), SARS-CoV-2 strains and the time between infection and second scan are also highly collinear. Additional follow-up of this cohort, not only increasing the number of cases that became infected 6 months or longer before their second scan, but also including individuals infected by the Delta variant, would be particularly valuable in determining the longer-term effects of infection on these limbic structures, as well as possible differential effects between the various strains.

Various possible explanations for our longitudinal brain results are provided in the Supplementary Discussion.

Many of our results were found using imaging biomarkers of grey matter thickness or volume, which can be sensitive markers of a neurodegenerative process compared with other imaging modali-ties[46], and are robust measurements that makes them ideal in a lon-gitudinal setting[47]. In fact, the longitudinal differences between the SARS-CoV-2-positive and control groups, although significantly local-ised in a limbic olfactory and gustatory network, seemed also—at a lower level—to be generalised, as illustrated by the significant shift in the distribution of Z values over the entire cortical surface (Supplemen-tary Fig. 4). This means that there is an overall stronger decrease in grey matter thickness across the entire cortex in the infected participants, but that this effect is particularly dominant in the olfactory system. A marked atrophy of fronto-parietal and temporal regions can also be seen when contrasting hospitalised and non-hospitalised cases, suggesting that there is increased damage in the less mild cases, with an additional significant shift in Z values (Supplementary Fig. 4). The pattern of loss of grey matter in the hospitalised patients compared with the milder cases is consistent with PET-FDG reports showing fronto-parietal and temporal decrease in glucose in hospitalised patients with COVID-19[18,45].

The overlapping olfactory- and memory-related functions of the regions shown to alter significantly over time in SARS-CoV-2—including the parahippocampal gyrus/perirhinal cortex, entorhinal cortex and hippocampus in particular (Supplementary Table 1)—raise the possi-bility that longer-term consequences of SARS-CoV-2 infection might in time contribute to Alzheimer's disease or other forms of dementia[2]. This has led to the creation of an international consortium including the Alzheimer's Association and representatives from more than 30 countries to investigate these questions[2]. In our sample of partici-pants who mainly had mild infection, we found no signs of memory impairment. However, these participants who tested positive for SARS-CoV-2 showed a worsening of executive function, taking a sig-nificantly greater time to complete trail A and particularly trail B of the Trail Making Test (Fig. 3). These findings remained significant after excluding the few hospitalised cases. Although the UK Biobank version of the Trail Taking Test is carried out online and unsupervised, there is good to very good agreement with the standard paper-and-pencil Trail Making Test on its measurements for completion of the two trails[48], two measures that are known to be sensitive to detect impairment of executive function and attention, for example, in affective disorders and in schizophrenia[49,50], and to discriminate mild cognitive impairment and dementia from healthy ageing[51]. In turn, the duration to complete the alphanumeric trail B was associated post hoc with the longitudinal changes in the cognitive part of the cerebellum, namely crus II, which is also specifically activated by olfactory tasks[40,52]. Consistent with this result, this particular part of the cerebellum has been recently shown to have a key role in the association with (and prediction of future) cogni-tive impairment in patients with stroke (subarachnoid haemorrhage)[53]. By contrast, the parahippocampal gyrus and other memory-related

regions did not show in our study any alteration at a functional level, that is, any post hoc association with the selected cognitive tests. It remains to be determined whether the loss of grey matter and increased tissue damage seen in these specific limbic regions may in turn increase the risk for these participants of developing memory problems[54], and perhaps dementia in the longer term[2,4,55].

The limitations of this study include a lack of stratification of the severity of the cases, beyond the information of whether they had been hospitalised (information on $O_2$ saturation levels and details of treatment or hospital procedures is currently available for only a few participants); a lack of clinical correlates as they are not currently available as part of the UK Biobank COVID-19-related links to health records (of particular relevance, potential hyposmic and hypogeusic symptoms and blood-based markers of inflammation); a lack of identification of the specific SARS-CoV-2 strain that infected each participant; a small number of participants from Asian, Black or other ethnic background other than white; and some of the cases' and controls' SARS-CoV-2 infection status was identified using antibody lateral flow test kits that have varied diagnostic accuracy[56]. However, note that any potential misclassification of controls as positive cases (due to false positives in testing) and positive cases as controls (due to the absence of confirmed negative status and/or false negative tests) could only bias our results towards the null hypothesis of no difference between cases and controls. For cases, no distinction is possible at present to determine whether a positive test is due to infection or vaccination, so potential cases identified only through lateral flow test in vaccinated participants were not included in this study. Information on the vaccination status, and how both vaccination dates might interact with the date of infection, is also currently unavailable. Although the two groups were not significantly different across major demographic and risk-factor variables, we identified a subtle pattern of lower cognitive abilities in the participants who went on to be infected, but this could not explain away our longitudinal findings. The individuals who were later infected also showed a lower subcortical volume, and higher diffusion abnormalities at baseline compared with the control individuals, in brain regions that did not overlap with our longitudinal results. One issue that is inherent to the recruitment strategy of UK Biobank, based on participants volunteering after being contacted at home for a possible re-imaging session, is the high number of mild cases. However, this can be seen as a strength of this study: the majority of the brain imaging publications to date have focussed on moderate to severe cases of COVID-19[9]; there is therefore a fundamental need for more information on the cerebral effects of the disease in its milder form. The UK Biobank COVID-19 re-imaging study is ongoing, and further information will eventually be made available. For the statistical approach, we chose a model form given strong priors of highly increased detrimental effects of SARS-CoV-2 and greater vulnerability of the brain with age. Using this objective model and rigorous statistical inference, we found significant and interpretable results. We have not tested all possible models for all of the possible IDPs; instead, we focussed on one possible model drawn from independent, existing literature and found that it is useful, that is, statistically significant. The model may not be optimal for every feature considered; in other words, this model might not be the most sensitive possible model for every IDP. However, the main expected outcome of using a suboptimal model would be that we would fail to find significant results, and not that there would be any inflation of false-positive results. Finally, on the imaging side, our exploratory approach revealed significant longitudinal differences in the volume of the whole brainstem, but the UK Biobank scanning protocol and processing does not allow us to clarify which specific nuclei (for example, potentially those that are key autonomic and respiratory control centres) might be involved, with the exception of the substantia nigra.

This is to our knowledge the first longitudinal imaging study comparing brain scans acquired from individuals before and after SARS-CoV-2 infection with those scans from a well-matched control group. It also is one of the largest COVID-19 brain imaging studies, with 785 participants including 401 individuals infected by SARS-CoV-2. Its unique design makes it possible to more confidently tease apart the pathogenic contribution associated, directly or indirectly, with the infection from pre-existing risk factors. By using automated, objective and quantitative methods, we uncovered a consistent spatial pattern of longitudinal abnormalities in limbic brain regions forming a mainly olfactory network. Whether these abnormal changes are a hallmark of the spread of the pathogenic effects, or of the virus itself in the brain, and whether these abnormalities may indicate a future vulnerability of the limbic system in particular, including memory, for these participants remains to be investigated.

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

## Methods

### Ethics

Human participants: UK Biobank has approval from the North West Multi-Centre Research Ethics Committee (MREC) to obtain and disseminate data and samples from the participants (http://www.ukbiobank.ac.uk/ethics/), and these ethical regulations cover the work in this study. Written informed consent was obtained from all of the participants.

### Study design

As part of the UK Biobank imaging study[60], thousands of participants had received brain scans before the start of the COVID-19 pandemic. Multimodal brain imaging data, collected at four sites with identical imaging hardware, scanner software, and protocols, and passing quality controls, were obtained from 42,729 participants over the age of 45 years, and made available to researchers worldwide.

Before the COVID-19 pandemic, longitudinal (first- and second-time-point scanning) had already begun in the UK Biobank imaging study, with about 3,000 participants returning for a second scan before scanning was paused in 2020 as a result of the pandemic. More recently, starting in February 2021, hundreds of UK Biobank participants who had already taken part in UK Biobank imaging before the pandemic were invited back for a second scan (the response rate was 60% for the cases, and 55% for the controls). This COVID-19 re-imaging study was set up to investigate the effects of SARS-CoV-2 infection by comparing imaging scans taken from the participants before versus after infection.

The full list of inclusion criteria for the participants in this re-imaging study is as follows (further details are provided in the online documentation: https://biobank.ndph.ox.ac.uk/showcase/showcase/docs/casecontrol_covidimaging.pdf):

- Had already attended an imaging assessment at one of the three imaging sites (the fourth opened just before the pandemic began).
- Still lived within the catchment area of the clinic they attended for their first imaging assessment.
- Had no incidental findings identified from their scans taken at the first imaging visit.
- Had not withdrawn or died.
- Had a valid email and postal address.
- Had high-quality scans from the first imaging visit.
- Lived within 60 km of the clinic (extended to 75 km in Feb 2021), due to travel restrictions during the lockdown period.

Among those, some participants were identified as having been infected with SARS-CoV-2 based on: (1) the results of diagnostic antigen tests identified through linkage to health-related records, (2) their primary care (GP) data or hospital records, or (3) the results of two antibody tests.

The diagnostic antigen tests results data for England, Scotland and Wales are made available on an ongoing basis by UK Biobank, and these data are provided by Public Health England (PHE), Public Health Scotland (PHS) and Secure Anonymised Information Linkage (SAIL, the databank from Wales), respectively. The data contain information on the date when the specimen was taken, origin (binary code for whether the patient was an inpatient when the specimen was taken) and result (binary code or positive and negative for SARS-CoV-2) of the tests along with encoded participant IDs. Further information on how regular updates of SARS-CoV-2 test results in England are made in UK Biobank is available online (biobank.ctsu.ox.ac.uk/crystal/ukb/docs/c19link_phe_sgss.pdf).

For the primary care (GP) data, UK Biobank used the following set of codes: (1) TPP: Y213a, Y228d, XaLTE (if the event date was after 1 January 2020), Y22b8, Y23f7, Y20d1, Y24ad, Y246f, Y269d, Y23f0, Y2a3b, Y2a15, Y212f, Y26a1, Y26b2, Y23e9, Y211c, Y23ec, Y2a3d; (2) EMIS: EMISNQCO303, 720293000, 720294006, 840535000, 840536004, 870361009, 870362002, 871552002, 871553007, 871555000, 871556004, 871557008, 871558003, 871559006, 871560001, 871562009,

1240581000000104, 1300721000000109, 1321541000000108, 1321551000000106, 1321661000000108, 1324881000000100. For the hospital records, the code used to identify positive SARS-CoV-2 cases was ICD10: U07.1. The dates of the records for both GP and hospital data were extracted along with the encoded participant IDs. In particular, the hospital records contain information on admission and discharge, including episode start and end dates, primary and secondary causes for admission, critical care if applicable and types of operations or procedures performed. We first identified hospitalised infected patients who had the ICD code U07.1 as a primary or secondary cause, and extracted information (such as admission/discharge date) relating to the episodes. OPCS-4 codes E85.1, E85.6, E85.2 and X52.9, as well as X52.8, E85.8, E85.9, E87.8 and E87.9, were used to find out whether the patients were provided respiratory support during the episodes. No other information, for example symptoms such as hyposmia or hypogeusia of particular relevance, was made available in these medical records.

Participants were also invited to take a home-based lateral flow test (Fortress Fast COVID-19 Home test, Fortress Diagnostics and ABC-19TM Rapid Test, Abingdon Health) to detect the presence of SARS-CoV-2 antibodies. A second kit was sent to all of the participants who recorded an initial positive result and who had indicated they had not yet been vaccinated, to reduce the number of false positives.

Participants were classified as SARS-CoV-2-positive cases if they had a positive test record in any of the three data sources described above. Date of diagnosis (Table 2) was determined on the basis of the information available in the public health-related records (1) and primary care and hospital records (2). For participants with multiple positive test records, we took the earliest date as the date of diagnosis.

Control participants were then selected by identifying, from the remaining previously imaged UK Biobank participants, those who had a negative antibody test result, as determined from the home-based lateral flow kits, and/or who had no record of confirmed or suspected COVID-19 from primary care, hospital records or diagnostic antigen test data. Control participants were selected to match 1:1 to positive SARS-CoV-2 cases according to five criteria: sex; ethnicity (white/non-white, as numbers were too low to allow for a finer distinction); date of birth (±6 months); location of first imaging assessment clinic; and date of first imaging assessment (±6 months).

Permission to use the UK Biobank Resource was obtained through Material Transfer Agreement (www.ukbiobank.ac.uk/media/p3zffurf/biobank-mta.pdf).

### Image processing

For this work, we primarily used the IDPs generated by our team on behalf of UK Biobank, and made available to all researchers by UK Biobank[60,61]. The IDPs are summary measures, each describing a different aspect of brain structure or function, depending on what underlying imaging modality was used[60,61].

The protocol includes three structural MRI scans (T1, T2 fluid attenuation inversion recovery (FLAIR) and susceptibility-weighted MRI), as well as diffusion MRI, and resting and task functional MRI. T1 scans make it possible to derive global measures of brain and CSF volumes, as well as localised measures of grey matter volume and cortical thickness. The T2 FLAIR scan identifies differences that might be indicative of inflammation or tissue damage. Susceptibility-weighted MRI is sensitive to iron and myelin content. Diffusion MRI measurements provide insights into the tissue microstructure integrity. Resting-state functional MRI is performed on an individual who is not engaged in any particular activity or task, and can provide indices related to the functional connectivity between brain regions[62]. Functional connectivity is intrinsically noisy when each region–pair connection is considered individually, so we focussed here our analysis on 6 dimensionally reduced functional connectivity networks[59]. We also did not consider a priori task-fMRI activation IDPs, as these

have previously been found to have very low reproducibility and heritability[63].

We used 1,524 existing UK Biobank IDPs, including: regional grey matter, brain and CSF volume, local cortical surface area, volume and thickness, cortical grey–white contrast, white matter hyperintensity volume, white matter microstructural measures such as fractional anisotropy and mean diffusivity, resting-state amplitude and dimensionally reduced connectivity measures. Furthermore, we also generated 1,106 new IDPs, as described below.

We computed additional IDPs obtained using quantitative susceptibility mapping (QSM), which has recently been added into our UK Biobank processing pipeline[64]. Magnitude and phase data from the susceptibility-weighted MRI acquisitions were processed to provide quantitative measures reflecting clinically relevant tissue susceptibility properties. Median T2* was calculated within 17 subcortical structures (with their regions of interest (ROIs) estimated from the T1) as IDPs; 14 of these are the same subcortical regions that were already estimated by the core UK Biobank pipeline, and here we added 3 more subcortical ROIs: left and right substantia nigra[65] and regions of white matter hyperintensities (lesions)[66]. Second, susceptibility-weighted MRI phase data were processed for QSM following a pipeline that was recently developed for UK Biobank[27,67]. QSM (CSF-referenced) IDPs were calculated in the same 17 subcortical structures as the T2* IDPs.

Additional IDPs were created using subsegmentations of the hippocampus, amygdala and thalamus as implemented in FreeSurfer[68–71]. We extracted these ROI masks from the FreeSurfer processing and applied them to the T2* and diffusion images (diffusion tensor model: MD and FA; NODDI model: OD, ISOVF, ICVF) to generate additional subcortical IDPs.

Finally, we generated new IDPs tailored to the olfactory and gustatory systems, as described below.

### Hypothesis-driven approach

On the basis of prior expectations from animal models and post mortem findings, we chose to focus a priori our primary analyses on a subset of 332 ROIs (297 of which passed the reproducibility thresholding; see the 'Reproducibility' section below) from the available 2,630 IDPs[23,24,38]; these correspond anatomically to the telencephalic primary and secondary connections of the olfactory and gustatory cortex. In brief, these include the piriform cortex, parahippocampal gyrus, entorhinal cortex, amygdala, insula, frontal/parietal operculum, medial and lateral orbitofrontal cortex, hippocampus and basal ganglia. As no labelling of the piriform cortex exists in any of the atlases used in the UK Biobank imaging processing, we refined a previously published ROI of the piriform cortex (frontal and temporal), anterior olfactory nucleus and olfactory tubercle, by limiting it to the cortical ribbon of our UK Biobank T1-weighted standard space (https://github.com/zelanolab/primaryolfactorycortexparcellation[30]). We further used maps from the same study's resting-state fMRI analysis of the functional connectivity of each of the four parts of this ROI (piriform frontal, piriform temporal, anterior olfactory nucleus and olfactory tubercle) to the rest of the brain to generate four additional extended ROIs of the functionally connected cortical and subcortical regions to these primary olfactory areas[30]. For this, we thresholded their connectivity $t$-value maps to retain only significant voxels ($P_{FWE} < 0.05$, with threshold-free cluster enhancement), and used the maps as weighted (and, separately, binarised) masks, to further extract grey matter volume, T2* and diffusion values; this was done by (1) regressing each of these maps into the GM, T2* or diffusion images in their respective native spaces and, separately, (2) by binarising the maps and extracting mean and 95th percentile values.

Moreover, masks for the left and right olfactory bulbs were generated by manually drawing a binary mask for the right olfactory bulb on an averaged template-space T2 FLAIR volume generated from 713 UK Biobank participants, and mirroring this to obtain the mask for the left (having confirmed by visual inspection that the symmetry in this region allowed for this to be effective). Both masks were then modulated by the T2 intensities in their respective ROIs to account for partial volume effects, generating the final label maps with values ranging between 0 and 1. For the hypothalamus, we combined and refined ROIs from two previously published and publicly available atlases of a probabilistic hypothalamus map (https://neurovault.org/collections/3145/[65]) and hypothalamic subregions[72]. Both the probabilistic hypothalamus map and the binarised map obtained from fusing the 26 hypothalamic subregions were transformed to our standard space in which the probabilistic map was then masked by the binarised map. We then extracted volume, and T2 mean and 95th percentile intensity measurements in the participants' native spaces, using the olfactory bulb and hypothalamus maps (unthresholded and thresholded at 0.3 to reduce concerns about arbitrariness of threshold selection when rebinarising these very thin ROI masks after interpolation, a step which is unavoidable when transforming masks from one space to another). For the hypothalamus, we also extracted these metrics from T2* and diffusion images. All of the above preprocessing steps were defined and completed before any analyses of longitudinal change and case–control modelling.

The full list of 297 predetermined and reproducible IDPs is provided in Supplementary Table 1.

### Exploratory approach

The full set of 2,630 IDPs described above was used for a more exploratory, inclusive analysis of SARS-CoV-2 infection effects on brain structure and function (the full list of reproducible IDPs is provided in Supplementary Table 1).

### Statistical modelling

The following modelling was applied in the same manner to both the hypothesis-driven analyses of a subset of IDPs, and the all-IDP exploratory analyses.

**Outlier identification of the IDPs.** All of the IDPs from all of the participants were pooled for initial processing (at this stage blinded to the SARS-CoV-2 status of the participants): 42,729 scan 1 datasets (all pre-pandemic); 2,943 pre-pandemic scan 2 datasets; and 890 scan 2 datasets acquired after the beginning of the COVID-19 pandemic. Outlier values (individual IDPs from individual scanning sessions) were removed on the basis of being more extreme than eight times the median absolute deviation from the median for a given IDP. Missing data for individual participants and specific IDPs can therefore occur because of this step, or because the IDP was missing in the original data (for example, because a given modality was not usable from a given participant). The fraction of total non-missing data, averaged across IDPs, is 0.93; all full results tables include the number of usable measurements for each IDP and for each statistical test. Importantly, there was no imbalance in amount of missing/outlier data between cases and controls: the number of cases with usable data, normalised by the total number of participants with usable data, has the following percentiles across IDPs: percentiles [0, 1, 50, 99, 100] = 0.50, 0.50, 0.52, 0.52, 0.60, that is, the median percentile is 0.52. From this analysis, the only three IDPs for which this fraction was greater than 0.53 were thalamic nuclei diffusion IDPs, which do not appear in any of our main results. These are also the only three IDPs with more than 24% missing/outlier data.

The IDPs from the 890 participants imaged during the pandemic (SARS-CoV-2-positive cases and controls), from both time points, were then retained. Participants were retained if at least the T1-weighted structural image was usable from both time points, resulting in IDPs at both time points (IDP1 and IDP2) from 785 participants. The data were then pooled into a single dataset comprising 785 × 2 = 1,570 imaging sessions, and cross-sectional deconfounding, treating all scans equivalently, was performed for head size, age, scanner table position and

image motion in the diffusion MRI data. This deconfounding is part of the data preprocessing, and is done at the level of individual scan sessions; thus, this needs to be carried out before combining all scans and participants together in the main modelling. These imaging confound variables first had outlier removal applied as described above, although using a higher threshold of 15 times the median absolute deviation, because some important confounds have extremely non-Gaussian underlying distributions (such as MRI scanner table position), and we found that a threshold of 8 was too aggressive for these variables for values that are perfectly acceptable when considered with the domain knowledge of these variables[61,73].

**Reproducibility of the IDPs.** We next evaluated the scan–rescan reproducibility of IDPs to discard IDPs that were not reasonably reproducible between scans. For each IDP, we correlated the IDP1 with IDP2 values, separately for cases and controls, resulting in two reproducibility measures (Pearson correlation $r$) for each IDP. The vectors of $r$ values (one value per IDP) derived from cases and from controls were extremely highly correlated ($r = 0.98$), showing that potential effects associated with infection are subtle compared with between-subject variability and IDP noise; we therefore averaged these cases and controls' $r$ values to give a single reproducibility measure for each IDP. From the initial set of 2,630 IDPs, the least reproducible IDPs ($r < 0.5$) were discarded, leaving 2,048 IDPs. Finally, IDPs with high levels of missing data (usable values from fewer than 50 participants) were discarded, leaving in total 2,047 IDPs.

**Main longitudinal model, deconfounding.** Despite initial case–control participant pairing (resulting in case and control groups being well matched), missing/outlier data potentially disrupted this exact paired matching, and we therefore also included in the modelling confound variables derived from those factors that were originally used as pairing criteria: the difference between the participants' ages at each of their two scans, the difference in the squares of the ages (to account for quadratic dependencies of IDPs on age), genetic sex and ethnicity (white versus non-white).

Longitudinal IDP change ($\Delta$IDP) was estimated by regressing IDP2 on IDP1 (ref. [74]), as well as including in the regression the confound variables listed above.

The case-versus-control difference in this longitudinal IDP effect was modelled with a group difference regressor comprising the case-versus-control binary variable modulated by a function of age at scan 2 (Age2, a close proxy for age at infection for the SARS-CoV-2 group, with an error of less than a year). We chose to focus on an objective age model given the strong prior knowledge of a highly increased detrimental effect, at older ages, of SARS-CoV-2 infection and a greater vulnerability of the brain with age. The age dependence has been found to be exponential in studies of the effects of COVID-19 on hospitalisation and fatality rate[75,76]. Here we used the exact age dependence found by a data-driven meta-regression of 28 studies, with no free or subjectively chosen parameters, to modulate the binary case-versus-control variable, based on age at scan 2 (ref. [76]).

The main case-versus-control group difference regressor of interest is therefore:

$$\text{Case-versus-Control} = \text{demeaned(Case-versus-Control binary variable)} \times 10^{\text{Age2} \times 0.0524 - 3.27}, \quad (1)$$

where the age-dependence constants are taken from the meta-regression analysis[76] (Supplementary Analysis 5). To ensure that the fitting of this term is not influenced by an effect that is common to controls and cases, we added a matching confound variable of $10^{\text{Age2} \times 0.0524 - 3.27}$, that is, the same ageing term without the group-difference multiplier.

Our main model of interest therefore simply combines IDP1 and IDP2, the above group-difference model and the confounds matrix:

$$\text{IDP2} \sim \text{Case-versus-Control} + \text{IDP1} + \text{Confounds}, \quad (2)$$

where the confound matrix comprises the terms described above: Age2 − Age1, Age2$^2$ − Age1$^2$, ethnicity, sex and $10^{\text{Age2} \times 0.0524 - 3.27}$.

By using a simple, single case-versus-control regressor for the main effect of interest, we optimised power for finding effects that follow this form, at the risk of suboptimal power (sensitivity to finding true effects) if the effect does not follow this form.

Many forms for the case-versus-control model might be used. Possible models include: a binary regressor; single-regressors with age-modulated differences (such as the one primarily used here); more flexible models with multiple-regressors. Without testing a huge number of possible different models, one cannot make claims of absolute optimality. Nevertheless, our primary aim is not to prove model optimality, but to identify the effects of disease. To that aim, we have found statistically significant results with the simple model used here. Importantly, the exact choice of exponential model also had little bearing on our findings. Even opting for a binary case-versus-control regressor—that is, without any age modulation—yielded similar, albeit a little weaker, primary results, consistent with our expectation of increased effects at higher ages (further details and discussion of non-modulated modelling results are provided in Supplementary Table 5 and Supplementary Analysis 5). Supplementary Analysis 6 provides further model-fitting validity and robustness evaluations, including diagnostic residual scatter-plots and residual $QQ$ plots, showing no obvious evidence of structured problems in model residuals or of model misspecifications.

The group-difference regressor is scaled to have average peak–peak height 1, such that the regression parameter from fitting case-versus-control can easily be converted into a percentage change measure, when normalised by the mean baseline value for a given IDP. For the main longitudinal modelling, this represents the average group difference in the longitudinal IDP change and, for the separate modelling of baseline IDPs only, this percentage reflects the average group difference in the baseline values. In addition to reporting percentage effects and associated s.e. values, we also report the statistical significance as $Z$-statistics (Gaussianised regression model $t$-statistics) and $P$ values. Here, $Z$ is more useful than $t$, because different IDPs have different patterns of missing data and, therefore, $Z$ is more usefully comparable across IDPs. The regression inference automatically takes care of the degrees-of-freedom, including accounting for missing data and confound variables. For each IDP, any missing data are ignored (that participant is left out for that analysis). As part of the estimation of the longitudinal IDP changes, $\Delta$IDP outliers (for each IDP, and each participant) were removed (set as missing), if they were more than 8 times the median absolute deviation from the median.

**Correction for multiple comparisons.** We used permutation testing to estimate $P_{\text{FWE}}$ values, that is, correcting for the multiple comparisons across IDPs while accounting for the dependences among IDPs. We randomly permuted the residualised case-versus-control regressor relative to the residualised IDP2s, with 10,000 random permutations. At each permutation, we computed the association $Z$ value for each IDP, and recorded the maximum absolute value across all IDPs. By taking the absolute value, we corrected for the two-tailed nature of the test, that is, we did not pre-assume the direction of any effect. After building up the null distribution of the maximum $|Z|$ across IDPs, we then tested the original $|Z|$ values against this distribution to obtain $P_{\text{FWE}}$ values, fully correcting for multiple comparisons across all IDPs. We also computed for each test the FDR at 5%, generating a threshold that can be applied to uncorrected $P$ values to determine their FDR significance.

We therefore computed both FDR- and FWE-corrected inferences as two distinct measures of strength of evidence for a given effect. In this

study, we primarily rely on FDR correction, which provides good power while controlling for multiple testing in a principled manner, but we wish to also indicate when a result additionally attains FWE significance. We therefore always specify the findings obtained using both correction methods in the main text and Supplementary Tables 1, 2, 3 and 5.

**Group comparisons.** In the rest of the manuscript, we refer to the main age-modulated group comparison analysis (comparing IDPs at second time point controlling for IDPs at baseline) between SARS-CoV-2-positive cases and control individuals, as described above, as Model 1.

As secondary follow-up analyses, we also applied the same hypothesis-driven and exploratory approaches as described above to compare non-hospitalised SARS-CoV-2-positive cases against controls (Model 2), and hospitalised patients against controls (Model 3). Separately, we also carried out the same analysis between hospitalised and non-hospitalised cases, adding as covariates three risk factors showing significant differences between these two SARS-CoV-2 groups (Model 4). For these secondary models (2–4), we again used age-modulated group-difference regressors as described above for Model 1. The power to detect effects in the two latter models, considering the hospitalised patients as a separate group, is of course considerably reduced given the small number of hospitalised cases in this cohort.

For all 4 models, testing was carried out twice: first using the a priori focussed subset of IDPs identified for the hypothesis-driven analyses, and then using the full set of IDPs for the exploratory analyses. In both cases, IDPs were identified as having significant group differences, corrected for multiple comparisons.

Thus, we carried out eight imaging group comparison longitudinal analyses:

- The primary analysis comparing all cases versus all controls (Model 1), first in the set of olfactory-related IDPs a priori drawn, then in the exploratory set of IDPs.
- Secondary ancillary analyses, using both hypothesis-driven and exploratory sets of IDPs:
  - All non-hospitalised cases versus all controls (Model 2).
  - All hospitalised cases versus all controls (Model 3).
  - All hospitalised cases versus all non-hospitalised cases (Model 4).

### Cognitive analysis

Although cognitive testing offers limited measurements of cognitive function in UK Biobank, we included in our ancillary cognitive analysis 10 variables sensitive to cognitive impairment. For this, we drew these variables using a data-driven approach based on identifying out-of-sample current and future dementia cases in UK Biobank, and comparing them to matched control individuals (Supplementary Analysis 2). The top most significant variables from this out-of-sample analysis were:

- Three variables from the UK Biobank Trail Making Test: both durations to complete trails A and B, as well as the total number of errors made traversing trail B.
- One variable from the Symbol Digit Test: the number of symbol digit matches made correctly.
- One measure of reaction time: mean time to correctly identify matches at the card game 'Snap'.
- One measure of reasoning: the 'fluid intelligence' score.
- One measure of numeric memory: the maximum number of digits remembered correctly.
- Three variables of the pairs matching test: numbers of correct and incorrect matches, and the time to complete the test.

On the basis of these 10 variables from 6 different cognitive tests, we carried out two analyses: (1) the same group comparison between SARS-CoV-2 cases and controls of the longitudinal effect as described above, but substituting ΔCOG for ΔIDP, (2) a post hoc regression analysis, in the SARS-CoV-2 group only, of the ΔCOG showing the most significant difference between cases and controls against the top 10 most

significant ΔIDPs for the hypothesis-driven approach and the top 10 for the exploratory approach. All of the results were evaluated for FWE and FDR significance, correcting for multiple comparisons across all cognitive or IDP variables where applicable.

### Additional analyses

**Baseline group comparisons. Risk factors.** We compared the group positive for SARS-CoV-2 and the control group at baseline across common risk factors for infection and severity of disease: age, sex, blood pressure (systolic and diastolic), weight (including BMI and waist–hip ratio), diabetes, smoking, alcohol consumption and socioeconomic status (using the Townsend deprivation index). For this, we used the 'last observation carried forward' (LOCF) imputation method, for which we considered all the values available closest to the scan 1 visit (for the majority of the values, these were available from the same visit, on the same day that scan 1 was acquired); we also tested that there was no difference between the group with SARS-CoV-2 and the control group in the distribution of the visits used to collect the LOCF values.

**All other non-imaging phenotypes.** We also examined whether the group with SARS-CoV-2 and the control group differed at baseline across all non-imaging phenotypes (lifestyle, environmental, health-related, dietary), across all UK Biobank visits. We assessed the 6,301 pre-scan 2 non-imaging phenotypes with at least 3% of values distinct from the majority value, and the results were corrected for multiple comparisons using FDR and FWE (that is, where relevant, we refer to both in the main text).

**IDPs.** To complement our longitudinal analyses, we performed a baseline-only (and, separately, second time point only) cross-sectional group comparison between SARS-CoV-2 cases and controls, across all 2,047 IDPs, correcting for multiple comparisons across all IDPs using the same permutation-testing procedure as described above.

In particular, this approach is of interest to test whether brain regions showing significant longitudinal changes demonstrate initial differences between the two groups that exist before the infection.

**Cognition.** We finally assessed whether the two groups differed at baseline in their cognition, based on the results from the ten variables from six different cognitive tests preselected above, correcting for multiple comparisons across cognitive variables.

**Lateralised effects.** As a post hoc analysis, we examined whether the longitudinal effects observed in grey matter thickness were lateralised, by subtracting the right ΔIDP from the corresponding left ΔIDP, for (1) all ΔIDPs of grey matter thickness showing significance in the main case-control analyses (across the hypothesis-driven and exploratory approaches), within the group with SARS-CoV-2 only (to avoid circularity); (2) all ΔIDPs of grey matter thickness across the entire cortex (151 pairs of left–right matched IDPs), and testing for associations between the left–right difference and the case-versus-control age modulated regressor. Results were corrected for multiple comparisons using FDR and FWE.

**Effect of time of SARS-CoV-2 infection.** For 351 SARS-CoV-2-positive participants who had an available date of infection (thus, in effect, excluding those identified through antibody lateral flow tests), we further looked post hoc at the possible effect of time interval between infection and second brain scan (acquired post-infection) on the significant IDPs from our hypothesis-driven approach to evaluate whether a longer interval might mean either a reduced loss of grey matter through, for example, potential progressive recovery of sensory inputs (olfaction), or a greater loss as a function of a longer, ongoing degenerative process.

**The effect of non-imaging factors.** We ran an additional analysis to test whether any non-imaging variables measured before SARS-CoV-2 infection might explain post hoc the longitudinal effects observed

in our significant IDPs. We considered non-imaging variables with at least 50% non-missing data in the participants ($n = 6,301$). We included individually each of these variables as an additional confounder for a repeat of the original Model 1 regression tests for those IDPs that were found to show significant longitudinal differences between the two groups, for both hypothesis-driven and exploratory approaches. If the strength of the original association was reduced by more than 25%, based on the regression $Z$-statistics, we considered a non-imaging variable to potentially explain the IDP–infection association. Further details are provided in Supplementary Analysis 7.

## Reporting summary

Further information on research design is available in the Nature Research Reporting Summary linked to this paper.

## Data availability

All source data are available on application for data access from UK Biobank.

## Code availability

Analysis code used in this study is available online (https://www.fmrib.ox.ac.uk/ukbiobank/covid/ and https://doi.org/10.5281/zenodo.5903258).

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

**Acknowledgements** We thank UK Biobank for making the data available, and all of the UK Biobank study participants, who generously donated their time to make this resource possible; B. Fischl and D. Greve for guidance with the FreeSurfer analyses. Analysis was carried out at the Oxford Biomedical Research Computing (BMRC) facility. BMRC is a joint development between the Wellcome Centre for Human Genetics and the Big Data Institute, supported by Health Data Research UK and the NIHR Oxford Biomedical Research Centre. This work was primarily supported by a Wellcome Trust Collaborative Award (215573/Z/19/Z). K.L.M. was supported by a Wellcome Trust Senior Research Fellowship (202788/Z/16/Z). The Wellcome Centre for Integrative Neuroimaging (WIN FMRIB) is supported by centre funding from the Wellcome Trust (203139/Z/16/Z). S.L. was supported by the Rina M. Bidin Foundation Fellowship in Research of Brain Treatment and the Pacific Parkinson's Research Institute. P.K. was supported by the UK Research and Innovation (MR/S034978/1). A.M.W. is supported by the NIH through ZIA-MH002781 and ZIA-MH002782. P.M.M. acknowledges personal and research support from the Edmond J. Safra Foundation and L. Safra, an NIHR Senior Investigator Award, the UK Dementia Research Institute and the NIHR Biomedical Research Centre at Imperial College London. This research has been conducted in part using the UK Biobank Resource under application number 8107.

**Author contributions** G.D., S.L., F.A.-A., C.A., C.W., P.M., F.L., J.L.R.A., L.G., E.D., S.J., K.L.M. and S.M.S. created, extracted and organised the imaging and clinical data. S.M.S. carried out the imaging analyses. B.T., A.M.W. and T.E.N. co-supervised the statistical analyses. R.C., P.M.M., N.A., K.L.M. and S.M.S. contributed to the creation of the UK Biobank COVID-19 re-imaging project. G.D., P.K., K.L.M. and S.M.S. conceived the brain imaging study. G.D. interpreted the results. G.D. and S.M.S. wrote the paper. All of the authors revised the paper.

**Competing interests** R.C. has been seconded from the University of Oxford as chief executive and principal investigator of UK Biobank, which is a charitable company. N.A. is chief scientist for UK Biobank. P.M.M. acknowledges consultancy fees from Novartis and Biogen; he has received recent honoraria or speakers' honoraria and research or educational funds from Novartis, Bristol Myers Squibb and Biogen. P.M.M. serves as the honorary chair of the UK Biobank Imaging Working Group and as an unpaid member of the UK Biobank Steering Committee; he is chair of the UKRI Medical Research Council Neurosciences and Mental Health Board.

**Additional information**
**Correspondence and requests for materials** should be addressed to Gwenaëlle Douaud.

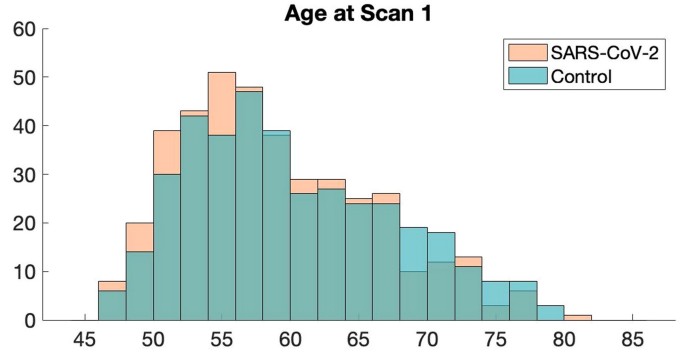

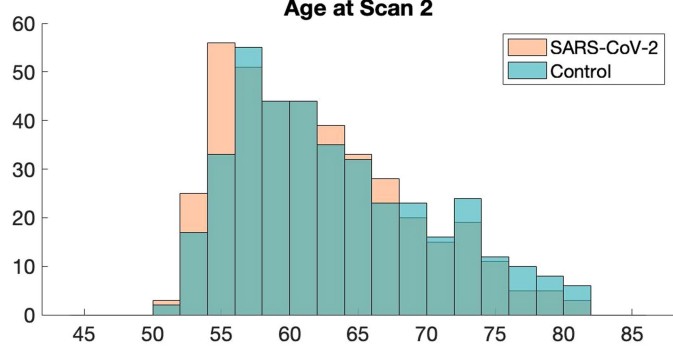

**Extended Data Fig. 1 | Age distributions for SARS-CoV-2 positive participants and controls at each time point do not differ significantly.**
Two-sample Kolmogorov-Smirnov was used to compute the P values for age comparisons, since age for each group was not normally distributed (Lilliefors P = 1e-03 for each group, and both age at Scan 1 or Scan 2). This showed no significant difference in age distribution between SARS-CoV-2 participants and controls at Scan 1: P = 0.15 or at Scan 2: P = 0.08.

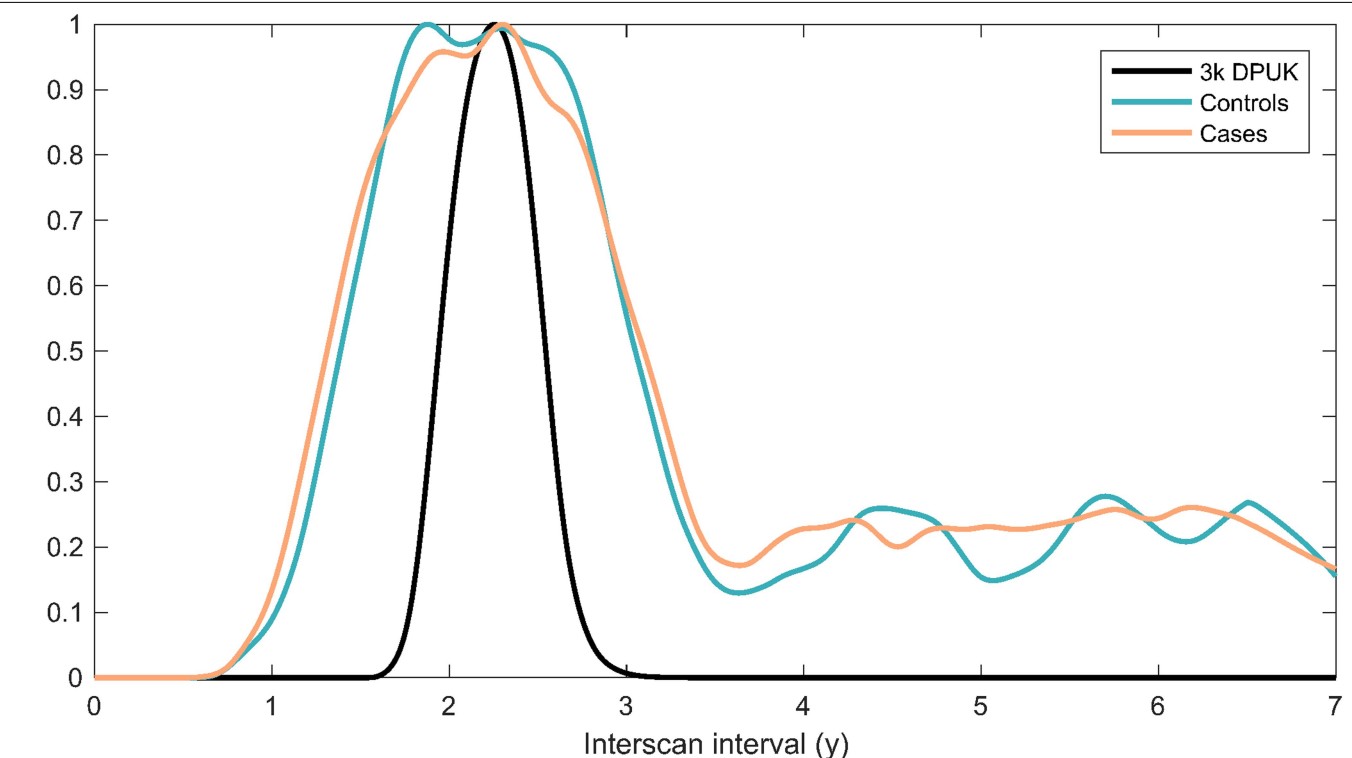

**Extended Data Fig. 2 | Histograms showing the well-matched distributions of Scan 1 - Scan 2 intervals for case and control groups.** The below IDP reproducibility Extended Data Fig. 3 shows, for comparison against the cases and controls, reproducibility from around 3,000 (2,943) UK Biobank participants who had returned for a second scan prior to the pandemic; hence we also show here the interscan intervals for this "3k" group, with tighter control over this interval (we have normalised each of those 3 groups to have a peak of 1, to make the relative comparison easier).

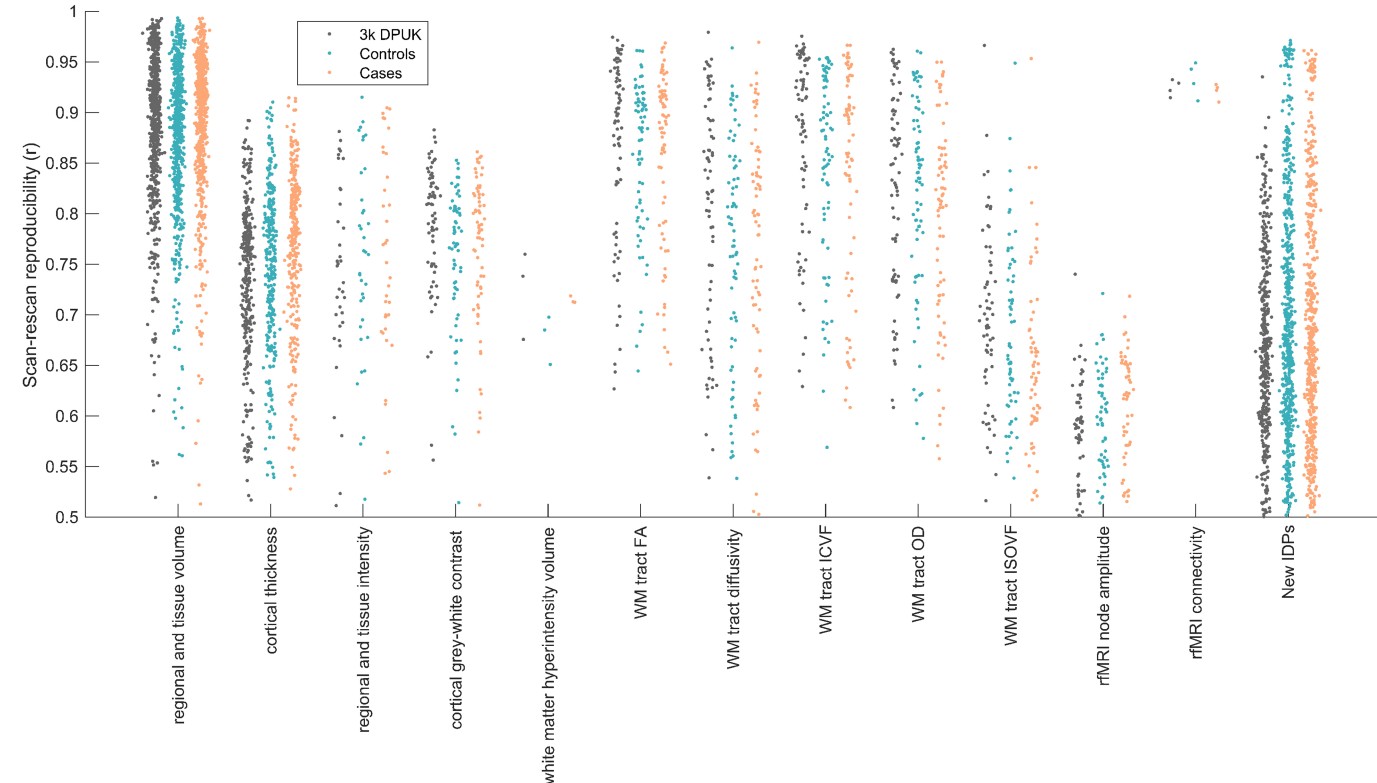

**Extended Data Fig. 3 | Scan-rescan reproducibility for all 2,047 IDPs used in the main modelling.** Each dot represents a single IDP, arranged into different classes of IDPs. For each IDP, the vector of values for each subject (i.e., 785x1 vector) from the first scan was correlated with the equivalent vector of IDP values from the second scan. The y axis shows the resulting correlation coefficient. These calculations are made separately for the pre-pandemic scan-rescan datasets ("3k DPUK"), and for cases and controls, demonstrating highly similar distributions within each IDP class for all 3 subject groups.

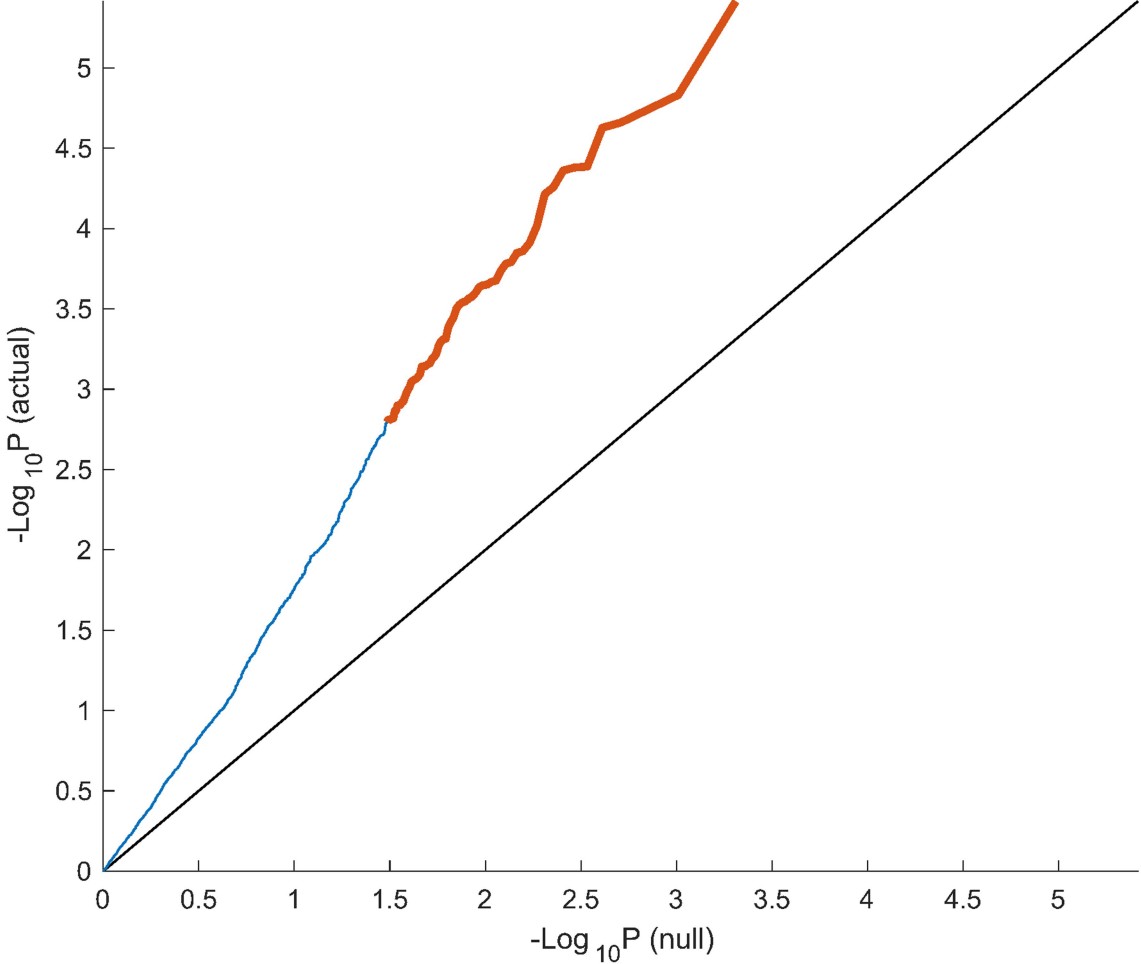

**Extended Data Fig. 4 | QQ plot for −log$_{10}$[$P_{uncorr}$] against the theoretical null distribution.** The black line at y=x shows the expected plot if no effects were present in the data. Orange points reflect ΔIDPs where the case-control effect passes FDR significance, and blue reflects those that do not.

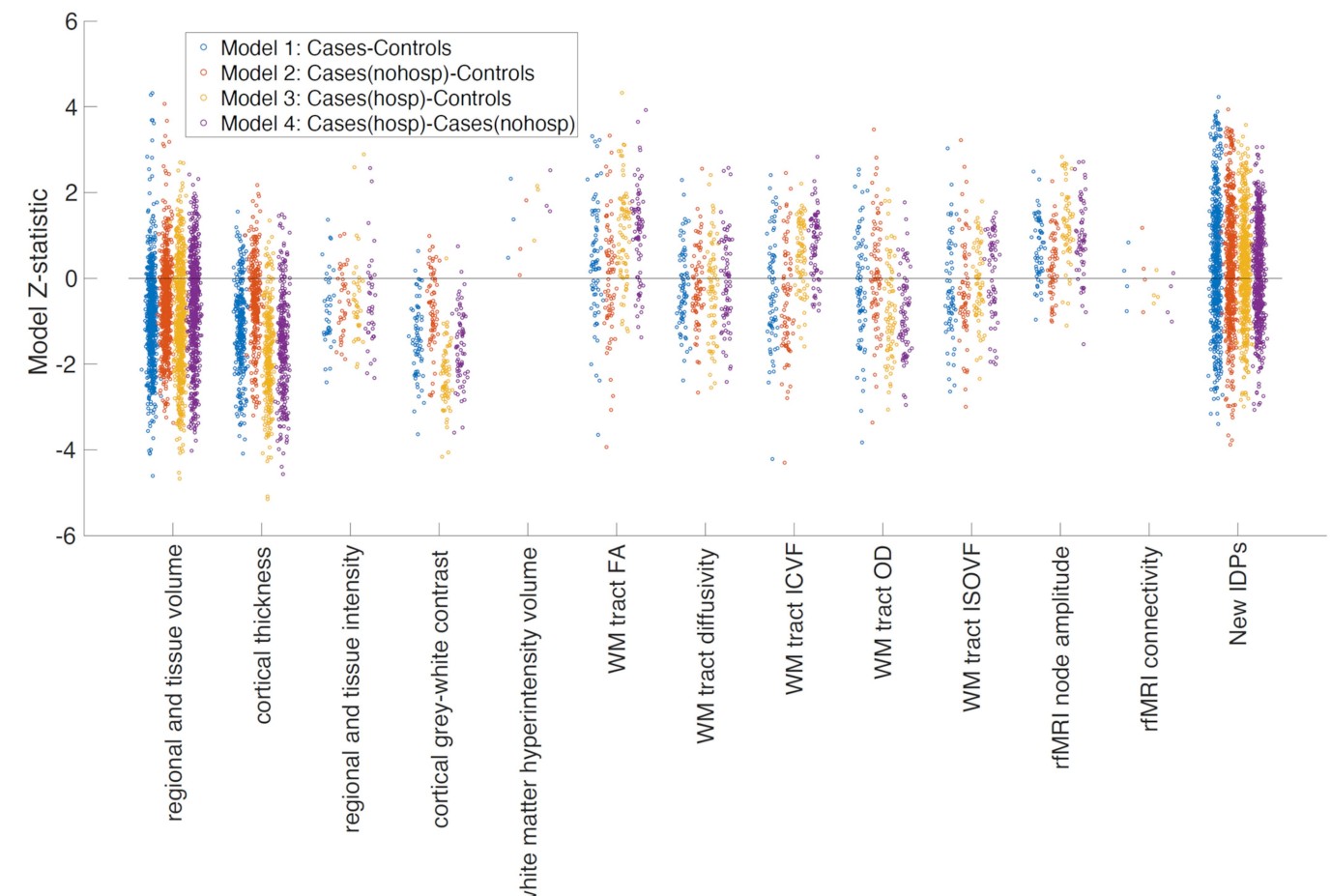

**Extended Data Fig. 5 | Model *Z*-statistics (one point per IDP, arranged in IDP classes) for the 4 main models.** Note that these are model Z-statistics, not raw effect size. Some IDP classes (e.g., cortical thickness and grey-white intensity contrast) show consistent group-difference effect directions across most IDPs (i.e., different brain regions), and all 4 models.

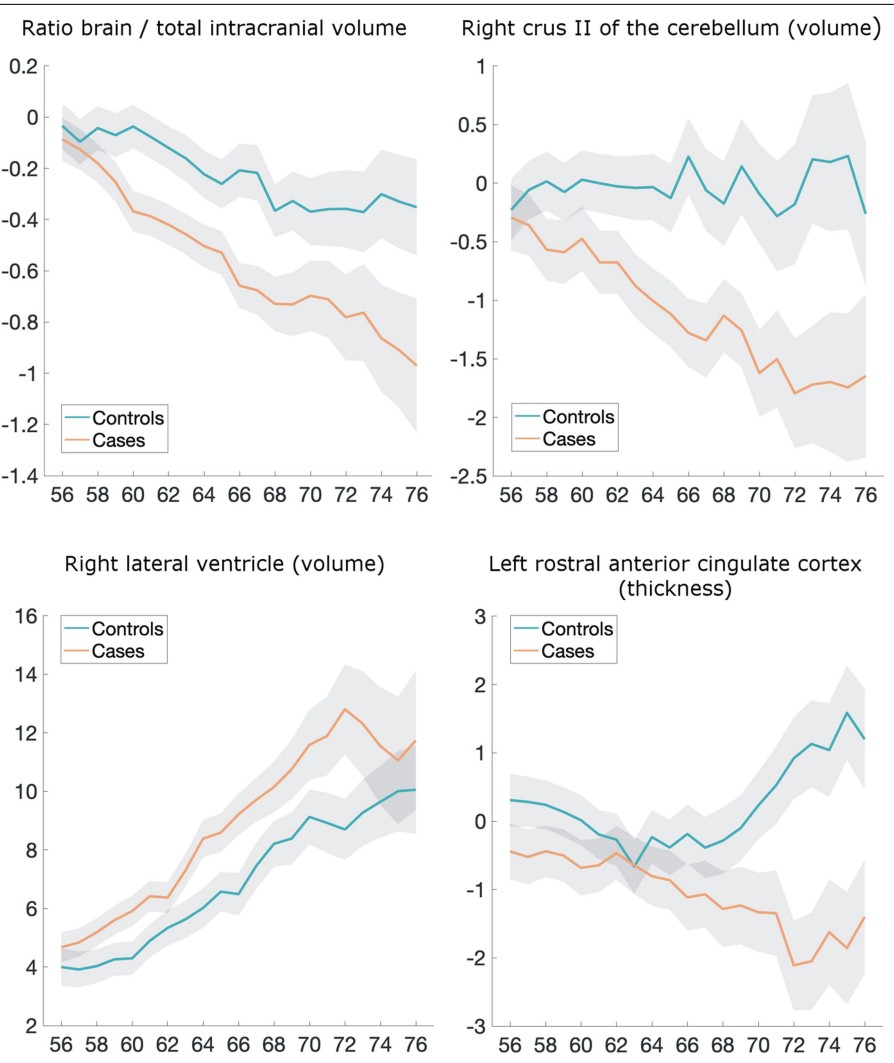

**Extended Data Fig. 6 | Examples of percentage change in some of the most significant longitudinal group comparison results from the exploratory approach.** Four amongst the top IDPs consistently showing longitudinal differences between SARS-CoV-2 cases and controls. All demonstrate either a greater reduction in local or global brain thickness and volume, or an increase in CSF volume. For each four IDP are the percentage changes with age for the two groups, obtained by normalising ΔIDP using as baseline the values for the corresponding IDPs across the 785 scans (created using a 10-year sliding window across cases and controls, with standard errors in grey). The counterintuitive increase in thickness in the rostral anterior cingulate cortex in older controls has been previously consistently reported in studies of ageing, together with that of the orbitofrontal cortex[57,58].

# Reporting Summary

## Statistics

For all statistical analyses, confirm that the following items are present in the figure legend, table legend, main text, or Methods section.

| n/a | Confirmed | |
|---|---|---|
| ☐ | ☒ | The exact sample size (*n*) for each experimental group/condition, given as a discrete number and unit of measurement |
| ☐ | ☒ | A statement on whether measurements were taken from distinct samples or whether the same sample was measured repeatedly |
| ☐ | ☒ | The statistical test(s) used AND whether they are one- or two-sided<br>*Only common tests should be described solely by name; describe more complex techniques in the Methods section.* |
| ☐ | ☒ | A description of all covariates tested |
| ☐ | ☒ | A description of any assumptions or corrections, such as tests of normality and adjustment for multiple comparisons |
| ☐ | ☒ | A full description of the statistical parameters including central tendency (e.g. means) or other basic estimates (e.g. regression coefficient) AND variation (e.g. standard deviation) or associated estimates of uncertainty (e.g. confidence intervals) |
| ☐ | ☒ | For null hypothesis testing, the test statistic (e.g. *F*, *t*, *r*) with confidence intervals, effect sizes, degrees of freedom and *P* value noted<br>*Give P values as exact values whenever suitable.* |
| ☒ | ☐ | For Bayesian analysis, information on the choice of priors and Markov chain Monte Carlo settings |
| ☒ | ☐ | For hierarchical and complex designs, identification of the appropriate level for tests and full reporting of outcomes |
| ☐ | ☒ | Estimates of effect sizes (e.g. Cohen's *d*, Pearson's *r*), indicating how they were calculated |

*Our web collection on statistics for biologists contains articles on many of the points above.*

## Software and code

Policy information about availability of computer code

| Data collection | Data were obtained from UK Biobank (available from UK Biobank upon data access application) |
|---|---|
| Data analysis | Data were analysed using the latest versions of FSL and FreeSurfer, and using MATLAB code supplied. |

For manuscripts utilizing custom algorithms or software that are central to the research but not yet described in published literature, software must be made available to editors and reviewers. We strongly encourage code deposition in a community repository (e.g. GitHub). See the Nature Portfolio guidelines for submitting code & software for further information.

## Data

Policy information about availability of data

All manuscripts must include a data availability statement. This statement should provide the following information, where applicable:
- Accession codes, unique identifiers, or web links for publicly available datasets
- A description of any restrictions on data availability
- For clinical datasets or third party data, please ensure that the statement adheres to our policy

All source data is available (upon data access application) from UK Biobank. See https://www.fmrib.ox.ac.uk/ukbiobank/covid/ for analysis code from this study.

March 2021

# Field-specific reporting

Please select the one below that is the best fit for your research. If you are not sure, read the appropriate sections before making your selection.

☒ Life sciences ☐ Behavioural & social sciences ☐ Ecological, evolutionary & environmental sciences

For a reference copy of the document with all sections, see nature.com/documents/nr-reporting-summary-flat.pdf

# Life sciences study design

All studies must disclose on these points even when the disclosure is negative.

| | |
|---|---|
| Sample size | Sample size was determined by data availability from UK Biobank. Exact samples sizes are listed in Methods. |
| Data exclusions | Extreme outlier values were removed on the basis of being more extreme than 8 times the median absolute deviation from the median for a given variable of interest (see Methods for full details). |
| Replication | N/A |
| Randomization | N/A |
| Blinding | N/A |

# Reporting for specific materials, systems and methods

We require information from authors about some types of materials, experimental systems and methods used in many studies. Here, indicate whether each material, system or method listed is relevant to your study. If you are not sure if a list item applies to your research, read the appropriate section before selecting a response.

## Materials & experimental systems

| n/a | Involved in the study |
|---|---|
| ☒ | ☐ Antibodies |
| ☒ | ☐ Eukaryotic cell lines |
| ☒ | ☐ Palaeontology and archaeology |
| ☒ | ☐ Animals and other organisms |
| ☐ | ☒ Human research participants |
| ☐ | ☒ Clinical data |
| ☒ | ☐ Dual use research of concern |

## Methods

| n/a | Involved in the study |
|---|---|
| ☒ | ☐ ChIP-seq |
| ☒ | ☐ Flow cytometry |
| ☐ | ☒ MRI-based neuroimaging |

## Human research participants

Policy information about studies involving human research participants

| | |
|---|---|
| Population characteristics | SARS-CoV-2 status based on primary care/hospital/public health records or two positive antibody lateral flow test positive results. |
| Recruitment | This study is part of the UK Biobank COVID-19 re-imaging project, which has imaging pre-pandemic for thousands of participants, and focused on inviting back for a second scan participants who had been infected with SARS-CoV-2, and matched controls (in terms of age, interval between scans, sex, ethnicity). Biases include those known for the UK Biobank, i.e. a generally wealthier, healthier, and less ethnically diverse population. Specific bias for the COVID-19 re-imaging project is that, because it is based on volunteering of previous UK Biobank participants, those who have been infected by SARS-CoV-2 tended to have milder symptoms (which can be seen as a strength rather than a weakness of this study). |
| Ethics oversight | Human subjects: UK Biobank has approval from the North West Multi-centre Research Ethics Committee (MREC) to obtain and disseminate data and samples from the participants (http://www. ukbiobank.ac.uk/ethics/), and these ethical regulations cover the work in this study. Written informed consent was obtained from all participants. A statement on this is included in the paper. |

Note that full information on the approval of the study protocol must also be provided in the manuscript.

# Clinical data

Policy information about clinical studies

All manuscripts should comply with the ICMJE guidelines for publication of clinical research and a completed CONSORT checklist must be included with all submissions.

| | |
|---|---|
| Clinical trial registration | *Provide the trial registration number from ClinicalTrials.gov or an equivalent agency.* |
| Study protocol | *Note where the full trial protocol can be accessed OR if not available, explain why.* |
| Data collection | *Describe the settings and locales of data collection, noting the time periods of recruitment and data collection.* |
| Outcomes | *Describe how you pre-defined primary and secondary outcome measures and how you assessed these measures.* |

# Magnetic resonance imaging

## Experimental design

| | |
|---|---|
| Design type | UK Biobank brain imaging data resting-state functional scans |
| Design specifications | N/A |
| Behavioral performance measures | N/A |

## Acquisition

| | |
|---|---|
| Imaging type(s) | UK Biobank brain imaging data: structural (T1, T2 fluid attenuation inversion recovery and susceptibility-weighted), diffusion, and resting-state functional scans. See Methods for full details. |
| Field strength | 3T |
| Sequence & imaging parameters | Please see Miller et al., Nature Neuroscience 2016 for a full list of the imaging parameters. |
| Area of acquisition | Whole brain |

Diffusion MRI     ☒ Used     ☐ Not used

| | |
|---|---|
| Parameters | Diffusion data are acquired with two b-values (b = 1,000 and 2,000 s/mm2) at 2-mm spatial resolution, with multiband acceleration factor of 3 (three slices are acquired simultaneously instead of just one). For each diffusion-weighted shell, 50 distinct diffusion-encoding directions were acquired (covering 100 distinct directions over the two b-values). (No cardiac gating.) |

## Preprocessing

| | |
|---|---|
| Preprocessing software | FSL 5.0.9, Freesurfer 6 and Freesurfer 7 for subcortical segmentations. |
| Normalization | Whenever applicable spatial normalisation was required (please see Miller et al., Nature Neuroscience 2016, and Alfaro-Almagro et al., Neuroimage 2018), non-linear registration was used based on the structural images (usually T1). |
| Normalization template | MNI152 and UK Biobank. |
| Noise and artifact removal | Please see full details in Miller et al., Nature Neuroscience 2016 |
| Volume censoring | N/A |

## Statistical modeling & inference

| | |
|---|---|
| Model type and settings | Regression/correlation (see Methods for full details). |
| Effect(s) tested | Effects of SARS-CoV-2 modulated by age (model based on a published meta-regression of 28 studies) - see Methods for details. |

Specify type of analysis:     ☐ Whole brain     ☐ ROI-based     ☒ Both

| | |
|---|---|
| Anatomical location(s) | ROI ("imaging-derived phenotypes", IDPs) cover the entire brain, and the entire cortical surface for FreeSurfer generated ROIs. In addition, we have created some visualisations of the effects voxel-wise and vertex-wise. |
| Statistic type for inference (See Eklund et al. 2016) | N/A |

| Correction | All results are (at least) false-discovery rate (FDR) significant, and we also report their family-wise error (FWE) corrected p-values. |
|---|---|

## Models & analysis

| n/a | Involved in the study |
|---|---|
| ☐ | ☒ Functional and/or effective connectivity |
| ☐ | ☐ Graph analysis |
| ☐ | ☐ Multivariate modeling or predictive analysis |

| Functional and/or effective connectivity | Based on partial correlation |
|---|---|

| Graph analysis | *Report the dependent variable and connectivity measure, specifying weighted graph or binarized graph, subject- or group-level, and the global and/or node summaries used (e.g. clustering coefficient, efficiency, etc.).* |
|---|---|

| Multivariate modeling and predictive analysis | *Specify independent variables, features extraction and dimension reduction, model, training and evaluation metrics.* |
|---|---|

