## [Peer Review File · Nature]

Manuscript Title: SARS-CoV-2 is associated with changes in brain structure in UK Biobank

Reviewer Comments & Author Rebuttals

Reviewer Reports on the Initial Version:

Referees' comments:

Referee #1 (Remarks to the Author):

This is a unique investigation into the effects of being infected by the COVID-19 virus on quantitative MRI metrics of brain structure and function. Given the known significant neuropsychiatric signs, symptoms, and sequelae of the COVID-19 pandemic on the human population, this re-imaging study takes strategic advantage of the robust infrastructure in place for the on-going collection and analysis of large-scale neuroimaging data from a population sample. Here including 785 participants, 401 positive cases and 384 controls. The rigorous methods for primary data acquisition, pre-processing, and extraction of the IDPs follow best practice standards, many of which were established by members of the study team working in collaboration with leaders across the field. The statistical analyses use appropriate methods to control for multiple comparisons. The exploration of the results and discussion are thoughtful, well supported, and clear. The one exception is the decision to include an age-modulation factor in the group comparison model and choice of that modulation factor, which is not well justified or properly explored. All analysis code is shared with the scientific community.

The focus of the study on a priori determined IDPs related to the functional neuroanatomy of the olfactory system given the prevalence of COVID-19 associated gustatory and olfactory impairments is appropriate and provides the opportunity for novel and important technical advances in generation of a focused set of relevant IDPs for hypothesis testing. The functional anatomical interpretation of the results is well supported, with plausible explanation for a lack of findings in the olfactory bulbs and piriform cortex- notoriously challenging regions for MRI. It is noteworthy that the authors did not provide an explicit analysis of the effect of presence/absence of hyposmia/hypogeusia on the imaging results and so state in the limitations section. Given the extensive documentation of medical records available for this cohort, it is not clear why that information was not available to be included in the study. It would greatly strengthen the results if such an investigation could be included.

The use of subset analyses that investigated the differences between the mild/moderate cases and those with more serious illness resulting in hospitalization provide supporting evidence that the results were not driven by these more extreme cases. The significant cognitive changes (e.g. increased time to complete Trails B) and the association of that change with a significant change in the volume of the crus II of the cerebellum is interesting and supportive of the results, but, even though only found in the cases, is not necessarily specific to the effects of COVID-19 infection for reasons noted below. Figures, tables, and supplementary materials all clearly convey the findings.

The magnitude of the reported significant longitudinal effects on brain indices and cognition associated with the infection are somewhat surprising – given the magnitude and extent of changes they find occur over a relatively short time-period in a cohort that predominately experienced relatively mild illness. Indeed, there are numerous neuroinflammatory processes that affect the brain and if even a subset of them had effect sizes like the results reported in this manuscript, nearly all people would experience similar changes to those ascribed to COVID-19. Interestingly, they report that for the n=351 cases where they had a reasonable estimate of the time interval between infection and second scan, none of the top 10 most significant results showed a significant effect of interval between infection and second scan. One interpretation of this finding, especially given the range of days between infection and Scan 2 (35-407 days with a mean of 141 and standard deviation of 79), is that the COVID-19 infection may not be the critical or only factor mediating the significant findings.

Which leads to the first of three major concerns with the study, the methods don't fully control for potentially significant differences between the cases and controls. The authors did case:control matching based on sex, ethnicity, date of birth, location of first imaging assessment clinic, and date of first imaging assessment. The cohorts are well matched on age at time of Scan 1, sex, ethnicity, years between Scan 1-Scan 2, SES as indexed by the Townsend deprivation index, and a limited number of very appropriate pre-COVID clinical assessments (systolic BP, diastolic BP, diabetes, weight, waist/hip ratio, BMI, alcohol intake, tobacco smoking). However, concerns remain and should be addressed, at a minimum by stating them clearly as a limitation in the interpretation of the results. Specifically, the number of markers of pre-COVID co-morbidity is quite limited and may not be sufficient to properly balance cases and controls. The additional confounder analyses of the 1,734 variables to examine the potential influence of baseline non-imaging factors, including many very appropriate indices of neuropsychiatric symptoms, is useful. However, the extremely large number of variables explored imposes a heavy (stringent) multiple comparison burden, thus might overlook sub-threshold associations that are real. Further, it has been repeatedly shown that the presentation, symptoms, and sequelae of COVID-19 infections are diverse and highly variable across individuals. Thus, in this cohort of cases, there could have been a range of different variables that reflect a common underlying disorder such as mild cognitive impairment that wouldn't rise to the level of significant when assessed individually. The authors should consider applying an approach that would cluster such variables to see if they jointly explain the case-control difference. This is very important because case control mismatch could yield significant type 1 errors.

The very careful attention to demonstrate no baseline differences in the IDPs and cognitive testing between cases and controls is appropriate and necessary, but not sufficient to establish adequate case:control matching. What is potentially obscured in the study is the possibility that the cases have multifactorial reasons for a greater propensity to become infected with COVID-19 (e.g. insidious onset of mild cognitive impairment that is not yet evidenced in the clinical records, baseline imaging or by cognitive testing) that could also result in the greater rate of generalized brain atrophy.

One potential way to deal with the question of the specificity of the findings is to examine an additional matched control cohort who have been carefully diagnosed with non-COVID-19 pneumonia.

The second major concern with the study is that the authors don't provide adequate explanation for what motivated their use and choice of an "age at time of COVID-19 infection modulation factor" for the case control binary variable. Why is this necessary to include? Please verify what the parameter estimates reflect. It appears to be a weighted average, where older people have higher weights. The authors should first report the results from analyses when this exponential term is not included in the model. If the results differ, how do they understand the results? How did they determine that this is a reasonable correction factor to use? What happens if a different correction factor is chosen? They are missing a sensitivity or robustness estimate for their modeling choice. The modulation factor is taken from a meta-regression analysis that studied fatality rates. Clearly not an issue here where all the participants are living, and most had mild COVID-19 infections.

The final major concern is that the conclusions overstate the significance of the results and pay little attention to the serious limitations. The basic claim made (e.g. that the changes exist in those who have been infected with the virus and the only question remaining is whether they persist) is misleading. Rather, this study may have found quantitative brain MRI evidence for pathophysiological changes in some people due to infection with COVID-19 in the brain, but that remains to be confirmed with further studies. The most critical limitations arise directly from the challenges of proving that the cases and controls are properly matched when doing a study with this design. But they also overstate the findings in their conclusions, even when properly reporting in the results the percentages of infected participants that showed a greater longitudinal change than the median value in the control ranged from 56-62% of the cohort. It is interesting to note that when drawing attention to the benefits of their longitudinal design over cross-sectional group comparisons, they clearly articulate the difficulties in interpreting cross-sectional post-infection differences (bottom of Page 18-top of Page 19). The same should be done for their main results from the longitudinal analyses.

The authors note a limitation of the study is the potential misclassification [e.g. COVID positive cases not all positive due to false positives in testing and COVID negative cases actually positive due to absence of confirmed negative status and or false negative tests]; however, that should bias toward the null hypothesis of no case:control difference.

The criteria used to remove IDPs were all appropriate, but there was minimal information provided about the removed IDPs. Please report the number of removed IDPs separately for the cases and controls. Also report metrics about which IDPs were removed by image data type and any patterns in the missing and outlier data. For example, it appears that many of the fMRI derived IDPs were removed, why is that?

The authors' state that the counter intuitive increases in cortical thickness in the L rostral anterior cingulate cortex and orbital frontal cortex in the control cases that is not seen in the cases is reported "consistently" in the literature, but only cite a single reference that first reported this. What further evidence supports that this is a consistent finding?

Minor points:

- 1) Please clarify why standard scaling by $\sqrt{N-3}$ is used instead of $\sqrt{N-3-\text{number of confounds}}$ when the correlations were Fisher-transformed to Z-statistics.
- 2) The final sentences at the bottom of Page 23 seem to have editing errors (e.g. “finding” should be “findings” and “... correlated post hoc with the [longitudinal changes in the] cognitive...”). Please clarify.
- 3) Pages 3, 4 and 18: have a reference improperly formatted as a DOI citation rather than a numeric value.
- 4) References 41 and 56 are the same.
- 5) Figure S2. Y axis is improperly labeled.

Referee #2 (Remarks to the Author):

With interest, I read the manuscript “Brain imaging before and after COVID-19 in UK Biobank by Douaud et al. However, there is a major principal and basic concern that not only substantially limits enthusiasm for the study, but also precludes a recommendation to publish these data:

Why is that: there is a basic problem with the way neuropsychology was evaluated, specifically, which tests were used, and what conclusions were drawn on the basis of the used tests. In short: reduced performance in an executive task, when the mnemonic system (parahippocampal gyrus etc.) is supposed to be affected exemplifies that things don't fit together.

Major concerns, in detail:

The manuscript reports many changes in mild-to moderate post-C19 patients, which should have their clinical correlation. This cannot, however, be drawn from the presented data. While I am having difficulties in interpreting the tables (which indicates that they are not well preprocessed), a z-score of 4 as an indicator for describing a difference appears to be high. Comparing Post-C19 to non-Post-C19 patients would be extremely helpful to make this z-score differences understandable, namely to grasp what this difference actually means.

Most importantly, the respective “references” of brain imaging, namely the neuropsychological tests, are highly problematic: in addition to the „classical” clinical COVID-19 symptoms, all investigated C19 patients are expected to have suffered for many months from symptoms, which are assigned to brain regions, where memory (parahippocampus), olfaction (olfactory cortex), or behaviour (amygdala) are located. The respective neuropsychological tests, which were used in this study, mostly probe executive-constructive functions. Two of the tests used in the study (insufficiently) assess memory functions. However, these two tests do not provide substantial differences in memory function (the delayed recall test would be adequate and should show differences – which it doesn't). On top, the assessment of the Trail-making test is incorrect: not the seconds need to be counted in absolute numbers (to execute the test), but rather whether the individual remains within a given time frame (in relative terms): pathology is given only if the subject does not adhere to the defined time frame.

Minor:

What also in at least surprising: 10 out of 11 changes were assigned to the left hemisphere: why is it, and how can be excluded that this is not rather a sign for an erroneous methodology?

In conclusion, I am not doubting that the reported differences/numbers are true. However, I am strongly fearing that these differences and values are not resulting from or reflecting the reasons, which the authors put forward. The biological and truly functional relevance, based on my arguments above, namely that – for what is supposed to be measured - inappropriate tests were used, make me believe that there is a non-causal correlation of alterations measured and reasons given/interpretation made (and remind me of the classical example of the correlation of the stork population and birth rate). A possible explanation of the alterations measured may be indirect or “downstream” or collateral changes of alterations upon virus entry through cranial nerves (reaching the olfactory bulb). However, in total, i.e. if I put all the tests together, namely what they can deliver qualitatively, as well the given interpretation, I am sorry to say that there is not much left that speaks in favor to follow the line of reasoning and claims of the authors.

Referee #3 (Remarks to the Author):

this is a very well written manuscript from a highly knowledgeable team. The authors report a multimodal exploration of the longitudinal effects in the brain associated with SARS-CoV-2 infection, made possible by leveraging the unique resource of the UK Biobank project, and its COVID-19 re-imaging sub-study.

The UK Biobank repository is a vital, large-scale, neuroimaging resource for studies such as this and the authors leverage this in a timely manner to address what many have feared - that COVID-19 is not merely just a disease which affects the respiratory system but one that, even in recovery, may have extensive impacts on the brain.

In this study, N=785 participants had two usable brain scan sessions: N=401 SARS-CoV-2 positive participants(cases), and N=384 controls who were matched for age, sex, ethnic background and interval between the two scans, as well as various risk factors. Hypothesis-driven and exploratory longitudinal analyses conducted by the authors revealed significant, neurological impacts associated with SARS-CoV-2: affecting mainly in the olfactory cortical system, as evident from diffusion measures which served as proxies for tissue damage; in regions functionally connected with the piriform cortex, olfactory tubercle, and anterior olfactory nucleus; as well as a more prominent reduction of grey matter thickness and contrast in the infected participants in the left parahippocampal gyrus and lateral orbitofrontal cortex.

What is more, while the greater atrophy for the SARS-CoV-2 positive participants was reported to be localized to a few, mainly limbic, regions, an increase in cerebrospinal fluid volume and decrease of whole brain volume suggests a diffuse loss of grey matter in addition to the regional effects observed in the olfactory-related areas.

Additional examination demonstrated that effects associated with infection, both in the cortical

thickness and mean diffusivity, extended mainly to the anterior cingulate cortex, insula, supramarginal gyrus and the temporal pole. Comparing those patients (N=15) who had been hospitalized with COVID-19 against comparable, non-hospitalized cases demonstrated a more distributed pattern with greater reduction in grey matter thickness in fronto-parietal and temporal regions.

Finally, significant cognitive decline was noted, which persisted even after excluding the hospitalized patients, in the SARS-CoV-2 positive group between the two imaging timepoints. This decline was associated with greater atrophy of the crus II sub-region of the cerebellum.

Such a study is very timely and would be of much value to the scientific community and to the public. I would be in strong favor of its publication at the earliest possible time to do so. Given the ongoing debate as to the long term effects associated with COVID-19 survival - especially in unvaccinated individuals - this study lends further support for public health measures to inoculate people against SARS-CoV-2 and its variants.

Referee #4 (Remarks to the Author):

The study by Douaud et al used a large imaging dataset from UK Biobank to investigate the changes in the brain by comparing a long list of brain measures between two time points (roughly three years apart). Two groups of subjects were included: 401 subjects were infected by SARS-CoV-2 between the two time points while 384 were controls. Over 2,000 brain measures (called image-derived phenotypes in the paper) plus various behavioral (cognitive) and non-imaging measures were considered. The goal was to find out which measures had a different trajectory in terms of brain change in the patients compared to the controls. Largely based on the statistical results from a sophisticated model, they found changes in some measures that were associated with the olfactory system.

The far-reaching impact of the current pandemic is an important topic from various research perspectives. The current study could contribute to our understanding about the short-term effects in the brain. With relatively large sample sizes, this imaging dataset provides a unique opportunity to gain various insights about the neurological impact of the COVID-19 virus. For example, even without involving their main models, the information extracted from a few brain measures is revealing as shown in Figure 1, Figure 3 and Figure S6. In addition, the authors presented a large amount of modeling work, and made available the analysis code and the full list of analytical results.

However, there are many methodological issues with the paper, some of which are quite severe. For example, the assumed age trajectory adopted from an epidemiological study is unjustified for curve fitting. In addition, the evidence for the lack of baseline differences is not convincing. Because of these methodological issues, the paper tends to emphasize “more on the destination (research claims and metrics) than on the journey” (Calster et al, 2021. Methodology over metrics: current scientific standards are a disservice to patients and society. *Journal of Clinical Epidemiology*. <https://doi.org/10.1016/j.jclinepi.2021.05.018>). As the result interpretation and the discussion of this study currently lack a solid foundation, the review below will focus only on the modeling aspect.

1) An unjustified assumption was adopted regarding the trajectory of brain changes relative to age. The authors intended to explore the age trajectory of brain changes during the period of ~3 years for each of those thousands of variables, and compare whether the patients and the controls differed in terms of their age trajectories in brain changes. To be able to make such comparisons, the authors assumed an age trajectory of exponential relationship $10^{(\text{Age} \times 0.0524 - 3.27)}$ based on the age dependence of infection fatality rates estimated by Levin et al (2020; doi:10.1007/s10654-020-00698-1). For several reasons, this assumption is inappropriate.

(a) Lack of justification. In fMRI studies, an empirical curve for hemodynamic response is typically used to capture the BOLD response. In that case, the empirical curve at least is rigorously verified through many hemodynamic response experiments. Even though the larger death rate of COVID-19 patients with older people is roughly captured to some extent by the exponential relationship of $10^{(\text{Age} \times 0.0524 - 3.27)}$ as shown in Levin et al (2020), such an empirical curve based on observational data is embedded with some extent of uncertainty. More crucially, it is difficult to see how the death rate among patients could be applied and related to the age trajectory of brain measures among healthy subjects. Even for those patients, it remains questionable as to why those brain measures would follow the same pattern as the death rate.

(b) No accommodation for the heterogeneity of age trajectory across measures. Even if such a trajectory of death rate based on epidemiological data could be a reasonable approximation for some brain measures, it remains problematic for other measures since there is likely a large amount of variation across those thousands of measures as shown in the limited few examples in Figures 1, 3, and S6. Some of the patterns in those three figures do not seem to match the assumed pattern of an exponential curve. Needless to say, it is difficult to assess the sensitivity of these models if the empirical relationship is not accurate enough.

(c) Poor interpretability of the analytical results. Under the current model structure, each model is meant to match the trajectory of a brain measure change with the death rate pattern, and then to obtain the difference of the pattern matching between patients and controls. However, the reported correlation “r” values in all the tables are difficult to interpret. In the end, the results largely stop at the qualitative, not quantitative, level. For example, Table 4 shows the “top 10 significant results” with the correlation “r” column under Model 1 in the range of [-0.14, 0.16]. Are those “r” values considered high or low in terms of brain measure difference between the two groups? In other words, it does not tell us the amount of decrease (or increase) for each measure (e.g., brain volume) between patients and controls. Also, the presentation of these results begs the question: why no “r” values were provided for models 2-4 in those tables? In addition, the Z-values in the tables do not reveal the amount of brain measure either other than its statistical significance. It would be better to provide quantities in the same physical scale in those tables as the y-axis in Figures 1, 3 and S6.

(d) Lack of model validation and model quality check. For example, some indices should be provided to assess the fitness of each model in general as well as for each brain measure. In addition, regardless of the modeling assumptions, the fitted patterns should be plotted against the raw data in, for example, Figures 1, 3 and S6 so that their quality can be visually verified.

2) The interpretation about the baseline differences (Tables S3 and S4) as well as those 7,545 non-imaging phenotypes (page 6) between patients and controls is problematic and unconvincing. The correction for multiple comparisons might be reasonable when one rejects the null hypothesis. However, it becomes very problematic when it is used for accepting the null hypothesis. It is well-known that the lack of strong statistical evidence does not mean a strong evidence for null effect. That is, one cannot make a strong statement when a null hypothesis is not rejected (especially when correction for multiple comparisons is applied). Some of the individual differences (e.g. the column of original p-value in Tables S3 and S4) do point to some extent of baseline differences. Furthermore, since the current curve-fitting in the models needs justifications, the authors should show/plot the original data at the baseline (similar to Figures 1, 3, and S6).

3) The analytical approach lacks consistency. Four slightly different models were adopted to compare the four groups (controls, patients, non-hospitalised patients, and hospitalised patients). Each of the four models were applied to two sets of brain measures: one hypothesis-driven (with about 300 measures) and the other exploratory (with above 2,000 measures). In addition to these eight analyses with the imaging data, some of the models were further applied to various behavioral measures, risk factors, correlation analyses, etc. It is difficult to track the total number of analyses performed in this study. Even though the correction for multiple comparisons was used within each analysis, no similar step was taken to make statistical adjustment across those tens of analyses. To avoid this pitfall, the authors could have pre-registered all the analytical details. On the other hand, a short list of analyses could be adopted to provide a much cleaner and more focused presentation.

4) Heavy reliance on statistical inference (e.g., lack of effect quantification, interpretation difficulty) can be perilous. With large sample sizes, small or even tiny effects could reach statistical significance. Similarly, even when an explanatory variable (e.g., using the trajectory of death rate for brain changes) is not well-justified in the context, results solely based on statistical significance can be misleading. In addition, there are some procedural inconsistencies and misinterpretations of the statistical results.

(a) With a model applied thousands of times to various measures, the issue of multiple comparisons is not the only major concern. Nevertheless, if one is concerned with the multiple comparisons issue, then corrections should be not limited to each analysis/model; instead, it should be equally applied to tens of different models/analyses.

(b) All the UKBank data were used for normalisation for those 785 subjects, but only those 785 subjects were used for normalisation in Figure 3.

(c) The specific correction adopted for multiple comparisons appears to be arbitrary and inconsistent. Sometimes both FWE and FDR are mentioned in the text while in all tables only FWE results are shown.

Below are some minor issues.

-- The word "network" in the paper should be clarified or replaced. A network is supposed to involve a number of brain regions. However, the usage in the paper is apparently different (e.g. Figure 1).

-- The usage of Z-statistics in the manuscript seems inconsistent/ambiguous. It is used to refer to the Fisher-transformed value (page 31) which is usually called Z-score, but the Z-statistics in other places of the text as well as in table 4 and Figure S7 is likely different.

-- It is misleading to label the study as longitudinal. The longitudinal aspect is only limited to the two time points (~3 years apart). In fact, this study remains largely cross-sectional because the age trajectory is basically captured with different subjects within the age range of 51-81.

-- For clarity, all the models should be explicitly expressed with symbols and subscripts representing various factors and as well as all the terms including intercept and confounding effects. Explain the detail and purpose of demeaning in model (1).

-- For better reference and communication during the reviewing process, add line numbers and page numbers.

-- Provide uncertainty information. In all tables, consider adding a column for standard error or confidence interval, which is much more informative and useful than those statistical numbers and various p-values.

-- In those tables for modeling results of group differences, provide the same information (effect magnitude and uncertainty) for each group so that the group differences can be assessed with proper references. In addition, the information at the baseline and the second scan should be added.

-- The criteria for outlier handling are arbitrary and inconsistent. Sometimes it was 8 times the median absolute deviation, while sometimes it was 15 times. What are the causes for those outliers? Also, can those missing data be considered as missing at random?

-- The residualisation step for Eq (2) is troubling. First, it may create some biases in effect estimation. Second, the factor "N-3" adopted in the Fisher transformation is not accurate because of the residualisation process.

-- Task fMRI data were mentioned in the paper, but what IDPs were obtained for such data?

-- Add a color bar in Figure 2 so that the color maps could be properly interpreted. What is the rationale for the threshold of 3 for Z-values?

-- Explain the post hoc correlation analysis for the cognitive data (bottom of page 33).

-- Axis labels, legend symbols and legend labels in some figures (for example, Figures 3 and S5) are too small.

Author Rebuttals to Initial Comments:

We thank the reviewers for their positive remarks and constructive suggestions, which we have addressed below. Modifications have been highlighted in the manuscript in yellow. In particular, we now: report effects as percentages relative to mean baseline (and SE on these); include supplementary analyses looking at the longitudinal effects associated with pneumonia; investigate further potential baseline differences between SARS-CoV-2 cases and controls in both brain imaging and non-imaging phenotypes (using for instance PCA on these phenotypes, and a PCA focussing on cognitive phenotypes); carry out an even more thorough and “aggressive” analysis of the effects that these baseline variables might have on our longitudinal results; and introduce an alternative, data-driven approach to selecting the scores used to assess cognition, based on out-of-sample participants who are the most likely to show cognitive impairment. We believe that these changes have significantly improved the paper and the clarity of its message.

Reviewer #1:

This is a unique investigation into the effects of being infected by the COVID-19 virus on quantitative MRI metrics of brain structure and function. Given the known significant neuropsychiatric signs, symptoms, and sequelae of the COVID-19 pandemic on the human population, this re-imaging study takes strategic advantage of the robust infrastructure in place for the on-going collection and analysis of large-scale neuroimaging data from a population sample. Here including 785 participants, 401 positive cases and 384 controls. The rigorous methods for primary data acquisition, pre-processing, and extraction of the IDPs follow best practice standards, many of which were established by members of the study team working in collaboration with leaders across the field. The statistical analyses use appropriate methods to control for multiple comparisons. The exploration of the results and discussion are thoughtful, well supported, and clear. The one exception is the decision to include an age-modulation factor in the group comparison model and choice of that modulation factor, which is not well justified or properly explored. All analysis code is shared with the scientific community

We first would like to thank Reviewer 1 for this very positive summary of our study.

- 1. The focus of the study on a priori determined IDPs related to the functional neuroanatomy of the olfactory system given the prevalence of COVID-19 associated gustatory and olfactory impairments is appropriate and provides the opportunity for novel and important technical advances in generation of a focused set of relevant IDPs for hypothesis testing. The functional anatomical interpretation of the results is well supported, with plausible explanation for a lack of findings in the olfactory bulbs and piriform cortex- notoriously challenging regions for MRI. It is noteworthy that the authors did not provide an explicit analysis of the effect of presence/absence of hyposmia/hypogeusia on the imaging results and so state in the limitations section. Given the extensive documentation of medical records available for this cohort, it is not clear why that*

information was not available to be included in the study. It would greatly strengthen the results if such an investigation could be included.

Regarding the lack of information on hyposmia/hypogeusia, we agree that it would have been desirable to have access to this information for our study, something we acknowledge at the top of our paragraph highlighting limitations. We rely however on what is collected and provided by UK Biobank, which **unfortunately does not contain this type of recent symptom information**. The only recent clinical information that we have access to are the clinical codes (diagnoses) available from primary care (general practitioners) or hospital records, and treatment procedures in a few cases. We have **now expanded on this further in the Discussion**:

Discussion: Limitations of this study include the (...) lack of clinical correlates as they are not currently available as part of UK Biobank's COVID-19-related links to health records (of particular relevance, potential hyposmic and hypogeusic symptoms and blood-based markers of inflammation)(...).

- 2. The use of subset analyses that investigated the differences between the mild/moderate cases and those with more serious illness resulting in hospitalization provide supporting evidence that the results were not driven by these more extreme cases. The significant cognitive changes (e.g. increased time to complete Trails B) and the association of that change with a significant change in the volume of the crus II of the cerebellum is interesting and supportive of the results, but, even though only found in the cases, is not necessarily specific to the effects of COVID-19 infection for reasons noted below. Figures, tables, and supplementary materials all clearly convey the findings. The magnitude of the reported significant longitudinal effects on brain indices and cognition associated with the infection are somewhat surprising – given the magnitude and extent of changes they find occur over a relatively short time-period in a cohort that predominately experienced relatively mild illness.*

We appreciate the requests from Reviewer 1 and Reviewer 4 for reporting the effect sizes in a more easily interpretable way, and fully agree. We have now **recomputed all our analyses using regression instead of correlation**. As noted in Methods in our original submission, this is theoretically equivalent when done correctly, and indeed gives virtually identical results (e.g., in terms of Z and P), but some readers may be more comfortable with regression analyses. This also allows us to report effects in more useful units. We have therefore **replaced correlation coefficients with % change group-difference effects**, showing therefore more clearly that the magnitude of these effects are modest, ranging from 0.2 to 2% for the vast majority of the significant longitudinal results. We also **give standard errors on these % change**, as requested below.

More specifically, for each IDP, the % group-difference effects are computed via a normalised version of the regression beta, where the normalisation uses the original IDP

value from the baseline scan averaged across the 785 subjects (and the regression model is scaled to have average peak-peak height of 1). Hence, for the main longitudinal results, the % effect is the average group difference of the longitudinal IDP change, expressed as a % of the mean baseline IDP value. For the single timepoint results, the % effect is the average group difference of the Scan 1 (baseline) IDP values, expressed as a % of the mean baseline IDP value. We have **now added % and SE information** instead of Pearson r **in all our Tables, detailed these new procedures in the Methods section, and mentioned this information in the Results section:**

Methods: The group-difference regressor is scaled to have average peak-peak height 1, so that the regression parameter from fitting g_m can easily be converted into a percentage change measure, when normalised by the mean baseline value for a given IDP. For the main longitudinal modelling, this represents the average group difference in the longitudinal change, and for the separate modelling of baseline IDPs only, this percentage reflects the average group difference in the baseline values. In addition to reporting % effects and associated standard errors, we also report the statistical significance as Z-statistics (Gaussianised regression model T-statistics), and P-values.

Hypothesis-driven results: For those significant IDPs, average percentage change differences between the two groups was moderate, ranging from ~0.2 to ~2%, with the largest differences seen in the volume of the parahippocampal gyrus and entorhinal cortex (**Table S1**).

Exploratory results: For those significant IDPs, average percentage change differences between the two groups was moderate, ranging from ~0.2 to ~2% (except for two diffusion measures in the fimbria at >6%, due to the very small size of these regions-of-interest), with the largest differences seen in the volume of the parahippocampal gyrus and caudal anterior cingulate cortex (**Table S1**).

3. *Indeed, there are numerous neuroinflammatory processes that affect the brain and if even a subset of them had effect sizes like the results reported in this manuscript, nearly all people would experience similar changes to those ascribed to COVID-19. Interestingly, they report that for the $n=351$ cases where they had a reasonable estimate of the time interval between infection and second scan, none of the top 10 most significant results showed a significant effect of interval between infection and second scan. One interpretation of this finding, especially given the range of days between infection and Scan 2 (35-407 days with a mean of 141 and standard deviation of 79), is that the COVID-19 infection may not be the critical or only factor mediating the significant findings.*

We believe that the most likely reason for this lack of statistically significant interval association is a lack of power for this test, compounded by the fact that we have only few cases at 6 months or more after infection (less than 20%). The effects seen here in the brain may of course also actually be static or not changing greatly (for the worse or the better) over this timescale post infection. We have **now expanded on this point in the Discussion:**

Discussion: There is for instance some very preliminary evidence, in a few previously hospitalised COVID-19 patients, that brain hypometabolism becomes less pronounced when followed-up 6 months later, even if it does not entirely resolve^{38,42}. In our much milder cohort, structural (as opposed to functional) changes might take longer and require larger numbers to be detected. When we tested whether time between infection and second brain scan had any relationship — positive, indicative of recovery, or negative, indicative of an ongoing degenerative process — with the grey matter loss or increase in diffusivity in the significant IDPs, we found no statistically significant effect. This result is also possibly owing to the relatively small range in duration of infection at the time of this study, between 1 and 13 months for those 351 infected participants for whom we had a diagnosis date, and particularly with less than 20% of these participants having been infected for over 6 months (**Figure S9**). Additional follow-up of this cohort, increasing the number of cases infected for 6 months or longer, would be particularly valuable in determining the longer-term effects of infection on these limbic structures.

Also, we have added a supplementary figure of the histogram for time between infection and second scan:

Figure S9: Histogram of the time between diagnosis and the second scan. Diagnosis date was available for 351 cases, more than 80% of whom had less than 6 months between infection and their second scan.

- 4 Which leads to the first of three major concerns with the study, the methods don't fully control for potentially significant differences between the cases and controls. The authors did case:control matching based on sex, ethnicity, date of birth, location of first imaging assessment clinic, and date of first imaging assessment. The cohorts are well matched on age at time of Scan 1, sex, ethnicity, years between Scan 1-Scan 2, SES as indexed by the Townsend deprivation index, and a limited number of very appropriate pre-COVID clinical assessments (systolic BP, diastolic BP, diabetes, weight, waist/hip ratio, BMI, alcohol intake, tobacco smoking). However, concerns remain and should be addressed, at a minimum by stating them clearly as a limitation in the interpretation of the results. Specifically, the number of markers of pre-COVID co-morbidity is quite limited and may not be sufficient to properly balance cases and controls. The additional confounder analyses of the 1,734 variables to examine the potential influence of baseline non-imaging factors, including many very appropriate indices of neuropsychiatric symptoms, is useful. However, the extremely large number of variables explored imposes a heavy (stringent) multiple comparison burden, thus

might overlook sub-threshold associations that are real. Further, it has been repeatedly shown that the presentation, symptoms, and sequelae of COVID-19 infections are diverse and highly variable across individuals. Thus, in this cohort of cases, there could have been a range of different variables that reflect a common underlying disorder such as mild cognitive impairment that wouldn't rise to the level of significant when assessed individually.

a. The authors should consider applying an approach that would cluster such variables to see if they jointly explain the case-control difference. This is very important because case control mismatch could yield significant type 1 errors.

b. The very careful attention to demonstrate no baseline differences in the IDPs and cognitive testing between cases and controls is appropriate and necessary, but not sufficient to establish adequate case:control matching. What is potentially obscured in the study is the possibility that the cases have multifactorial reasons for a greater propensity to become infected with COVID-19 (e.g. insidious onset of mild cognitive impairment that is not yet evidenced in the clinical records, baseline imaging or by cognitive testing) that could also result in the greater rate of generalized brain atrophy.

We appreciate these concerns, although we do feel that the subject matching in the study design by UK Biobank, and the additional confound variables utilised here, provide already a strong control –and possibly the most comprehensive to date– against many of the largest areas of possible bias. This holds particularly true when combined with the pre-vs-post longitudinal nature of this study, the feature which makes this a unique dataset for helping understand the effects of SARS-CoV-2 infection.

However, it is true that multiple comparison correction across thousands of baseline imaging-derived phenotypes (IDPs) or non-imaging-derived phenotypes (nIDPs) comes at the cost of loss of sensitivity, just as it does for the main longitudinal findings reported. We agree with the reviewers that pushing “more aggressively” to find possible baseline effects may be valuable. We have therefore considered the specific points raised above by Reviewer 1 (point 4), as well as by Reviewer 4 (point 2.) and **have now carried out several further investigations to help address this**, as summarised below:

- Following Reviewer 1's recommendations (**4.a.**), we have now included a PCA reduction (one common form of variable clustering / latent factor identification) at a wide range of dimensionalities, (d=1 to 700), from all 6,301 nIDPs (a rich set of non-imaging-derived phenotypes and demographics, all acquired before or at baseline visits) across the 785 participants of this study, and **did not find any PC that was different at baseline** between the control and SARS-CoV-2 groups.

- To address more specifically the question of whether baseline cognition might be showing a subtly different pattern between the two groups (4.b.), we have also carried out a focussed PCA ($d=10$) over 516 cognitive scores obtained before infection. While no single score (of the 516) was significantly different at baseline between controls and cases, we found **two PCs (PC1 and PC4) that were significantly different at baseline**. PC1 was mainly constituted of fluid intelligence, duration to complete trails A and B from the Trail Making Test and correct number of symbol digit matches, while PC4 was mainly composed of time to complete a numeric memory test. Importantly, **neither of these PCs had any notable effect on our longitudinal brain imaging results** (see next paragraph).

- We have carried out a more aggressive and comprehensive testing of whether baseline nIDPs can explain away longitudinal results (*point 4*):

- (i) by allowing 50% of missing data instead of 20% previously, we thereby expanded the number of nIDPs tested from 1,734 to 6,301, and (ii) only allowing for a 25% instead of a 50% reduction in Z, after including each nIDP individually as an additional confound in the model, for each of the “top 10 IDPs” (significantly different longitudinally, 10 from hypothesis-driven and 10 from exploratory analyses), to make the test more sensitive. Across the 6,301 nIDPs tested, we applied no kind of correction for having multiple tests here, wanting to maximise sensitivity to finding any baseline factors that might “explain away” the longitudinal results.
- as an additional supplementary analysis, instead of adding each nIDP as an additional confound one at a time, we have estimated the top 100 components from a PCA across the 6,301 nIDPs, and then added this *entire* 100-variable matrix of potential confounds into the main model, for each of the top 10+10 significant IDPs.
- focussing on baseline cognition, we have added PC1 and (separately) PC4, and also all 10 cognitive PCs, as confounds into the main longitudinal model.

From all of these additional analyses, we have **not found any results suggesting that nIDPs at baseline can explain the longitudinal effects**, in any of these more aggressive analyses. For cognition in particular, the largest reduction in Z was found for crus II of the cerebellum when adding PC1 to the model, with a **decrease in Z of only 5.7%** (from $Z=4$ to $Z=3.77$), while the Z-statistics for all other IDPs were reduced by less than 5%. Adding PC4 to our main model **reduced Z by 0.4% at most**.

Further details can be found in the Methods and Results sections, and relevant Supplementary Analyses:

Methods: The two groups showed no statistical differences across all 6,301 non-imaging phenotypes after FDR or FWE correction for multiple comparisons (lowest $P_{fwe}=0.12$, and no uncorrected P values survived FDR correction). However, due to the stringent correction for multiple comparisons that this analysis imposes, we investigated further whether subtle patterns of baseline differences could be seen using dimension reduction with principal component analysis on all 6,301 variables, and using a separate principal component analysis focused on baseline cognition (see **Supplementary Analyses**). We found no principal component that differed significantly between the two groups when exploring all the non-imaging variables. With respect to cognitive tests, while no single cognitive score was significantly different at baseline between controls and future cases, we identified two cognitive principal components that were different (**Supplementary Analyses**). These subtle baseline cognitive differences suggest slightly lower cognitive abilities for the future cases when compared with the controls. Importantly, none of these principal components—cognitive or otherwise—could statistically account for the longitudinal imaging results (see **Additional tests of potential influence of baseline non-imaging factors**).

Additional tests of potential influence of baseline non-imaging factors on significant results: We repeated the main analysis modelling for those top 10 IDPs found to show longitudinal differences between the SARS-CoV-2 and control groups, across both hypothesis-driven and exploratory approaches. For each of 6,301 non-imaging variables available (see **Methods**), we included that variable as an additional confounder in the longitudinal analyses. On the basis of the regression Z-statistic values, the strength of the original associations was not reduced by more than 25% for any of the non-imaging variables.

We further carried out the same analyses, but using dimension reduction (principal component analysis) applied to these 6,301 non-imaging phenotypes ($d=1$ to $d=700$), and also just focusing on cognition, with 540 cognitive variables ($d=10$). We found no substantial reduction in our longitudinal results with any of these principal components. In particular, for cognition where two components were significantly different at baseline (PC1 and PC4, **Supplementary Analyses**), the strongest reduction in Z was found for crus II of the cerebellum when adding PC1 to the model, with a decrease in Z of only 5.7 % (from $Z=4$ to $Z=3.77$), while the Z values associated with all the other IDPs were reduced by less than 5%. Adding PC4 to our main model reduced Z by 0.4% at most.

All of these additional analyses give us confidence that the pattern of non-imaging phenotypes' baseline differences, showing subtle differences in cognition between the control and SARS-CoV-2 groups, has no bearing on our longitudinal imaging results.

c. Concerns remain and should be addressed, at a minimum by stating them clearly as a limitation in the interpretation of the results. Specifically, the number of markers of pre-COVID co-morbidity is quite limited and may not be sufficient to properly balance cases and controls

As requested, we have now also **considerably expanded our comments on possible selection bias effects**, in the limitation section of the Discussion:

Discussion: (...) An illustrative example of the benefit of a longitudinal design is that, if looking solely at cross-sectional group comparisons at the second timepoint post infection (i.e., the analysis that would, by necessity, be carried out in post hoc studies), the strongest effect is seen in the volume of the thalamus. This effect disappears when taking into account the baseline scans however, since the thalamus of the participants who will later become infected appears to already differ from the controls years before infection. This highlights the difficulties in interpreting cross-sectional post-infection imaging differences as being necessarily the consequence of the infection itself. One of the benefits of the longitudinal design is to reveal differences over time—in this case a timeline marked by SARS-CoV-2 infection for one of the two groups—above and beyond any potential baseline differences. Difficulties of interpretation can remain in a longitudinal study of course, but the existence of a baseline pre-infection scan, combined with the matching of the patients and controls for age, sex, scan interval and ethnicity, increases the interpretability of the results, and confidence that the longitudinal abnormalities observed in those limbic and olfactory-related brain regions are, whether directly or indirectly, associated with the infection. When looking at brain imaging baseline differences between the two groups across all IDPs, particularly in an age-modulated way, we did find a few further significant baseline differences beyond the volume of the thalamus (**Table S2**). These were principally using diffusion imaging, but also using grey matter volume in the subcortical structures. Importantly, none of these baseline imaging differences spatially overlapped with the regions found to be different longitudinally (**Figure S7**). As this study is observational (as opposed to a randomised interventional study), one cannot make claims of disease causality with absolute certainty however, but interpretational ambiguities are greatly reduced compared with *post hoc* cross-sectional studies. A question remains as to whether the two groups are actually perfectly matched, as controls and cases could not be randomised *a priori*. Across the main risk factors, as well as thousands of lifestyle, health data and environment variables available in UK Biobank, we did not identify any significant differences when looking at each variable in isolation (only a few variables showed some trends at $P < 0.001$ uncorrected, see **Table S4**). This does not preclude the possibility of a sub-threshold pattern of differences making one group more at risk of getting infected by SARS-CoV-2, and this risk perhaps interacting with the effects of the coronavirus. This motivated the use of principal component analyses, both applied across the entire set of non-imaging phenotypes, and, separately, focusing on cognitive variables only (**Supplementary Analyses**). This aimed to find clusters of variables, which might give greater statistical sensitivity to finding baseline differences, compared with investigating individual non-imaging variables. Only the latter analysis revealed two principal components that were significantly different at baseline, suggesting subtle lower cognitive abilities in the participants who went on to get infected. Importantly, neither of these two cognitive components had any bearing on our longitudinal imaging results (reducing at most the strength of Z-statistics from $Z=4$ to $Z=3.77$ for the crus II of the cerebellum, when added in as an extra confound to the longitudinal analysis). Whether any of these imaging and cognitive differences at baseline played a subsequent role in those patients being more likely to get infected by the coronavirus, or to develop symptoms from infection, would need further investigation.

5. *One potential way to deal with the question of the specificity of the findings is to examine an additional matched control cohort who have been carefully diagnosed with non-COVID-19 pneumonia.*

Fewer than 4,000 of the UK Biobank participants (around 3,000 before the pandemic, plus the COVID-19 re-imaging study after the start of the pandemic) have been scanned twice; this means numbers of participants for each medical condition emerging *between* the two scans will be quite low. Nevertheless, fortunately there are **11 cases of pneumonia not related to COVID-19** who meet that criterion. Together with 261 matched controls, we felt this was just large enough to provide a useful investigation, in part because this is similar group size to the COVID-19 hospitalised group that showed significant results (and indeed the results do include some significant group differences). This longitudinal investigation of the effects associated with pneumonia showed some **significant group differences in IDPs, but with no overlap with those IDPs we found for SARS-CoV-2**.

We have also looked into influenza, but there are only 5 cases. This group may be too small to be useful; we nevertheless matched it to 127 controls, and found no significant results for the main model, except for a very few when focusing on hospitalised cases (N=3) vs controls.

These two new “specificity” evaluations results have been added to the manuscript, and described in detail in the Supplementary Analyses:

Additional, out-of-sample tests of longitudinal effects of pneumonia and influenza: To investigate whether pneumonia might have had an impact on our longitudinal findings, we assessed the age-modulated effects associated with pneumonia in an out-of-sample UK Biobank cohort that had been scanned twice. We identified 11 participants who contracted pneumonia not related to COVID-19 between the two scans, matched these to 261 controls, and applied our main analysis (Model 1) to these two groups. This longitudinal investigation showed some significant group differences in IDPs, but with no overlap with those IDPs we found for SARS-CoV-2 (all in the white matter, **Supplementary Analyses**). Overall, correlation between all IDPs’ (unthresholded) Z-statistics from pneumonia and SARS-CoV-2 longitudinal group comparisons was very low ($r=0.057$).

The sample size of cases who contracted influenza between the two scans in the out-of-sample UK Biobank cohort was unfortunately much smaller ($n=5$, including $n=3$ hospitalised cases), likely due to the low probability of influenza being recorded by a medical professional (GP or hospital). Nevertheless, for completeness, we also assessed longitudinally these two very small groups, compared with 127 matched controls. No result was significant for the 5 influenza cases, although a few IDPs showed significant longitudinal age-modulated effects, with just one IDP in the brainstem common to the SARS-CoV-2 findings (**Supplementary Analyses**). Correlation of Z-statistics between influenza and SARS-CoV-2 longitudinal group comparisons was again low ($r=0.077$).

- 6. The second major concern with the study is that the authors don’t provide adequate explanation for what motivated their use and choice of an “age at time of COVID-19 infection modulation factor” for the case control binary variable. Why is this necessary to include? Please verify what the parameter estimates reflect. It appears to be a weighted average, where older people have higher weights. The authors should first report the results from analyses when this exponential term is not included in the model. If the results differ, how do they understand the results? How did they determine that this is a reasonable correction factor to use? What happens if a different correction*

factor is chosen? They are missing a sensitivity or robustness estimate for their modeling choice. The modulation factor is taken from a meta-regression analysis that studied fatality rates. Clearly not an issue here where all the participants are living, and most had mild COVID-19 infections.

Reviewer 1 is right: this age modulation factor accounts for more pronounced effects in older people. We chose to focus on an **“objective” age model given the strong prior knowledge of a highly increased detrimental effect, at older ages, of SARS-CoV-2 infection** and a greater vulnerability of the brain with age. We therefore used the only already-published data-driven curve of age-effect in SARS-CoV-2 infection we could find at the time, with no free or subjectively-chosen parameters—hence having the same degrees-of-freedom fitting power as a binary model. (Since then, we have identified a separate set of data for age-dependent mortality rates, from the UK Office of National Statistics; this has given an almost identical exponential age dependence: males: 0.0451 ± 0.0008 (units 1/years), females: 0.0475 ± 0.0005 , very close to the value of 0.0524 in the Levin study.) In another analysis, of *hospitalisations* due to COVID-19, the function of age has also been found to be exponential in form, with exponent dependence ~ 0.02 (Palmer et al. (2021). *J R Soc Interface* 18(176)). If we use this age modulation instead, extremely similar results are found for the hypothesis-driven Z-statistics, $\text{corr}(Z)=0.96$, with the strongest original results ($|Z|>3.5$) *increasing* in strength by an average of $Z=0.16$. This is **now addressed in more detail in Supplementary Analyses**, and referred to in Methods:

Methods: The main case-vs-control group difference regressor of interest g_m is therefore formed using the demeaned binary two-group variable g and the age at Scan 2 a_2 :

$$g_m = g \times 10^{a_2 \times 0.0524 - 3.27} \quad [1]$$

where the age-dependence constants are taken from the meta-regression analysis⁷⁸ (see **Supplementary Analyses** for further details on age-modulation choices, and discussion of non-modulated modelling results).

Finally, we note that, as a general rule, if a model was suboptimal, the main result would be a loss of sensitivity, not an increase in false positives (see *point 1.b* by Reviewer 4).

In the interest of completeness, we have **now also tested—as a secondary analysis—the binary modelling between control and SARS-CoV-2 groups**, with all other factors held the same. These two (age-modulated and non-modulated) models do not give very different primary results (hypothesis-driven results presented in Tables 1 and S1). **The findings are highly similar, if a little weaker, consistent with our expectation of increased effects at higher ages.** For instance, from the 10 most significant results reported in Table 1, 10 associations had passed FDR correction, and 6 passed FWE correction. When switching to the binary (non-modulated) group modelling, 9 out of 10 continue to pass FDR correction and 4 out of 6 pass FWE correction. The correlation between the Z-statistics from all 297 hypothesis-driven IDPs is high ($r=0.88$), and the mean change in $|Z|$ across the 10 top associations was a reduction of just 0.33. We have added

results obtained with the binary regressor in all four Models for comparison in Supplementary Table S11.

- The final major concern is that the conclusions overstate the significance of the results and pay little attention to the serious limitations. The basic claim made (e.g. that the changes exist in those who have been infected with the virus and the only question remaining is whether they persist) is misleading. Rather, this study may have found quantitative brain MRI evidence for pathophysiological changes in some people due to infection with COVID-19 in the brain, but that remains to be confirmed with further studies. The most critical limitations arise directly from the challenges of proving that the cases and controls are properly matched when doing a study with this design. But they also overstate the findings in their conclusions, even when properly reporting in the results the percentages of infected participants that showed a greater longitudinal change than the median value in the control ranged from 56-62% of the cohort. It is interesting to note that when drawing attention to the benefits of their longitudinal design over cross-sectional group comparisons, they clearly articulate the difficulties in interpreting cross-sectional post-infection differences (bottom of Page 18-top of Page 19). The same should be done for their main results from the longitudinal analyses.*

We believe to have been particularly careful in emphasising the limitations of this study, including by making it clear that it cannot replace an RCT in terms of unambiguously establishing causality (p19). In addition, we have stated clearly that the origin of such brain differences can be manifold, including the possibility that the effects on the limbic/olfactory-related regions are non-specific to COVID-19 but rather secondary to some of its complications or symptoms. With respect to presenting the results, we visually show each datapoint for transparency, and report, as mentioned by Reviewer 1 above, % of the SARS-CoV-2 cases deviating from the controls' median for the main findings. We do really appreciate however the need for this study to be exceptionally cautious. As mentioned in response to *point 4.d.* above, we have now considerably **expanded our comments on baseline risk factors differences**, and their possible interplay with COVID-19. In addition, we have made sure to correct any part of the text that might be worded too strongly throughout the manuscript. Finally, we have **now also emphasised in the first paragraph of the Discussion the need to keep in mind these percentages given in the results:**

Discussion: It is worth noting that these structural and microstructural longitudinal significant differences are modest in size, the strongest differences in changes observed between the SARS-CoV-2 positive and control groups corresponding to around 2% of mean baseline IDP value. Our statistics also represent an average effect; not every infected participant will display brain longitudinal abnormalities. Indeed, SARS-CoV-2 positive participants had a greater longitudinal change than the median value in the controls in 56% of the cases for the regions connected to the temporal piriform cortex, 62% of cases

for the regions connected to the olfactory tubercle, 57% for the left parahippocampal gyrus and 60% for the left orbitofrontal cortex.

8. *The authors note a limitation of the study is the potential misclassification [e.g. COVID positive cases not all positive due to false positives in testing and COVID negative cases actually positive due to absence of confirmed negative status and or false negative tests]; however, that should bias toward the null hypothesis of no case:control difference.*

Thanks - indeed. We have now **made this clearer in the relevant paragraph:**

Discussion: However, it is worth noting that any potential misclassification of controls as positive cases (due to false positives in testing) and positive cases as controls (due to the absence of confirmed negative status and/or false negative tests) could only bias our results toward the null hypothesis of no difference between cases and controls.

9. *The criteria used to remove IDPs were all appropriate, but there was minimal information provided about the removed IDPs. Please report the number of removed IDPs separately for the cases and controls. Also report metrics about which IDPs were removed by image data type and any patterns in the missing and outlier data. For example, it appears that many of the fMRI derived IDPs were removed, why is that?*

We removed *a priori* individual-connection functional connectivity edges and task-fMRI activation IDPs, as these have very low reproducibility and have for example previously been found to have very low heritability, due primarily to being intrinsically noisy (e.g., Smith et al., Nat Neurosci 2021). However, a reduction of the hundreds of functional connectivity edges to a greatly-reduced set of 6 latent factors achieves a large noise reduction and signal boost (e.g., for use in heritability estimates, as discussed and presented in Elliott et al., Nature 2018), and so we retained those 6.

However, we suspect that the reviewer is more likely asking for more details about which IDPs were removed *during* the preprocessing of IDPs prior to the longitudinal modelling. IDPs (both existing and new) were removed automatically as follows:

- (a) For all cases and controls, IDPs were removed if their scan-rescan reproducibility, averaged across cases and controls, was less than $r=0.5$. This reduced the number of IDPs from 2,630 to 2,048.
- (b) For all cases and controls, IDPs were removed if less than 50 participants (in total) had valid data. This removed just one IDP from the full ("Exploratory") list, leaving 2,047 IDPs.
- (c) For each remaining IDP, for each subject, an IDP was set to "missing" if it was more than 8 times the median absolute deviation around the median (across all subjects). This outlier removal is applied at both the individual-timepoint stage in IDP preprocessing, and also after computing the IDP2-IDP1 longitudinal change.

Missing data for individual subjects and specific IDPs can therefore occur because of step (c) above, or because the IDP was missing in the original data (e.g., because a given

modality was not usable from a given participant). The fraction of total non-missing data, averaged across IDPs, is 0.93; all full xls results tables include the number of usable measurements for each IDP and for each statistical test. Importantly, there was no imbalance in amount of missing/outlier data between cases and controls: the number of cases with usable data, normalised by the total number of subjects with usable data, has the following percentiles across IDPs: percentiles [0, 1, 50, 99, 100] = 0.50, 0.50, 0.52, 0.52, 0.60, i.e., the median percentile is 0.52. From this analysis, the only 3 IDPs having this fraction greater than 0.53 were thalamic nuclei diffusion IDPs, which do not appear in any of our main results. These are also the only 3 IDPs with more than 24% missing/outlier data. We would be happy to provide an even finer-grained breakdown of the patterns of missing/outlier data across IDPs and/or subjects, if the reviewers/editor would like additional Supplementary Material, but we believe that the above summaries are sufficient.

All of the new information above is now included in the revised paper:

Methods: Functional connectivity is intrinsically noisy when each region-pair connection is considered individually, so we focused here our analysis on 6 dimensionally-reduced functional connectivity networks⁶⁵. We also did not consider *a priori* task-fMRI activation IDPs, as these have previously been found to have very low reproducibility and heritability⁶⁶.

Methods: Outlier values (individual IDPs from individual scanning sessions) were removed on the basis of being more extreme than 8 times the median absolute deviation from the median for a given IDP. Missing data for individual subjects and specific IDPs can therefore occur because of this step, or because the IDP was missing in the original data (e.g., because a given modality was not usable from a given participant). The fraction of total non-missing data, averaged across IDPs, is 0.93; all full results tables include the number of usable measurements for each IDP and for each statistical test. Importantly, there was no imbalance in amount of missing/outlier data between cases and controls: the number of cases with usable data, normalised by the total number of subjects with usable data, has the following percentiles across IDPs: percentiles [0, 1, 50, 99, 100] = 0.50, 0.50, 0.52, 0.52, 0.60, i.e., the median percentile is 0.52. From this analysis, the only 3 IDPs having this fraction greater than 0.53 were thalamic nuclei diffusion IDPs, which do not appear in any of our main results. These are also the only 3 IDPs with more than 24% missing/outlier data.

10. *The authors' state that the counter intuitive increases in cortical thickness in the L rostral anterior cingulate cortex and orbital frontal cortex in the control cases that is not seen in the cases is reported "consistently" in the literature, but only cite a single reference that first reported this. What further evidence supports that this is a consistent finding?*

Thank you for pointing this out. Since it was mentioned briefly in the caption, and references are limited, we had listed only the historical reference. This is however an effect consistently observed in the anterior/rostral/subgenual cingulate cortex and orbitofrontal cortex in many ageing/lifespan study of grey matter thickness, as seen for instance in:

Fjell et al. (2009) *Cerebral Cortex*. High consistency of regional cortical thinning in aging across multiple samples. <https://academic.oup.com/cercor/article/19/9/2001/281320>

"The ACC was not consistently prone to thinning with age, and showed increased thickness in some of the samples. Finally, parts of the medial orbitofrontal cortex showed preservation with age, both in American and Scandinavian samples."

and:

Dotson et al. (2016) *Frontiers in Aging Neuroscience*. Age Differences in Prefrontal Surface Area and Thickness in Middle Aged to Older Adults. <https://www.frontiersin.org/articles/10.3389/fnagi.2015.00250/>

"In contrast, cortical thickness analyses revealed greater cortical thickness at older ages in the (...) bilateral ACC (left $B = 0.007$, $p = 0.021$, $R^2 = 0.269$; right $B = 0.009$, $p = 0.011$, $R^2 = 0.407$) after controlling for mean thickness, while age was not associated with cortical thickness in the OFC."

and:

Zhao et al. (2019) *Cerebral Cortex*. Age-related differences in brain morphology and the modifiers in middle-aged and older adults. <https://doi.org/10.1093/cercor/bhy300>

" (...) quadratic age effects [as in higher in older age] were found in (...) the medial orbitofrontal cortex (mOFC), (...) and the cingulate cortex, for all measures"

"These functional and structural findings support the hypothesis that the increased CTh [cortical thickness] at late ages in the ACC, which was consistently observed in the current (Fig. 1) and previous studies (Thambisetty et al. 2010; Fjell et al. 2014; Storsve et al. 2014; Yang et al. 2016) is likely related to neuroplasticity at older ages (Engvig et al. 2010)."

Because of references constraints, we have now also **added the last of these references**, as it itself refers to many other previous publications:

References: 29. Zhao, L. et al. Age-Related Differences in Brain Morphology and the Modifiers in Middle-Aged and Older Adults. *Cereb Cortex* 29, 4169-4193, doi:10.1093/cercor/bhy300 (2019).

11. *Minor points*:

a. Please clarify why standard scaling by $\sqrt{N-3}$ is used instead of $\sqrt{N-3-\text{number of confounds}}$ when the correlations were Fisher-transformed to Z-statistics.

Thanks to both Reviewer 1 and Reviewer 4 (*point 5.i.*) - you are correct, although the relative change to N in this correction is very small, and we had already verified that we obtain almost identical results (mentioned briefly in the original manuscript) when using regression, which explicitly corrects degrees-of-freedom for number of confounds, so the effect on all results is very small indeed. For example, correlation between Z-statistics computed using the two approaches is $r=1.0000$, mean absolute difference between Z-statistics is 0.0035, correlation between $-\log_{10}(P_{fwe})$ is 0.9996 and mean absolute difference is 0.005.

In any case, as we have now switched to the equivalent regression option as mentioned above, **the degrees-of-freedom are now adjusted** for the number of confounds. We now **mention this in the Methods section**:

Methods: Here, Z is more useful than T, because different IDPs have different patterns of missing data, and hence Z is more usefully comparable across IDPs. The regression inference automatically takes care of the degrees-of-freedom, including accounting for missing data and confound variables.

b. The final sentences at the bottom of Page 23 seem to have editing errors (e.g. “finding” should be “findings” and “... correlated post hoc with the [longitudinal changes in the] cognitive...”. Please clarify.

Thank you very much. The typo has now been **corrected**, and the statement regarding *post hoc* correlation **clarified**:

Discussion: These findings remained significant after excluding the few hospitalised cases. In turn, the duration to complete the alphanumeric trail B was associated *post hoc* with the longitudinal changes in a part of the cerebellum linked to cognition, namely crus II, which is also specifically activated by olfactory tasks^{37,59}.

c. Pages 3, 4 and 18: have a reference improperly formatted as a DOI citation rather than a numeric value.

Unfortunately, EndNote at present does **not accept the formatting of pre-print references** (medRxiv, bioRxiv, etc.) so we chose to add the full DOI and will make sure to correct this.

d. References 41 and 56 are the same.

Thank you - this has now been **corrected**.

e. Figure S2. Y axis is improperly labeled.

Apologies; this is best dealt with by **clarifying this in the caption**, which now states that these are histogram density plots for the interscan intervals of the different participant groups, with each of those normalised to have peak of 1 to make relative comparison easy:

Figure S2: Histograms showing the well-matched distributions of Scan 1 - Scan 2 intervals for case and control groups. The below IDP reproducibility **Figure S3** shows, for comparison against the cases and controls, reproducibility from around 3,000 (2,943) UK Biobank participants who had returned for a second scan prior to the pandemic; hence we also show here the interscan intervals for this “3k” group, with tighter control over this interval (we have normalised each of those 3 groups to have a peak of 1, to make the relative comparison easier).

Reviewer #2:

With interest, I read the manuscript “Brain imaging before and after COVID-19 in UK Biobank by Douaud et al. However, there is a major principal and basic concern that not only substantially limits enthusiasm for the study, but also precludes a recommendation to publish these data:

Why is that: there is a basic problem with the way neuropsychology was evaluated, specifically, which tests were used, and what conclusions were drawn on the basis of the used tests. In short: reduced performance in an executive task, when the mnemonic system (parahippocampal gyrus etc.) is supposed to be affected exemplifies that things don’t fit together.

We have now replaced our set of cognitive scores, using an objective, data-driven approach based on out-of-sample participants who are the most likely to show cognitive impairment. We show below that both original and new tests do measure what we meant to capture: possible cognitive impairment.

We appreciate that Reviewer 2’s expertise is in neuropsychology, and that they are therefore focusing on—and frustrated by—this ancillary aspect of what is first and foremost a longitudinal brain imaging study. It would seem however that Reviewer 2 is perhaps anticipating a one-to-one mapping between cognitive function (e.g., memory) and relevant regions of interest in the brain (e.g., parahippocampal gyrus), but such a mapping is not as simple as this (more on this below).

We do, for better or worse, entirely depend on the information that is available in the UK Biobank. One cost of having this amazing (imaging) Big Data resource, the largest of its kind and the source of this unique longitudinal COVID-19 dataset, is that some aspects of it cannot be covered in as much depth as with a study dataset dedicated to a single question; as a consequence, the neuropsychological testing, as we have mentioned (p36), “offers *limited* measurements of cognitive function in UK Biobank”. We would like to note however that, while “brief and bespoke”, the cognitive assessment in UK Biobank can show very good agreement with the reference tests (Fawns-Ritchie and Deary, 2020; again, more below).

- 1. The manuscript reports many changes in mild-to-moderate post-C19 patients, which should have their clinical correlation. This cannot, however, be drawn from the presented data. While I am having difficulties in interpreting the tables (which indicates that they are not well preprocessed), a z-score of 4 as an indicator for describing a difference appears to be high. Comparing Post-C19 to non-Post-C19 patients would be extremely helpful to make this z-score difference understandable, namely to grasp what this difference actually means.*

These Z-statistic values, as was explicitly **highlighted in the caption of each main Table**, “reflect the statistical strength of the longitudinal group-difference modelling, and are not raw data effect sizes”. We understand that relying on *r* or *Z* only could be confusing, and

have therefore switched to (equivalent) regression modelling, and **now report in the Tables the % group-difference IDP change**, as described in detail above (see also *point 2.* by Reviewer 1 and *point 1.c.* by Reviewer 4). We hope that this change will make these Tables easier to understand.

2. *Most importantly, the respective “references” of brain imaging, namely the neuropsychological tests, are highly problematic: in addition to the „classical“ clinical COVID-19 symptoms, all investigated C19 patients are expected to have suffered for many months from symptoms, which are assigned to brain regions, where memory (parahippocampus), olfaction (olfactory cortex), or behaviour (amygdala) are located. The respective neuropsychological tests, which were used in this study, mostly probe executive-constructive functions. Two of the tests used in the study (insufficiently) assess memory functions. However, these two tests do not provide substantial differences in memory function (the delayed recall test would be adequate and should show differences – which it doesn't). On top, the assessment of the Trail-making test is incorrect: not the seconds need to be counted in absolute numbers (to execute the test), but rather whether the individual remains within a given time frame (in relative terms): pathology is given only if the subject does not adhere to the defined time frame.*

We agree with Reviewer 2 that deeper assessment of memory function would be good, but as explained above, we depend on what is available in the UK Biobank. We had indeed intended to probe more than just memory function (as we did not necessarily anticipate memory issues), and thus had selected several tests known to be sensitive to various aspects of cognitive impairment. In response to the reviewer's concerns, we now introduce an alternative, objective, data-driven approach to selecting the scores used to assess cognition, based on out-of-sample participants who are the most likely to show cognitive impairment (details below).

First, it is worth noting that we have not stated that memory function should be affected in the SARS-CoV-2 cases; only that their longitudinal brain differences compared with the controls are in limbic and olfactory regions that *also* subserve memory function and that, as such, a follow-up would be key to see if there is some partial recovery or further cognitive impairment, including memory-related. We were therefore not particularly surprised to find a longitudinal difference in executive function in the cases (no more than we were surprised *not* to find a difference in memory function). Indeed, based on previous literature, we simply chose an array of tests, from the set available to us in UK Biobank, known to be particularly sensitive to various aspects of cognitive impairment rather than being tailored to memory issues *per se*: three variables from the Trail Making Test (duration to complete trails A and B, errors for trail B), two variables based on Matrix Reasoning, one variable from the Symbol Digit Test (number of correct answers), one measure of reaction time, the fluid intelligence score, and one measure of delayed recall.

However, since Reviewer 2 has concerns with this set of tests, we have **now chosen instead to use an objective, data driven approach to select cognitive variables that are sensitive to cognitive impairment, determined using a separate (out-of-sample) cohort of Biobank participants who are most likely to show cognitive impairment.** For this, we have explored the *out-of-sample* cases in the UK Biobank who either had diagnosed dementia, or developed dementia at further follow-up visits, and who had undergone the most extensive set of cognitive testing (some tests were only introduced at later visits). We have identified 778 such cases, and have paired them with out-of-sample controls based on age at the time of cognitive testing, sex, ethnicity and qualification, amongst other variables. In the 7 originally-selected cognitive scores available in this out-of-sample cohort (two others being only available at later visits for a lower number of participants), these (pre)dementia cases show significant differences, except for the prospective memory test, which is significant only when focusing on *already* diagnosed dementia cases. This demonstrates that most of the **cognitive variables selected in our original analysis are indeed sensitive to various aspects of cognitive impairment.** To draw a more objective set of cognitive tests however, we have now selected the **top 10 most significant tests showing differences between out-of-sample (pre)dementia cases and controls**, the first 6 of which overlap with those we had selected in the original analysis. These are: three variables from the Trail Making Test (duration to complete trails A and B, errors for trail B), one variable from the Symbol Digit Test (number of correct answers), one measure of reaction time, the fluid intelligence score, one measure of numeric memory (maximum number of digits remembered correctly), and three variables of the Pairs Matching test (numbers of correct and incorrect matches, time to complete). We have **now carried out the same longitudinal analysis with these 10 cognitive variables**, and found the same results as before, pointing at differences, in SARS-CoV-2 cases vs. controls, in the duration to complete both numeric and alphanumeric trails A and B. The details of this **new cognitive analysis**, using objectively drawn variables, based on out-of-sample (pre)dementia cases, **have now replaced the original cognitive ancillary analysis in the Methods and Results**, and additional details can be found in Supplementary Analyses:

Cognitive analysis: While cognitive testing offers limited measurements of cognitive function in UK Biobank, we included in our ancillary cognitive analysis 10 variables sensitive to cognitive impairment. For this, we drew these variables using a data-driven approach based on identifying out-of-sample current and future dementia cases in UK Biobank, and comparing them to matched controls (**Supplementary Analyses**). The top most significant variables from this out-of-sample analysis were:

- three variables from the Trail Making Test: both durations to complete trails A and B, as well as the total number of errors made traversing trail B,
- one variable from the Symbol Digit Test: the number of symbol digit matches made correctly,
- one measure of reaction time: mean time to correctly identify matches at the card game “Snap”,
- one measure of reasoning: the “fluid intelligence” score,
- one measure of numeric memory: the maximum number of digits remembered correctly,

- three variables of the Pairs Matching test: numbers of correct and incorrect matches, and time to complete the test.

Cognitive Results: Using the main model used to compare longitudinal imaging effects between SARS-CoV-2 positive participants and controls (Model 1), we explored differences between the two groups in 10 scores from 6 cognitive tasks. These 10 scores were selected using a data-driven approach based on out-of-sample participants who are the most likely to show cognitive impairment (**Supplementary Analyses**). After FDR correction, we found a significantly greater increase of the time taken to complete Trails A (numeric) and B (alphanumeric) of the Trail Making Test in the SARS-CoV-2 infected group (Trail A: 7.8%, $P_{\text{uncorr}}=0.0002$, $P_{\text{fwe}}=0.005$; Trail B: 12.2%, $P_{\text{uncorr}}=0.00007$, $P_{\text{fwe}}=0.002$; **Figure 3**). These findings remained significant when excluding the 15 hospitalised cases (Model 2: Trail A: 6.5%, $P_{\text{uncorr}}=0.002$, $P_{\text{fwe}}=0.03$; Trail B: 12.5%, $P_{\text{uncorr}}=0.00009$, $P_{\text{fwe}}=0.002$).

We have also **expanded in the Discussion on this result**, in particular making clear that there is no impairment of memory function seen in the cases:

Discussion: In our sample of infected participants with mainly mild symptoms, we found no signs of memory impairment. However, these SARS-CoV-2 positive participants showed a worsening of executive function, taking a significantly greater time to complete trail A and particularly trail B of the Trail Making Test (**Figure 3**). While the UK Biobank version of the Trail Making Test is carried out online and unsupervised, there is good to very good agreement with the standard paper-and-pencil Trail Making Test on its measurements for completion of the two trails⁵⁷, two measures known to be sensitive to detect, and discriminate, mild cognitive impairment and dementia from healthy ageing⁵⁸. These findings remained significant after excluding the few hospitalised cases. In turn, the duration to complete the alphanumeric trail B was associated *post hoc* with the longitudinal changes in the cognitive part of the cerebellum, namely crus II, which is also specifically activated by olfactory tasks^{37,59}. In line with this result, this particular part of the cerebellum has been recently shown to play a key role in the association with (and prediction of future) cognitive impairment in patients with stroke (subarachnoid haemorrhage)*. It remains to be determined whether the loss of grey matter and increased tissue damage seen in memory-related regions of the brain may in turn increase the risk for these participants of developing dementia in the longer term^{2,4,60}.

* Accepted in Scientific Reports: Early brain injury and cognitive impairment after aneurysmal subarachnoid haemorrhage. Rowland et al., 2021.

We are concerned that **Reviewer 2** is focussing on the parahippocampus, olfactory cortex and amygdala (*"all investigated C19 patients are expected to have suffered for many months from symptoms, which are assigned to brain regions, where memory (parahippocampus), olfaction (olfactory cortex), or behaviour (amygdala) are located"*), when in fact many other limbic and olfactory regions of the brain, serving a wide range of functions, were identified, including the orbitofrontal cortex, anterior cingulate cortex and anterior insula, and other areas (please see Figure 2). If we understand correctly, there also seems to be the suggestion that each region (e.g., the parahippocampus) could only serve one (e.g.,

mnesic) function, when this is not the case; to take a concrete example in the general (older) population sampled in the UK Biobank project at the first imaging visit (~40,000 participants), **the strongest cognitive association with the left parahippocampus thickness is in fact found with the alphanumeric Trail B of the UK Biobank version of the Trail Making Test (errors), and not with memory scores.** Finally, regarding the remark by Reviewer 2 that some tests are administered differently in the UK Biobank, this is indeed the case. However, an extensive investigation by Ian Deary's group has shown that, in particular, the Trail Making Test—and predominantly trail B, our most significant result—is **strongly and positively correlated with the paper-and-pencil, standard version of the test** that neuropsychologists are more familiar with (Fawns-Ritchie and Deary, 2020). We have **included this information and reference in the Discussion** (see above).

3. *Minor:*

What also in at least surprising: 10 out of 11 changes were assigned to the left hemisphere: why is it, and how can be excluded that this is not rather a sign for an erroneous methodology?

We believe that Reviewer 2 is referring to p11: “While significant IDPs related to grey matter thickness were found, using Model 1, to be **bilateral for both the anterior parahippocampal gyrus (perirhinal cortex) and entorhinal cortex**, 10 of the 11 remaining significant IDP were left-lateralised”? If so, our sentence does mention the fact that some of these results are, in fact, bilateral. The sentence that follows in the manuscript sheds more light on this *apparent* lateralisation of some of the results: “We thus directly investigated (left - right) differences in the SARS-CoV-2 group only for those significant IDPs, and found that the **infected participants did not have significantly more reduced grey matter thickness on the left than on the right hemisphere** (lowest $P_{\text{uncorr}}=0.30$).” This (absence of) result suggests that there are some effects in those grey matter IDPs in the right hemisphere that are, while not significant, strong enough to show no marked difference with the left hemisphere. Figure 2, showing all effects with $|Z|>3$, highlights further the danger of solely relying on the list of significant IDPs, as it reveals again bilateral effects in e.g., the parahippocampal gyrus, temporal pole and anterior cingulate cortex. In summary, there are no significant differences between left and right hemispheres in our longitudinal results, and to the best of our knowledge, no erroneous methodology (we have followed the best practice for the software defining grey matter thickness, in active collaboration with those who developed this highly utilised and validated tool at MGH).

In conclusion, I am not doubting that the reported differences/numbers are true. However, I am strongly fearing that these differences and values are not resulting from or reflecting the reasons, which the authors put forward. The biological and truly functional relevance, based on my arguments above, namely that – for what is supposed to be measured - inappropriate tests were used, make me believe that

there is a non-causal correlation of alterations measured and reasons given/interpretation made (and remind me of the classical example of the correlation of the stork population and birth rate). A possible explanation of the alterations measured may be indirect or “downstream” or collateral changes of alterations upon virus entry through cranial nerves (reaching the olfactory bulb). However, in total, i.e. if I put all the tests together, namely what they can deliver qualitatively, as well the given interpretation, I am sorry to say that there is not much left that speaks in favor to follow the line of reasoning and claims of the authors.

We hope to have now shown clearly that the original tests we had used, and the new ones we use as (an even more objectively chosen) replacement, measure exactly what we meant to capture: possible cognitive impairment. We did identify, in the SARS-CoV-2 cases, signs of such cognitive impairment in executive functioning, worsening with age. This was associated with crus II of the cerebellum, a cognitive and olfactory-related brain region that we found to be one of the most severely altered areas longitudinally. Of note, we also have clearly stated that the design of this longitudinal study does not allow us to assess with certainty the causality of the observed abnormalities (see answer to Reviewer 1, point 4.c.). Finally, we would respectfully like to reiterate that this is primarily a brain imaging study. Of course, if it had been possible at all, we too would have liked to include even more in-depth characterisation of cognition and clinical symptoms, but we strongly believe that the striking results of longitudinal differences in the brain associated with SARS-CoV-2 infection stand on their own.

Reviewer #3:

this is a very well written manuscript from a highly knowledgeable team. The authors report a multimodal exploration of the longitudinal effects in the brain associated with SARS-CoV-2 infection, made possible by leveraging the unique resource of the UK Biobank project, and its COVID-19 re-imaging sub-study.

The UK Biobank repository is a vital, large-scale, neuroimaging resource for studies such as this and the authors leverage this in a timely manner to address what many have feared - that COVID-19 is not merely just a disease which affects the respiratory system but one that, even in recovery, may have extensive impacts on the brain.

In this study, N=785 participants had two usable brain scan sessions: N=401 SARS-CoV-2 positive participants(cases), and N=384 controls who were matched for age, sex, ethnic background and interval between the two scans, as well as various risk factors. Hypothesis-driven and exploratory longitudinal analyses conducted by the authors revealed significant, neurological impacts associated with SARS-CoV-2: affecting mainly in the olfactory cortical system, as evident from diffusion measures which served as proxies for tissue damage; in regions functionally connected with the piriform cortex, olfactory tubercle, and anterior olfactory nucleus; as well as a more prominent reduction of grey matter thickness and contrast in the infected participants in the left parahippocampal gyrus and lateral orbitofrontal cortex.

What is more, while the greater atrophy for the SARS-CoV-2 positive participants was reported to be localized to a few, mainly limbic, regions, an increase in cerebrospinal fluid volume and decrease of whole brain volume suggests a diffuse loss of grey matter in addition to the regional effects observed in the olfactory-related areas.

Additional examination demonstrated that effects associated with infection, both in the cortical thickness and mean diffusivity, extended mainly to the anterior cingulate cortex, insula, supramarginal gyrus and the temporal pole. Comparing those patients (N=15) who had been hospitalized with COVID-19 against comparable, non-hospitalized cases demonstrated a more distributed pattern with greater reduction in grey matter thickness in fronto-parietal and temporal regions.

Finally, significant cognitive decline was noted, which persisted even after excluding the hospitalized patients, in the SARS-CoV-2 positive group between the two imaging timepoints. This decline was associated with greater atrophy of the crus II sub-region of the cerebellum.

Such a study is very timely and would be of much value to the scientific community and to the public. I would be in strong favor of its publication at the earliest possible time to do so. Given the ongoing debate as to the long term effects associated with COVID-19 survival - especially in unvaccinated individuals - this study lends further support for public health measures to inoculate people against SARS-CoC-2 and its variants.

We would like to thank Reviewer 3 very much for this very positive appraisal of our work and of its timeliness.

Reviewer #4:

The study by Douaud et al used a large imaging dataset from UK Biobank to investigate the changes in the brain by comparing a long list of brain measures between two time points (roughly three years apart). Two groups of subjects were included: 401 subjects were infected by SARS-CoV-2 between the two time points while 384 were controls. Over 2,000 brain measures (called image-derived phenotypes in the paper) plus various behavioral (cognitive) and non-imaging measures were considered. The goal was to find out which measures had a different trajectory in terms of brain change in the patients compared to the controls. Largely based on the statistical results from a sophisticated model, they found changes in some measures that were associated with the olfactory system.

The far-reaching impact of the current pandemic is an important topic from various research perspectives. The current study could contribute to our understanding about the short-term effects in the brain. With relatively large sample sizes, this imaging dataset provides a unique opportunity to gain various insights about the neurological impact of the COVID-19 virus. For example, even without involving their main models, the information extracted from a few brain measures is revealing as shown in Figure 1, Figure 3 and Figure S6. In addition, the authors presented a large amount of modeling work, and made available the analysis code and the full list of analytical results.

However, there are many methodological issues with the paper, some of which are quite severe. For example, the assumed age trajectory adopted from an epidemiological study is unjustified for curve fitting. In addition, the evidence for the lack of baseline differences is not convincing. Because of these methodological issues, the paper tends to emphasize “more on the destination (research claims and metrics) than on the journey” (Calster et al, 2021. *Methodology over metrics: current scientific standards are a disservice to patients and society. Journal of Clinical Epidemiology.* <https://doi.org/10.1016/j.jclinepi.2021.05.018>). As the result interpretation and the discussion of this study currently lack a solid foundation, the review below will focus only on the modeling aspect.

Thank you to Reviewer 4 for pointing out the potential contribution of our study to the ongoing COVID-19 research, and for acknowledging our efforts in making openly available our extensive results and code. We have now discussed the adopted age trajectory in detail above (please see answers to Reviewer 1, *point 6.*), and have carried out comprehensive and more aggressive analyses of baseline differences (primarily discussed in detail above, in answer to Reviewer 1 *point 4.*, but also below).

1. *An unjustified assumption was adopted regarding the trajectory of brain changes relative to age. The authors intended to explore the age trajectory of brain changes during the period of ~3 years for each of those thousands of variables, and compare whether the patients and the controls differed in terms of their age trajectories in brain changes. To be able to make such comparisons, the authors assumed an age trajectory of exponential relationship $10^{(\text{Age} \times 0.0524 - 3.27)}$ based on the age dependence of infection fatality rates estimated by Levin et al (2020; doi:10.1007/s10654-020-00698-1). For several reasons, this assumption is inappropriate.*

We appreciate this concern, and hope that our detailed response (see Reviewer 1, *point 6.*), additional Supplementary Analyses and results are useful with respect to this issue.

a. Lack of justification. In fMRI studies, an empirical curve for hemodynamic response is typically used to capture the BOLD response. In that case, the empirical curve at least is rigorously verified through many hemodynamic response experiments. Even though the larger death rate of COVID-19 patients with older people is roughly captured to some extent by the exponential relationship of $10^{(\text{Age} \times 0.0524 - 3.27)}$ as shown in Levin et al (2020), such an empirical curve based on observational data is embedded with some extent of uncertainty. More crucially, it is difficult to see how the death rate among patients could be applied and related to the age trajectory of brain measures among healthy subjects. Even for those patients, it remains questionable as to why those brain measures would follow the same pattern as the death rate.

Again, we appreciate this concern, but, to briefly summarise what we have written above and the new results from Supplementary Analyses, we feel it is clear *a priori* that some form of age-dependence in the model is sensible, given the strong **prior knowledge of a highly increased detrimental effect of COVID-19 at older ages**, and a greater vulnerability of the brain with age. In addition, the results do not **depend strongly on the exact choice of age-dependence made** – even without any age modulation in the main model, many of the primary results remain. We have also discussed above two additional (external) estimates of the form of the age-modulation in COVID-19, and found that these

give very similar results (all in Reviewer 1 point 6. and corresponding Supplementary Analyses).

b. No accommodation for the heterogeneity of age trajectory across measures. Even if such a trajectory of death rate based on epidemiological data could be a reasonable approximation for some brain measures, it remains problematic for other measures since there is likely a large amount of variation across those thousands of measures as shown in the limited few examples in Figures 1, 3, and S6. Some of the patterns in those three figures do not seem to match the assumed pattern of an exponential curve. Needless to say, it is difficult to assess the sensitivity of these models if the empirical relationship is not accurate enough.

We agree that the model may not be *optimal* for every feature (IDP) considered; in other words, this model might not be the most sensitive possible model for every IDP. However, where this is the case, the main expected outcome would be that we could fail to find significant results, and not that there would be any inflation of false positives. This is always the case when choosing any given model. Were we to use more flexible (less parsimonious) models, the use of F-tests with multiple (numerator) degrees of freedom would likely result in reduced power, and the use of active model selection would require a multiple testing correction that would again reduce power.

Nevertheless, we have now also added new Supplementary Analyses, showing **no obvious evidence of structured problems in model residuals**. We created diagnostic plots from the top 10 most significant IDPs showing longitudinal differences between cases and controls (10 from the hypothesis-driven approach, 10 from the exploratory), with scatterplots of residuals against model-predictions, showing **no evidence of model misspecification**. We also include QQ plots of model residuals for the top 10+10 IDPs, again with **no evidence of model-fitting problems**. Finally, we **assessed sensitivity to additional, higher-order, confound variables**, with up to 3-way interactions, finding **no large changes in the main results** (all in Supplementary Analyses).

c. Poor interpretability of the analytical results. Under the current model structure, each model is meant to match the trajectory of a brain measure change with the death rate pattern, and then to obtain the difference of the pattern matching between patients and controls. However, the reported correlation “r” values in all the tables are difficult to interpret. In the end, the results largely stop at the qualitative, not quantitative, level. For example, Table 4 shows the “top 10 significant results” with the correlation “r” column under Model 1 in the range of [-0.14, 0.16]. Are those “r” values considered high or low in terms of brain measure difference between the two groups? In other words, it does not tell us the amount of decrease (or increase) for each measure (e.g., brain volume) between patients and controls. Also, the presentation of these results begs the question: why no “r” values were provided for models 2-4 in those tables?

Thanks - this was purely to avoid over-cluttering the table. However, as both Reviewer 1 (see also *point 2.*) and Reviewer 4 commented on reporting the effect sizes in a more easily interpretable way, we have **now replaced r values with % change and standard errors**, for all models. This does not change the statistical analyses (e.g., P-values), but we agree that it does improve the interpretability of the results, and ties in nicely with the information presented in the figures.

In addition, the Z-values in the tables do not reveal the amount of brain measure either other than its statistical significance. It would be better to provide quantities in the same physical scale in those tables as the y-axis in Figures 1, 3 and S6.

Thanks – we agree – now done, as described above. In addition to the information provided by Figures 1, 3 and S6, we now also **provide the boxplots and age-bin curves for all 10+10 top IDPs** as additional Supplementary Material.

d. Lack of model validation and model quality check. For example, some indices should be provided to assess the fitness of each model in general as well as for each brain measure. In addition, regardless of the modeling assumptions, the fitted patterns should be plotted against the raw data in, for example, Figures 1, 3 and S6 so that their quality can be visually verified.

Thanks – indeed, as described above for *point 1.b.*, we have now added several new Supplementary Analyses (including residuals vs model fit, and residual QQ plots) and now include all data-point-plotting and boxplots, as well as age-bin curves, for all top 10+10 IDPs.

- 2. The interpretation about the baseline differences (Tables S3 and S4) as well as those 7,545 non-imaging phenotypes (page 6) between patients and controls is problematic and unconvincing. The correction for multiple comparisons might be reasonable when one rejects the null hypothesis. However, it becomes very problematic when it is used for accepting the null hypothesis. It is well-known that the lack of strong statistical evidence does not mean a strong evidence for null effect. That is, one cannot make a strong statement when a null hypothesis is not rejected (especially when correction for multiple comparisons is applied). Some of the individual differences (e.g. the column of original p-value in Tables S3 and S4) do point to some extent of baseline differences.*

Thanks - we agree with these concerns, and have now added several deeper and “more aggressive” evaluations of potential baseline effects, both in imaging and non-imaging variables. Please see our detailed response above to the similar questions raised by Reviewer 1 (*point 4.*), and our new Supplementary Analyses. In short, we have now also **evaluated both univariate and dimensionally-reduced versions of baseline effects**, to avoid sensitivity-loss associated with multiple comparisons, and have **more aggressively and directly tested for whether any baseline effects could “explain away” the primary longitudinal effects**. These results strengthen our assessment of baseline differences, and increase confidence that the observed effects in patients with COVID-19 are not a result of such differences.

Furthermore, since the current curve-fitting in the models needs justifications, the authors should show/plot the original data at the baseline (similar to Figures 1, 3, and S6).

We have now added as Supplementary Material the boxplots and age-bin curves for the top 10+10 significant results. In addition, we provide the **full list of baseline imaging results** (using both binary and, separately, age-modulated regressors), referred to in the Results section:

Additional cross-sectional imaging results: When looking at binary baseline differences between controls and future cases, none of the IDPs with significant longitudinal effects for either hypothesis-driven or exploratory approaches demonstrated significant differences at baseline between the two groups (lowest $P_{\text{fwe}}=0.59$, nothing surviving FDR correction; **Table S2**). When applying age-modulation in the two-group modelling of IDPs at baseline, a few of the IDPs demonstrated significant differences between control and future SARS-CoV-2 groups, mainly for diffusion indices in the olfactory functional networks, as well as in the subcortical grey matter. As some IDPs cover spatially extended regions of the brain, we visually explored whether these baseline differences had any spatial overlap with our longitudinal results, but found none (**Supplementary Analyses** and **Figure S7**). The full list of (binary and age-modulated) results from group comparisons between the two groups at baseline, and separately, at the second timepoint are available in **Tables S2 and S3**.

In addition, we have now also **created a figure showing the voxel-by-voxel baseline differences** in grey matter volume, finding that the effects were confined to the subcortical structures, and we also show the sparse clusters of baseline differences in diffusion imaging (mean diffusivity):

Figure S7. Voxel-wise, cross-sectional baseline group differences between future infected participants and controls in grey matter volume and mean diffusivity (age-modulated). Top row. The thresholded map ($|Z| > 3$) shows that the strongest, localised lower grey matter volume at baseline in the future 401 SARS-CoV-2 positive participants compared with the 384 controls are bilaterally in the subcortical structures, more specifically the caudate nucleus, putamen, ventral striatum, thalamus and hippocampus, and in the brainstem. None of these regions spatially overlap with the main longitudinal results. **Bottom row.** The thresholded map ($|Z| > 3$) of mean diffusivity shows very few, scattered clusters of baseline differences (green) not overlapping with those longitudinal differences between the two groups (orange). We show the voxel-wise cross-sectional effects for illustrative purposes, avoiding any thresholding based on significance (as this would be statistically circular).

3. *The analytical approach lacks consistency. Four slightly different models were adopted to compare the four groups (controls, patients, non-hospitalised patients, and hospitalised patients). Each of the four models were applied to two sets of brain measures: one hypothesis-driven (with about 300 measures) and the other exploratory (with above 2,000 measures). In addition to these eight analyses with the imaging data, some of the models were further applied to various behavioral measures, risk factors, correlation analyses, etc. It is difficult to track the total number of analyses performed in this study. Even though the correction for multiple comparisons was used within each analysis, no similar step was taken to make statistical adjustment across those tens of analyses.*

Yes - we should have been clearer about the distinctions between the models; the presentation of the 4 different group-comparison models (the primary one and the secondary 3) in the main tables may have contributed to this. Model 1 (the main modelling of all cases vs controls) is the primary analysis - the results from the main question in our study. Models 2-4 are very much secondary analyses and never intended to be primary evaluations. It is not accepted practice to apply multiple comparison correction across primary *and* follow-up/secondary modelling; indeed, Reviewer 1 states that “*The statistical analyses use appropriate methods to control for multiple comparisons*”. Similarly, the hypothesis-based evaluations test an *a priori*-defined set of IDPs that is a *subset* of the full set, and not a separate set of IDPs; hence, we are following standard statistical practice here. However, we agree that clearly labelling tests as “primary” and “secondary” can help the general reader focus on the most important evaluations. We thus have **now clarified throughout the manuscript that Model 1 corresponds to our primary hypothesis, and Models 2-4 are secondary follow-up analyses.**

To avoid this pitfall, the authors could have pre-registered all the analytical details. On the other hand, a short list of analyses could be adopted to provide a much cleaner and more focused Presentation.

Thank you for this excellent suggestion: we **have introduced a list of the longitudinal analyses carried out in our study:**

Methods: We thus carried out 8 imaging group comparison longitudinal analyses:

- the primary analysis comparing all cases vs all controls (Model 1), first in the set of olfactory-related IDPs *a priori* drawn, then in the exploratory set of IDPs
- secondary, ancillary analyses, using both hypothesis-driven and exploratory sets of IDPs:
 - all non-hospitalised cases vs all controls (Model 2),
 - all hospitalised cases vs all controls (Model 3),
 - all hospitalised cases vs all non-hospitalised cases (Model 4).

4. *Heavy reliance on statistical inference (e.g., lack of effect quantification, interpretation difficulty) can be perilous. With large sample sizes, small or even tiny effects could reach statistical significance. Similarly, even when an explanatory variable (e.g., using the trajectory of death rate for brain changes) is not well-justified in the context, results solely based on statistical significance can be misleading. In addition, there are some procedural inconsistencies and misinterpretations of the statistical results.*

Thanks - again, for increased clarity, we now **report % change and standard errors throughout**, as discussed in detail above (Reviewer 1 point 2 and Reviewer 4 point 1.c).

a. With a model applied thousands of times to various measures, the issue of multiple comparisons is not the only major concern. Nevertheless, if one is concerned with the multiple comparisons issue, then corrections should be not limited to each analysis/model; instead, it should be equally applied to tens of different models/analyses.

As discussed above, this is not a concern here - we fully correct for multiple comparisons in our main model, and separately correct for multiple comparisons within secondary follow-up analyses.

b. All the UKBank data were used for normalisation for those 785 subjects, but only those 785 subjects were used for normalisation in Figure 3.

Thanks - that is correct; the difference is very small (the mean of the raw IDP values across the 785 subjects from Scan 1, compared with mean across ~40,000 subjects, correlates at $r=0.99997$, and the *fractional* mean absolute difference is ~0.01), but to avoid any confusion, **all normalisation by these means is now done using the 785 subjects.**

c. The specific correction adopted for multiple comparisons appears to be arbitrary and inconsistent. Sometimes both FWE and FDR are mentioned in the text while in all tables only FWE results are shown.

Apologies if we were not clear: for any given model, we first computed the uncorrected p-values; we then computed both FDR- and FWE-corrected inferences as two distinct measures of strength of evidence for an effect while accounting for multiple testing. Primarily we rely on FDR correction, which provides good power while controlling for multiple testing in a principled manner, but when a result additionally attains FWE significance, we wish to indicate this. Thus, in the Results section and the main Tables, we always specify the findings obtained using both correction methods.

5. Minor issues

a. The word “network” in the paper should be clarified or replaced. A network is supposed to involve a number of brain regions. However, the usage in the paper is apparently different (e.g. Figure 1).

Thanks - it is true that “network” can be used with a wide variety of meanings, but indeed would generally be taken to mean a set of not necessarily contiguous regions in the brain imaging literature. The images (maps) generated through the estimation of functional connectivity with the primary olfactory cortical regions (one set of the new custom IDPs identified for this work) are indeed “functional networks” defined through connectivity, and all contain multiple disconnected regions; this terminology is the accepted and understood one in the brain imaging community. However, we have checked carefully through the paper to make sure that no confusing/inconsistent usages of “network” remain.

b. The usage of Z-statistics in the manuscript seems inconsistent/ambiguous. It is used to refer to the Fisher-transformed value (page 31) which is usually called Z-score, but the Z-statistics in other places of the text as well as in table 4 and Figure S7 is likely different.

It is common statistical practice to refer to the normalisation of *data* (demeaning to zero mean, and scaling to unity standard deviation) as transforming data into Z-scores. Although there are various stages where data normalisation is carried out (as described in Methods), we do not use either term (“Z-score” or “Z-statistics”) when describing/discussing this data preprocessing in the paper.

With respect to model-fitting output test statistics (e.g., r , T , Z , P), it is true that some of the literature refers to (model test statistic) Z as “Z-statistics” and some literature as “Z-scores”, but, because of our points above, we prefer the less ambiguous term “Z-statistics”, and this is following the majority of the literature. This includes when applying the Fisher transformation to turn r (correlation coefficient test statistics) into Z-statistics, such as in Table 4 and Figure S7. We have now made sure we **consistently refer to “Z-statistics” throughout the manuscript**, which we feel is the right approach, and our main modelling now uses regression instead, so **we focus on % change** as described above.

c. It is misleading to label the study as longitudinal. The longitudinal aspect is only limited to the two time points (~3 years apart). In fact, this study remains largely cross-sectional because the age trajectory is basically captured with different subjects within the age range of 51-81.

We respectfully disagree - the primary measure at the heart of our modelling is the longitudinal change, i.e. the Δ IDP between the pre-pandemic imaging scan and the post-infection (for cases) scan. We then have random-effects analyses across subjects to compare this Δ IDP effect between groups. The fact that the core measure is a two-timepoint longitudinal change is a major factor that differentiates this work from purely-cross sectional post-infection-only studies such as the UK’s CMORE, PHOSP and COVID-CNS studies.

d. For clarity, all the models should be explicitly expressed with symbols and subscripts representing various factors and as well as all the terms including intercept and confounding effects. Explain the detail and purpose of demeaning in model (1).

We have now changed the model using symbols and subscripts, and explained why we demeaned the model:

Methods: The main case-vs-control group difference regressor of interest g_m is therefore formed using the demeaned binary two-group variable g and the age at Scan 2 a_2 :

$$g_m = g \times 10^{a_2 \times 0.0524 - 3.27} , \quad [1]$$

where the age-dependence constants are taken from the meta-regression analysis⁷⁸ (see **Supplementary Analyses** for further details on age-modulation choices, and discussion of non-modulated modelling results). To ensure that the fitting of this term is not influenced by an effect that is common to controls and cases, we added a matching confound variable of $g_0 = 10^{a_2 \times 0.0524 - 3.27}$, i.e., the same ageing term without the group-difference multiplier g .

Our main model of interest therefore simply combines IDP at Scan 2 (I_2), IDP at Scan 1 (I_1), the above group-difference model, and confounds matrix c :

$$I_2 \sim g_m + I_1 + c , \quad [2]$$

where the confounds matrix comprises the terms described above: $a_2 - a_1$, $a_2^2 - a_1^2$, ethnicity, sex, and g_0 (with a_1 being age at Scan 1).

The group-difference regressor is scaled to have average peak-peak height 1, so that the regression parameter from fitting g_m can easily be converted into a percentage change measure, when normalised by the mean baseline value for a given IDP. For the main longitudinal modelling, this represents the average group difference in the longitudinal change, and for the separate modelling of baseline IDPs only, this percentage reflects the average group difference in the baseline values.

e. For better reference and communication during the reviewing process, add line numbers and page numbers.

Done.

f. Provide uncertainty information. In all tables, consider adding a column for standard error or confidence interval, which is much more informative and useful than those statistical numbers and various p-values.

Done (see also Reviewer 1 point 2.).

g. In those tables for modeling results of group differences, provide the same information (effect magnitude and uncertainty) for each group so that the group differences can be assessed with proper references. In addition, the information at the baseline and the second scan should be added.

The model is a regression model including various confound factors and the group effect regressor (and not, for example, a two-group t-test). Hence there is indeed a group-

difference effect (for convenience of interpretation reported as a % change and standard error on this), but there are not separate parameters and uncertainties estimated for the two groups separately within this modelling. However, as discussed above, we now report the effect size and uncertainty for every IDP and every model. **Additional supplementary tables cover the cross-sectional modelling**, with full details (including % effect).

h. The criteria for outlier handling are arbitrary and inconsistent. Sometimes it was 8 times the median absolute deviation, while sometimes it was 15 times. What are the causes for those outliers?

Thanks - this is correct; for IDPs, the outlier threshold was 8 MAD, and for confounds it was 15. The only reason for raising the threshold for confounds is that some have extremely non-Gaussian underlying distributions (e.g., table position), and we found that a threshold of 8 was too aggressive for these variables (for values that are perfectly acceptable when considered with the domain knowledge of these variables; Alfaro-Almagro et al., 2020). We have now **expanded our description of these issues in Methods**:

Methods: These imaging confound variables first had outlier removal applied as described above, though using a higher threshold of 15 times the median absolute deviation, because some important confounds have extremely non-Gaussian underlying distributions (e.g., MRI scanner table position), and we found that a threshold of 8 was too aggressive for these variables, for values that are perfectly acceptable when considered with the domain knowledge of these variables^{63,74}.

Also, can those missing data be considered as missing at random?

As discussed in response to Reviewer 1 point 9., we have **no evidence of missingness happening not-at-random**.

i. The residualisation step for Eq (2) is troubling. First, it may create some biases in effect estimation. Second, the factor "N-3" adopted in the Fisher transformation is not accurate because of the residualisation process.

Thanks. The previously-done residualisation step (for the correlation-based analysis) does not create bias, being mathematically equivalent to the combined multiple regression approach when correcting the degrees-of-freedom as discussed in detail above; however, in any case, following the same point raised by Reviewer 1 (point 11.a.), we have now switched to the equivalent regression approach, which corrects the issue of degrees-of-freedom rightly raised by Reviewers 1 and 4.

j. Task fMRI data were mentioned in the paper, but what IDPs were obtained for such data?

Thanks - yes we had mentioned task-fMRI being included in the scanning protocol, but had missed out including a sentence describing that we have not considered task-fMRI IDPs in this study. Please see response to Reviewer 1 point 9.: we have now included **more details on these specifics in the revision**. It is worth mentioning that had we included task-fMRI IDPs in the work, they would have then been automatically excluded anyway by the reproducibility quality thresholding preprocessing step.

k. Add a color bar in Figure 2 so that the color maps could be properly interpreted. What is the rationale for the threshold of 3 for Z-values?

Done. With respect to the thresholding at $|Z| > 3$, this is common practice in brain imaging, in order to present results corresponding in effect to $P_{\text{uncorr}} < 0.001$. Here, we show the voxel-wise or vertex-wise longitudinal effects for *illustrative* purposes, intentionally avoiding any thresholding based on significance, as this would be statistically circular and potentially misleading (this issue is similar to what we have done and highlighted previously in Elliot et al., Nature 2018). We also have now **made this clear in the caption:**

Figure 2: (...) We show the voxel-wise or vertex-wise longitudinal effects for illustrative purposes, avoiding any thresholding based on significance (as this would be statistically circular - similar to our previous analyses reported in ³⁰).

l. Explain the post hoc correlation analysis for the cognitive data (bottom of page 33).

Thanks - this is now clarified further in Methods.

m. Axis labels, legend symbols and legend labels in some figures (for example, Figures 3 and S5) are too small.

Thank you - we have now fixed this.

Reviewer Reports on the First Revision:

Referees' comments:

Referee #1 (Remarks to the Author):

The authors have fully, appropriately and elegantly addressed all concerns. I have no further comments.

Referee #2 (Remarks to the Author):

Review of revised version of the manuscript "Brain imaging before and after COVID-19 in UK Biobank by Douaud et al.

I do appreciate the substantial efforts of the authors to improve the study, especially in relation to the aspects I regard(ed) as problematic. I am aware of the limitations provided by the data stored in the the UK Biobank – which, if not there, cannot be retrieved, obviously - and I am equally aware that this is mainly an imaging study; nevertheless one must be careful to not overstretch these data (and its correlation(s)) to yield information that is difficult to retrieve based from what is there. To state it clearly: I am not opposing to recommend publishing this piece of (intense) work as long as the limitations of this study are stated explicitly – something that is more obvious in the revised version. Along that line, the discussion appears to be more balanced insofar as the overall small effects are discussed in light the high n of subjects investigated and the mentioning that there are many asymptomatic or mild cases within the in der Covid-19 cohort.

However, I still have comments/critiques, which need to be addressed:

1. On page 9, the authors now write: "For those significant IDPs, average percentage change differences between the two groups was moderate, ranging from ~0.2 to~2%, with the largest differences seen in the volume of the parahippocampal gyrus and entorhinal cortex (Table S1)." This difference appears to be minute – which probably only due to the high n becomes statistical significant. Nevertheless, the question of true biological relevance of such a small difference remains (when focusing on the individual case as one does, after all, in a clinical setting).

2. Further, on page 9: "Finally, we found no significant differences between the 15 hospitalised patients and 386 non-hospitalised SARS-COV-2 cases, likely due to the large reduction in sample size, but effect sizes suggested similar effects once more in the orbitofrontal, insula, parahippocampal and frontal piriform cortex functionally connected brain regions (all $|Z| \geq 3$, Model 4, Table S1)." Shouldn't one expect a stronger impact in these selected severe Covid-19 cases with a detrimental disease course (as opposed to mild cases, which obviously will not be stored in the UK Biobank, since they all survived Covid-19)?

3. I am still struggling with this part on page 11: "While significant IDPs related to grey matter thickness were found, using our main case-vs control analysis (Model 1), to be bilateral for both the anterior parahippocampal gyrus (perirhinal cortex) and entorhinal cortex, 10 of the 11 remaining

significant IDP were leftlateralised (Table S1). We thus directly investigated (left - right) differences in the SARS-CoV-2 group only for those significant IDPs, and found that the infected participants did not have significantly more reduced grey matter thickness on the left than on the right hemisphere (lowest Puncorr=0.30). None of the top 10 most significant results showed a significant effect of interval between infection and second scan in the SARS-CoV-2 positive participants for whom we had a date of diagnosis (n=351; lowest Puncorr=0.08).” Does this mean that probing the model by doing the second scan abolishes the initial data re. the affection of anatomical distinct regions? This needs to be explained again.

4. Page 18: „In addition, none of the 10 cognitive variables showed significant difference at baseline between SARS-CoV-2 and control groups (min Puncorr=0.08). With age-modulation, only one cognitive score, time to complete Pairs Matching round, showed a trend difference at baseline (Puncorr<0.05, P_{fwe}=0.29, not passing FDR), while the longitudinal cognitive results were observed in the UK Biobank Trail Making Test.“ This part – to my understanding – proves that changes in the olfactory cortex do not go along with changes in the parahippocampal cortex (which is crucial for learning paradigms), and thus no alteration on a functional level can be concluded (skipping the problematic topic of extrapolating from form to function).

5. On page 27: “While the UK Biobank version of the Trail Making Test is carried out online and unsupervised, there is good to very good agreement with the standard paper-and-pencil Trail Making Test on its measurements for completion of the two trails, two measures known to be sensitive to detect, and discriminate, mild cognitive impairment and dementia from healthy ageing. In turn, the duration to complete the alphanumeric trail B was associated post hoc with the longitudinal changes in the cognitive part of the cerebellum, namely crus II, which is also specifically activated by olfactory tasks. In line with this result, this particular part of the cerebellum has been recently shown to play a key role in the association with (and prediction of future) cognitive impairment in patients with stroke (subarachnoid haemorrhage)*. It remains to be determined whether the loss of grey matter and increased tissue damage seen in memory related regions of the brain may in turn increase the risk for these participants of developing dementia in the longer term.” This is a too narrow interpretation, as subjects suffering from depression or schizophrenia also show a reduced performance in the Trail Making Test <https://doi.org/10.1080/00207450601059452>; Age effects on trail making test during acute depressive and manic episode; <https://doi.org/10.1080/13854040590947498>: Trail making test errors and executive function in schizophrenia and depression). In these settings, executions of tasks within a given time is crucial, and not the cumulative time per se. Since the parahippocampus is the brain region for mnemonic capacity (do I learn 4 or 7 out of 10 words?), this type of interpretation in the manuscript is still incompatible. This needs, at least, to be rephrased. Without any doubt, including the pneumonia group adds a lot, but putting this in context one could assume that the reported effects are not Covid-19 specific, but rather a common phenomenon.

Referee #4 (Remarks to the Author):

I appreciate that the authors provided detailed responses to my comments. Despite their great effort, the major problems raised during the previous round were largely not properly addressed.

The quality of their modeling work is disappointing and does not match the richness this dataset offers. I'm sorry but I cannot accept their revision as a valid reply to my comments.

1. The biggest problem remains that their model (2) lacks adequate justification.

a) No evidence is provided that age trajectory would follow an exponential pattern for any of the IDPs. The two citations the authors provided are regarding either the death rate of COVID-19 patients with older people (Levin et al, 2020) or COVID-19 hospitalization rate (Palmer et al., 2021). None of them are related to any IDPs.

b) Even if an exponential function can be a reasonable approximation for IDPs, why would the parameters values of 0.0524 and -3.26 in the expression $10^{(\text{Age} \times 0.0524 - 3.27)}$ be a good fit for all IDPs? The authors seem to prefer to choose a model “with no free or subjectively-chosen parameters—hence having the same degrees-of-freedom fitting power as a binary model”. Why is “the same degrees-of-freedom fitting” an important issue? Different IDPs have different patterns of missing data anyway.

c) If the expressions g_m and g_0 are supposed to fully capture the age effect, why do the authors need other age terms such as $a_2 - a_1$, $a_2^2 - a_1^2$ in the model (2)? Also, the parameter g in the formula (1) is “the demeaned binary two-group variable” - what does this mean? What are the values of g ?

d) Other confounds discussed on lines 22-24 page 34 should be added in model (2) instead of adopting a separate step during data preprocessing.

e) They offered the following argument as their defense for the adoption of model (2): “as a general rule, if a model was suboptimal, the main result would be a loss of sensitivity, not an increase in false positives.” This is not a legitimate argument. With their large data sample size, many candidates would likely be able to fit the category of such suboptimal models and show strong statistical evidence. Why adopt a particular exponent pattern with fixed parameters that cannot be justified? Furthermore, the authors did not really provide any model validation for the quality of their model. In addition, the model fits are not even plotted in any of the relevant figures (Figures 1, 3, S6) to show their fit quality. It might be true that with large sample size, many models (including the authors’ model (2)) could provide small p-values for those IDPs shown in the current figures, but this does not mean that one could pick up a convenient model without substantiation. More importantly, many IDPs might have been missed simply because of the unsubstantiated model. The information provided in the supplemental material (pages 12-13) does not address this problem.

f) The expression in (2) is not a formal presentation for a statistical model. Consider a more rigorous formulation.

2. Baseline differences are still a sticky point. The authors have so far compared the average IDPs and some principal components between the two groups. However, their major research interest is not about average IDP values or principal components. If the main focus in the current paper regards the age pattern differences between the two groups, it would be more appropriate to address the

issue of potential differences in age pattern at the baseline for all relevant IDPs (not just those “top 10”). For example, properly model the age patterns at the baseline and assess the cross-group differences.

3. Conceptual issue. The word “significant” (and significantly) has been used many times throughout the paper. The reader could interpret it to refer to the amplitude of the effect under discussion or the statistical strength. It is unclear what the authors meant in all these cases considering the amplitude “ranging from 0.2 to 2% for the vast majority of the significant longitudinal results” given that their model is reasonable. Without clarification, the usage can lead to misunderstanding and confusion.

4. As for the choice of correction method for multiple comparisons, previously I raised the issue of the arbitrariness between FWE and FDR. They responded with the following: “Primarily we rely on FDR correction, which provides good power while controlling for multiple testing in a principled manner, but when a result additionally attains FWE significance, we wish to indicate this.” I could understand if one decided on one before the project and stuck with it, but is it problematic to present the results based on which of the two rendered more preferable results? If FDR is the primary choice, were all the permutations performed with FDR as a metric at the end?

Minor issues

** Outliers were misclassified as part of missing data. First, outliers are not missing data. The IDPs that were deemed as not reproducible were also treated as outliers (lines 9-10, page 35). What is the portion of the outlier IDPs? The arbitrariness of thresholding for outliers remains. If “a threshold of 8 was too aggressive”, why not use the same threshold of 15 for all scenarios? In addition, state the sources for those outliers.

Line 14, page 33: “using the olfactory bulb and hypothalamus maps (unthresholded and thresholded at 0.3)” - Why a threshold of 0.3 is adopted? And why two separate volume extraction methods?

Lines 22-24, page 34: “cross-sectional deconfounding... was carried out for head size, scanner table position, and image motion in the diffusion MRI data.” Instead of a separate modeling step that may cause biases, would it be better to add these confounds to your main model (2)?

Author Rebuttals to First Revision:

Reviewer #2:

Review of revised version of the manuscript “Brain imaging before and after COVID-19 in UK Biobank by Douaud et al.

I do appreciate the substantial efforts of the authors to improve the study, especially in relation to the aspects I regard(ed) as problematic. I am aware of the limitations provided by the data stored in the the UK Biobank – which, if not there, cannot be retrieved, obviously - and I am equally aware that this is mainly an imaging study; nevertheless one must be careful to not overstretch these data (and its correlation(s)) to yield information that is difficult to retrieve based from what is there. To state it clearly: I am not opposing to recommend publishing this piece of (intense) work as long as the limitations of this study are stated explicitly – something that is more obvious in the revised version. Along that line, the discussion appears to be more balanced insofar as the overall small effects are discussed in light the high n of subjects investigated and the mentioning that there are many asymptomatic or mild cases within the in der Covid-19 cohort.

We would like to thank Reviewer 2 for their appreciation of our efforts in tackling their concerns, and for acknowledging that the revised manuscript has been improved as a consequence.

However, I still have comments/critiques, which need to be addressed:

1. On page 9, the authors now write: “For those significant IDPs, average percentage change differences between the two groups was moderate, ranging from ~0.2 to ~2%, with the largest differences seen in the volume of the parahippocampal gyrus and entorhinal cortex (Table S1).” This difference appears to be by minute – which probably only due to the high n becomes statistical significant. Nevertheless, the question of true biological relevance of such a small difference remains (when focusing on the individual case as one does, after all, in a clinical setting).

Reviewer 2 is right: this study certainly benefits from a relatively large N, for which differences with small effect sizes can be significant. We would like to note however that an additional loss in the infected participants of 0.7% on average across the olfactory-related brain regions — and specifically ranging from 1.3% to 1.8% for the FreeSurfer volume of the parahippocampal/perirhinal and entorhinal cortex — is not insubstantial, considering for instance that the hippocampus loses “only” 0.2% (in middle age) to 0.3% (in older age) of volume per year (Fraser et al., 2021). We have now added a comment, as well as this reference, in the Discussion when mentioning the effect sizes:

Discussion: It is worth noting that these structural and microstructural longitudinal significant differences are modest in size, the strongest differences in changes observed between the SARS-CoV-2 positive and control groups corresponding to around 2% of mean baseline IDP value (**Extended Data Table 1**). This additional loss in the infected participants of 0.7% on average across the olfactory-related brain regions — and specifically ranging from 1.3% to 1.8% for the FreeSurfer volume of the parahippocampal/perirhinal and entorhinal cortex — can be helpfully compared with, for instance, the longitudinal loss per year of ~0.2% (in middle age) to 0.3% (in older age) of hippocampal volume in community-dwelling individuals²⁹.

2. Further, on page 9: “Finally, we found no significant differences between the 15 hospitalised patients and 386 non-hospitalised SARS-COV-2 cases, likely due to the large reduction in sample size, but effect sizes suggested similar effects once more in the orbitofrontal, insula, parahippocampal and frontal piriform cortex functionally connected brain regions (all $|Z| \geq 3$, Model 4, Table S1).” Shouldn’t one expect a stronger impact in these selected severe Covid-19 cases with a detrimental disease course (as opposed to mild cases, which obviously will not be stored in the UK Biobank, since they all survived Covid-19)?

Absolutely: we do find a stronger impact when directly comparing the hospitalised with non-hospitalised cases. This is something that can be seen in Figure 2, bottom row, in which we show where hospitalised patients demonstrate a greater atrophy over time than the mild cases ($Z \geq 3$, regardless of significance, since, as indicated, lower N did not yield any significant results). We realise that the sentence we used and that is quoted above is unclear, and have therefore rephrased it:

Hypothesis-driven results: Finally, we found no significant differences between the 15 hospitalised patients and 386 non-hospitalised SARS-COV-2 cases, likely due to the large reduction in sample size, but effect sizes and direction of these effects suggested stronger detrimental effects for the hospitalised cases in the orbitofrontal, insula, parahippocampal and frontal piriform cortex functionally-connected brain regions (all $|Z| \geq 3$, Model 4, **Extended Data Table 1**).

3. I am still struggling with this part on page 11: “While significant IDPs related to grey matter thickness were found, using our main case-vs control analysis (Model 1), to be bilateral for both the anterior parahippocampal gyrus (perirhinal cortex) and entorhinal cortex, 10 of the 11 remaining significant IDP were left lateralised (Table S1). We thus directly investigated (left - right) differences in the SARS-CoV- 2 group only for those significant IDPs, and found that the infected participants did not have significantly more reduced grey matter thickness on the left than on the right hemisphere (lowest Puncorr=0.30). None of the top 10 most significant results showed a significant effect of interval between infection and second scan in the SARS-CoV-2 positive participants for whom we had a date of diagnosis (n=351; lowest Puncorr=0.08).” Does this mean that probing the model by doing the second scan abolishes the initial data re. the affection of anatomical distinct regions? This needs to be explained again.

Presumably Reviewer 2 here is confused specifically about the last sentence of the quote, i.e. “None of the top 10 most significant results showed a significant effect of interval between infection and second scan in the SARS-CoV-2 positive participants for whom we had a date of diagnosis”. Our main model shows that there are significant differences between infected and non-infected participants in the changes between second scan (during the pandemic) and first scan (pre-pandemic). As a separate, corollary analysis in the SARS-CoV-2 group only, we interrogate whether, for these regions that show the most divergence over time between the two groups, there is also show some correlation between how long the participants have been infected (i.e., date of second scan minus date of recorded infection) and how much these brain areas have changed. These changes could show an improvement over time, suggesting that the effects of infection are normalising; they could show a deterioration, denoting perhaps an ongoing degenerative process; or they could be a mixture of the two or stay stable, in which instance there would

be no correlation. In our study, we don't see any significant correlation. We now slightly rephrase this part of the manuscript section to clarify these results:

Hypothesis-driven results: Of the top 10 IDPs showing a longitudinal effect between first and second scans, none correlated significantly with the time interval between their infection and their second scan, in the SARS-CoV-2 positive participants for whom we had a date of diagnosis (n=351; lowest Puncorr=0.08).

4. Page 18: „In addition, none of the 10 cognitive variables showed significant difference at baseline between SARS-CoV-2 and control groups (min Puncorr=0.08). With age-modulation, only one cognitive score, time to complete Pairs Matching round, showed a trend difference at baseline (Puncorr<0.05, P_{fwe}=0.29, not passing FDR), while the longitudinal cognitive results were observed in the UK Biobank Trail Making Test.“ This part – to my understanding – proves that changes in the olfactory cortex do not go along with changes in the parahippocampal cortex (which is crucial for learning paradigms), and thus no alteration on a functional level can be concluded (skipping the problematic topic of extrapolating from form to function).

Some of our wording might have been ambiguous, so we have now clearly stated in the Discussion that no alteration associated with SARS-CoV-2 infection on a functional level could be seen in the parahippocampal gyrus:

Discussion: On the other hand, the parahippocampal gyrus and other memory-related regions did not show in our study any alteration on a functional level, i.e., any *post hoc* association with the selected cognitive tests.

And we have now also slightly rephrased the quoted passage above to make it clearer:

Additional baseline investigations: In addition, none of the 10 pre-selected cognitive variables showed significant difference at baseline between SARS-CoV-2 and control groups (min P_{uncorr}=0.08). With age-modulation, only one cognitive score, time to complete Pairs Matching round, showed a trend difference at baseline (P_{uncorr}<0.05, P_{fwe}=0.29, not passing FDR). This is a different cognitive score from the one showing longitudinal cognitive effects between the two groups, the UK Biobank Trail Making Test.

5. On page 27: “While the UK Biobank version of the Trail Making Test is carried out online and unsupervised, there is good to very good agreement with the standard paper-and-pencil Trail Making Test on its measurements for completion of the two trails, two measures known to be sensitive to detect, and discriminate, mild cognitive impairment and dementia from healthy ageing. In turn, the duration to complete the alphanumeric trail B was associated *post hoc* with the longitudinal changes in the cognitive part of the cerebellum, namely crus II, which is also specifically activated by olfactory tasks. In line with this result, this particular part of the cerebellum has been recently shown to play a key role in the association with (and prediction of future) cognitive impairment in patients with stroke (subarachnoid haemorrhage). It remains to be determined whether the loss of grey matter and increased tissue damage seen in memory related regions of the brain may in turn increase the risk for these participants of developing dementia in the longer term.” This is a too narrow interpretation, as subjects suffering from depression or

schizophrenia also show a reduced performance in the Trail Making Test <https://doi.org/10.1080/00207450601059452>; Age effects on trail making test during acute depressive and manic episode; <https://doi.org/10.1080/13854040590947498>: Trail making test errors and executive function in schizophrenia and depression). In these settings, executions of tasks within a given time is crucial, and not the cumulative time per se. Since the parahippocampus is the brain region for mnemonic capacity (do I learn 4 or 7 out of 10 words?), this type of interpretation in the manuscript is still incompatible. This needs, at least, to be rephrased. Without any doubt, including the pneumonia group adds a lot, but putting this in context one could assume that the reported effects are not Covid-19 specific, but rather a common phenomenon.

In line with Reviewer 2's previous comment, we have rephrased this part of the Discussion to highlight the points that this reviewer raises, and added the corresponding references:

Discussion: While the UK Biobank version of the Trail Making Test is carried out online and unsupervised, there is good to very good agreement with the standard paper-and-pencil Trail Making Test on its measurements for completion of the two trails⁴⁶, two measures known to be sensitive to detect impairment of executive function and attention, for instance in affective disorders and in schizophrenia^{47,48}, and to discriminate mild cognitive impairment and dementia from healthy ageing⁴⁹. In turn, the duration to complete the alphanumeric trail B was associated *post hoc* with the longitudinal changes in the cognitive part of the cerebellum, namely crus II, which is also specifically activated by olfactory tasks^{38,50}. In line with this result, this particular part of the cerebellum has been recently shown to play a key role in the association with (and prediction of future) cognitive impairment in patients with stroke (subarachnoid haemorrhage)⁵¹. On the other hand, the parahippocampal gyrus and other memory-related regions did not show in our study any alteration on a functional level, i.e., any *post hoc* association with the selected cognitive tests. It remains to be determined whether the loss of grey matter and increased tissue damage seen in these specific limbic regions may in turn increase the risk for these participants of developing memory problems, and perhaps dementia in the longer term^{2,4,52}.

References: 47 Mahlberg, R., Adli, M., Bschor, T. & Kienast, T. Age effects on trail making test during acute depressive and manic episode. *Int J Neurosci* 118, 1347-1356, doi:10.1080/00207450601059452 (2008).

48 Mahurin, R. K. et al. Trail making test errors and executive function in schizophrenia and depression. *Clin Neuropsychol* 20, 271-288, doi:10.1080/13854040590947498 (2006).

We include below our responses to R4's review of our resubmission. To help with the readability of this rebuttal, we have pasted relevant text from our previous *Response-to-reviews and Revised-manuscript in orange italics*.

Referee #4 (Remarks to the Author):

I appreciate that the authors provided detailed responses to my comments. Despite their great effort, the major problems raised during the previous round were largely not properly addressed. The quality of their modeling work is disappointing and does not match the richness this dataset offers. I'm sorry but I cannot accept their revision as a valid reply to my comments.

1. The biggest problem remains that their model (2) lacks adequate justification.

With respect, we strongly disagree that our Response/Revision did not contain adequate justification - we justified our model in the context of relevant studies, we included additional (similar) results using a non-age-modulated model, and we discussed statistical issues around model fitting.

We now discuss these 3 points in a little more detail:

1. Our Response argued in detail why it is reasonable *a priori* to take age into account when modelling COVID-19 disease effect, given the huge amount of epidemiological data showing that both disease severity and outcome severity increase with age. We described several independent, published age-dependent models, all of which found a similar form to the one we used in our primary analyses. This includes a recent study considering disease severity (Palmer), which (as described in our Response) gives significant associations for the IDPs we identified with our original model.

Response p10: Reviewer 1 is right: this age modulation factor accounts for more pronounced effects in older people. We chose to focus on an "objective" age model given the strong prior knowledge of a highly increased detrimental effect, at older ages, of SARS-CoV-2 infection and a greater vulnerability of the brain with age. We therefore used the only already-published data-driven curve of age-effect in SARS-CoV-2 infection we could find at the time, with no free or subjectively-chosen parameters—hence having the same degrees-of-freedom fitting power as a binary model. (Since then, we have identified a separate set of data for age-dependent mortality rates, from the UK Office of National Statistics; this has given an almost identical exponential age dependence: males: 0.0451 ± 0.0008 (units 1/years), females: 0.0475 ± 0.0005 , very close to the value of 0.0524 in the Levin study.) In another analysis, of hospitalisations due to COVID-19, the function of age has also been found to be exponential in form, with exponent dependence ~ 0.02 (Palmer et al. (2021). J R Soc Interface 18(176)). If we use this age modulation instead, extremely similar results are found for the hypothesis-driven Z-statistics, $\text{corr}(Z)=0.96$, with the strongest original results ($|Z|>3.5$) increasing in strength by an average of $Z=0.16$. This is now addressed in more detail in Supplementary Analyses, and referred to in Methods.

Response p23: Again, we appreciate this concern, but, to briefly summarise what we have written above and the new results from Supplementary Analyses, we feel it is clear a priori that some form of age-dependence in the model is sensible, given the strong prior knowledge of a highly increased detrimental effect of COVID-19 at older ages, and a greater vulnerability of the brain with age.

2. An even simpler point is that **we also included full results from all tests carried out without the age modulation, i.e., using the binary (non-age-modulated) case-control group-difference regressor**. We have described in detail that most of the primary results remain significant. These supplemental results were described in detail in our Response, and provided in our Revision, both in the form of comprehensive additional Tables, and additional figures, but R4 seems to have ignored all of this.

Response p10: In the interest of completeness, we have now also tested—as a secondary analysis—the binary modelling between control and SARS-CoV-2 groups, with all other factors held the same. These two (age-modulated and non-modulated) models do not give very different primary results (hypothesis-driven results presented in Tables 1 and S1). The findings are highly similar, if a little weaker, consistent with our expectation of increased effects at higher ages. For instance, from the 10 most significant results reported in Table 1, 10 associations had passed FDR correction, and 6 passed FWE correction. When switching to the binary (non-modulated) group modelling, 9 out of 10 continue to pass FDR correction and 4 out of 6 pass FWE correction. The correlation between the Z-statistics from all 297 hypothesis-driven IDPs is high ($r=0.88$), and the mean change in $|Z|$ across the 10 top associations was a reduction of just 0.33. We have added results obtained with the binary regressor in all four Models for comparison in Supplementary Table S11.

Response p23: In addition, the results do not depend strongly on the exact choice of age-dependence made – even without any age modulation in the main model, many of the primary results remain. We have also discussed above two additional (external) estimates of the form of the age-modulation in COVID-19, and found that these give very similar results (all in Reviewer 1 point 6. and corresponding Supplementary Analyses).

3. With respect to model-fitting statistics, we include some more specific points below, but repeat here the core points from our previous Response:

- a. If our model was sub-optimal, “the main expected outcome would be that we could fail to find significant results, and not that there would be any inflation of false positives” (i.e., we would not see invalid results).

Response p10: Finally, we note that, as a general rule, if a model was suboptimal, the main result would be a loss of sensitivity, not an increase in false positives (see point 1.b. by Reviewer 4).

Response p24: We agree that the model may not be optimal for every feature (IDP) considered; in other words, this model might not be the most sensitive possible model for every IDP. However, where this is the case, the main expected outcome would be that we could fail to find significant results, and not that there would be any inflation of false positives. This is always the case when choosing any given model. Were we to use more flexible (less parsimonious) models, the use of F-tests with multiple (numerator) degrees of freedom would likely result in reduced power, and the use of active model selection would require a multiple testing correction that would again reduce power.

- b. Nevertheless, we have shown in many supplemental analyses, the validity of the model-fitting, including diagnostic residual scatter-plots and residual QQ plots (and indeed we also found exact convergence between parametric inference and permutation-based inference).

Response p24: Nevertheless, we have now also added new Supplementary Analyses, showing no obvious evidence of structured problems in model residuals. We created diagnostic plots from the top 10 most significant IDPs showing longitudinal differences between cases and controls (10 from the hypothesis-driven approach, 10 from the exploratory), with scatterplots of residuals against model-predictions, showing no evidence of model misspecification. We also include QQ plots of model residuals for the top 10+10 IDPs, again with no evidence of model-fitting problems. Finally, we assessed sensitivity to additional, higher-order, confound variables, with up to 3-way interactions, finding no large changes in the main results (all in Supplementary Analyses).

Response p25: Thanks – indeed, as described above for point 1.b., we have now added several new Supplementary Analyses (including residuals vs model fit, and residual QQ plots) and now include all data-point-plotting and boxplots, as well as age-bin curves, for all top 10+10 IDPs.

a) No evidence is provided that age trajectory would follow an exponential pattern for any of the IDPs. The two citations the authors provided are regarding either the death rate of COVID-19 patients with older people (Levin et al, 2020) or COVID-19 hospitalization rate (Palmer et al., 2021). None of them are related to any IDPs.

We have motivated a model form from the most relevant literature, and one where the interpretation of the model parameters is straightforward. We have then used this model, finding statistically significant and interpretable results. Noting our points above, by definition, **this is evidence that this model-fit has statistical significance (for the relevant IDPs), and this is the evidence for this age-trajectory.** Of course, our study conforms to the wisdom that “All models are wrong, but some are useful” - we have not tested all possible models for all possible IDPs; instead, we have focussed on one possible model drawn from independent, existing literature and found that is “useful” (i.e., statistically significant). Please see above (and the quoted texts from our Response above) for our further points on this.

b) Even if an exponential function can be a reasonable approximation for IDPs, why would the parameters values of 0.0524 and -3.26 in the expression $10^{(\text{Age} \times 0.0524 - 3.27)}$ be a good fit for all IDPs?

Again, we do not claim this is the very best possible model out of the infinity of possible models, nor that the “ideal” model (assuming one could avoid overfitting and/or power-loss as a result of searching across multiple models) would be identical for all IDPs. Rather, we have followed the standard scientific practice of proposing a simple model grounded in the relevant literature. We use valid and rigorous statistical inference to test for model parameter significance, and report IDPs that show a significant effect, while fully controlling for multiple tests.

The authors seem to prefer to choose a model “with no free or subjectively-chosen parameters—hence having the same degrees-of-freedom fitting power as a binary model”. Why is “the same degrees-of-freedom fitting” an important issue? Different IDPs have different patterns of missing data anyway.

We believe R4 is confusing the *tested effects* degrees-of-freedom (this being 1 here, i.e., a single model regressor) with the degrees-of-freedom of the model *residuals* (dominated by the number of non-missing samples for any given IDP). Changes in the latter have a very small effect on the modelling statistics (e.g., final P values), while changes in the former (e.g., using a greater number of modelling regressors, and combining these with an F statistic) can have a large effect on power. Hence, by using a simple, single regressor for the main effect of interest, we optimise power for finding effects that follow this form, at the risk of sub-optimal power (sensitivity to finding true effects) if the effect does not follow this form. We are of course happy to clarify this distinction further if considered necessary.

c) If the expressions g_m and g_0 are supposed to fully capture the age effect, why do the authors need other age terms such as $a_2 - a_1$, $a_2^2 - a_1^2$ in the model (2)? Also, the parameter g in the formula (1) is “the demeaned binary two-group variable” - what does this mean? What are the values of g ?

The difference here is simply between the regressor of interest and confounds of no interest. As explicitly described in the text, g_m is “The main case-vs-control group difference regressor of interest”. All other terms are present to account for confounds, including age-related confounds that are equally present *in cases and controls* (e.g., accounting for variations in difference in age between scans, for all subjects, to reduce uninteresting unmodelled noise, and noting that the subject groups are well-matched on this). We felt that our text on this was clear enough, but we are happy of course to include more explicit details.

Parameter g is described as “the demeaned binary two-group variable”; the values are either 0 (controls) or 1 (cases), and the full vector (across all subjects) of 0s and 1s was then demeaned so as to have zero mean, while still having peak-to-peak height 1. We feel that our text on this was clear enough, but again we are happy of course to specify this in more explicit detail.

d) Other confounds discussed on lines 22-24 page 34 should be added in model (2) instead of adopting a separate step during data preprocessing.

This is part of the data preprocessing, and is at the level of *individual scan sessions* (for example, accounting for position of subject’s head in the scanner), and hence this needs to be carried out before combining all scans and subjects together in the main modelling.

e) They offered the following argument as their defense for the adoption of model (2): “as a general rule, if a model was suboptimal, the main result would be a loss of sensitivity, not an increase in false positives.” This is not a legitimate argument. With their large data sample size, many candidates would likely be able to fit the category of such suboptimal models and show strong statistical evidence. Why adopt a particular exponent pattern with fixed parameters that cannot be justified? Furthermore, the authors did not really provide any model validation for the quality of their model. In addition, the model fits are not even plotted in any of the relevant figures (Figures 1, 3, S6) to show their fit quality. It might be true that with large sample size, many models (including the authors’ model (2)) could provide small p-values for those IDPs shown in the current figures, but this does not mean that one could pick up a convenient model without substantiation. More importantly, many IDPs might have been missed simply because of the unsubstantiated model. The information provided in the supplemental material (pages 12-13) does not address this problem.

We believe these points have been addressed in our previous Response, and in our comments above. In short: yes, there are many models one might adopt; some might fit the data (for a given IDP) better, and some might fit less well. Importantly, we have avoided the pitfall of overfitting, by not searching over different models, but rather simply proposing and testing one plausible, evidence-based model.

We do not intend to be claiming that this specific age-modulated model is *the* optimal model; the core point of the paper is not to claim that *this* model best describes the exact form of case-control differences associated

with SARS-CoV-2. If necessary we would be happy to explicitly include some relevant text in the Discussion, for example, something like:

“Many forms for the case-control group-difference model might be used. Possible models include: a binary regressor; single-regressors with age-modulated differences (such as the one primarily used here); more flexible models with multiple-regressors. Without testing a huge number of possible different models, one cannot make claims of absolute optimality. Nevertheless, our primary aim is not to prove model optimality, but to identify the effects of disease. To that aim, we have found statistically significant results with the simple model used here.”

f) The expression in (2) is not a formal presentation for a statistical model. Consider a more rigorous formulation.

We already replaced the core equation using symbols instead of words, in response to R4’s original review, despite our view that this actually reduced clarity for readers (given the overall simplicity of the model, and the very small number of equations in the paper). We believe that the current formulation is quite clear (though we would still prefer to revert to the original presentation). If necessary, we can of course replace “~” with “=” (and add the error/residual term e , and regression betas) in Eq2, giving $I_2 = g_m \beta_g + I_1 \beta_{I1} + c\beta_c + e$, but we feel this loses, rather than gains, clarity.

2. Baseline differences are still a sticky point. The authors have so far compared the average IDPs and some principal components between the two groups. However, their major research interest is not about average IDP values or principal components. If the main focus in the current paper regards the age pattern differences between the two groups, it would be more appropriate to address the issue of potential differences in age pattern at the baseline for all relevant IDPs (not just those “top 10”). For example, properly model the age patterns at the baseline and assess the cross-group differences.

R4 seems unfortunately to have missed that we had indeed included all of this in our Revision. This was all described in our Response document and mentioned in the revised main article:

1. Supplementary Analysis 2 (“Are nIDPs group-different at baseline, when clustering nIDPs using PCA?”) described in detail a range of baseline evaluations with non-imaging-derived phenotypes, with **both binary and age-modulated** baseline group regressors.
2. The 19 Supplementary Figures (“Baseline hypothesis-driven results”) show visually the baseline IDP values for the two subject groups as a (model-free) function of age, visualising the baseline age patterns.
3. Of most direct relevance to this point from R4, Supplementary Table S2 gives full results for all IDPs, **with separate full results tables for binary, and for age-modulated, baseline modelling**.
4. We provide a Supplementary Figure showing the voxelwise age-modulated differences at baseline between the cases and controls.

Response p25: *Thanks - we agree with these concerns, and have now added several deeper and “more aggressive” evaluations of potential baseline effects, both in imaging and non-imaging variables. Please see our detailed response above to the similar questions raised by Reviewer 1 (point 4.), and our new Supplementary Analyses. In short, we have now also evaluated both univariate and dimensionally-reduced versions of baseline effects, to avoid sensitivity-loss associated with multiple comparisons, and have more aggressively and directly tested for whether any baseline effects could “explain away” the primary longitudinal effects. These results strengthen our assessment of baseline differences, and increase confidence that the observed effects in patients with COVID-19 are not a result of such differences.*

Response p25: *We have now added as Supplementary Material the boxplots and age-bin curves for the top 10+10 significant results. In addition, we provide the full list of baseline imaging results (using both binary and, separately, age-modulated regressors), referred to in the Results section.*

Revision p18:

Additional cross-sectional imaging results

When looking at binary baseline differences between controls and future cases, none of the IDPs with significant longitudinal effects for either hypothesis-driven or exploratory approaches demonstrated significant differences at baseline between the two groups (lowest $P_{fwe}=0.59$, nothing surviving FDR correction; Table S2). When applying age-modulation in the two-group modelling of IDPs at baseline, a few of the IDPs demonstrated significant differences between control and future SARS-CoV-2 groups, mainly for diffusion indices in the olfactory functional networks, as well as in the subcortical grey matter. As some IDPs cover spatially extended regions of the brain, we visually explored whether these baseline differences had any spatial overlap with our longitudinal results, but found none (Supplementary Analyses and Figure S7). The full list of (binary and age-modulated) results

from group comparisons between the two groups at baseline, and separately, at the second timepoint are available in Tables S2 and S3.

In addition, none of the 10 cognitive variables showed significant difference at baseline between SARS-CoV-2 and control groups (min $P_{\text{uncorr}}=0.08$). With age-modulation, only one cognitive score, time to complete Pairs Matching round, showed a trend difference at baseline ($P_{\text{uncorr}}<0.05$, $P_{\text{fwe}}=0.29$, not passing FDR), while the longitudinal cognitive results were observed in the UK Biobank Trail Making Test.

Additional tests of potential influence of baseline non-imaging factors on significant results

We repeated the main analysis modelling for those top 10 IDPs found to show longitudinal differences between the SARS-CoV-2 and control groups, across both hypothesis-driven and exploratory approaches. For each of 6,301 non-imaging variables available (see Methods), we included that variable as an additional confounder in the longitudinal analyses. On the basis of the regression Z-statistic values, the strength of the original associations was not reduced by more than 25% for any of the non-imaging variables.

We further carried out the same analyses, but using dimension reduction (principal component analysis) applied to these 6,301 non-imaging phenotypes ($d=1$ to $d=700$), and also just focusing on cognition, with 540 cognitive variables ($d=10$). We found no substantial reduction in our longitudinal results with any of these principal components. In particular, for cognition where two components were significantly different at baseline (PC1 and PC4, Supplementary Analyses), the strongest reduction in Z was found for crus II of the cerebellum when adding PC1 to the model, with a decrease in Z of only 5.7 % (from $Z=4$ to $Z=3.77$), while the Z values associated with all the other IDPs were reduced by less than 5%. Adding PC4 to our main model reduced Z by 0.4% at most.

3. Conceptual issue. The word “significant” (and significantly) has been used many times throughout the paper. The reader could interpret it to refer to the amplitude of the effect under discussion or the statistical strength. It is unclear what the authors meant in all these cases considering the amplitude “ranging from 0.2 to 2% for the vast majority of the significant longitudinal results” given that their model is reasonable. Without clarification, the usage can lead to misunderstanding and confusion.

Nature has strict rules on use of the word "significant", and in every instance we have used this word with the correct intended meaning: relating to statistical (P) significance.

4. As for the choice of correction method for multiple comparisons, previously I raised the issue of the arbitrariness between FWE and FDR. They responded with the following: “Primarily we rely on FDR correction, which provides good power while controlling for multiple testing in a principled manner, but when a result additionally attains FWE significance, we wish to indicate this.” I could understand if one decided on one before the project and stuck with it, but is it problematic to present the results based on which of the two rendered more preferable results? If FDR is the primary choice, were all the permutations performed with FDR as a metric at the end?

We refer to the answer given in our Response p28: “... for any given model, we first computed the uncorrected p-values; we then computed both FDR- and FWE-corrected inferences as two distinct measures of strength of evidence for an effect while accounting for multiple testing. Primarily we rely on FDR correction, which provides good power while controlling for multiple testing in a principled manner, but when a result additionally attains FWE significance, we wish to indicate this. Thus, in the Results section and the main Tables, we always specify the findings obtained using both correction methods.”

FDR and FWE do not represent “independent” statistical tests, but provide different stringency of multiple comparison control, and with both being widely used and accepted: both FDR and FWE provide control of multiple comparisons, with FDR usually providing more power by admitting more false positives (while still controlled as a proportion of all detections), and FWE having lower power while providing more stringent control of false positives (controlling chance of any false positives at all). Hence, reporting all tests that pass FDR, and also noting which of those further pass FWE, is simply analogous to (for example), listing all tests passing $P<0.01$, while also further noting which of these even pass $P<0.001$.

Minor issues

** Outliers were misclassified as part of missing data. First, outliers are not missing data. The IDPs that were deemed as not reproducible were also treated as outliers (lines 9-10, page 35). What is the portion of the outlier IDPs? The arbitrariness of thresholding for outliers remains. If “a threshold of 8 was too aggressive”, why not use the same threshold of 15 for all scenarios? In addition, state the sources for those outliers.

This is a misunderstanding. The causes of a data point being missing vs being an outlier are in general different. We have not “classified” outliers as missing data, but by necessity we do treat both missing data points and outlier points in the same way: they are both ignored, as per simple conservative modelling practice.

R1 had quite rightly asked us to include much greater detail on amounts of missing data and outliers (e.g., Response p12-13, and Revision p34-35); the criteria for determining outliers was described and explained in Response p31 (in short, $MAD > 15$ would be a weak (high) threshold for variables with close-to-normal distribution, while of course the choice of any MAD threshold is ultimately “arbitrary”).

Response p12:

We removed a priori individual-connection functional connectivity edges and task-fMRI activation IDPs, as these have very low reproducibility and have for example previously been found to have very low heritability, due primarily to being intrinsically noisy (e.g., Smith et al., Nat Neurosci 2021). However, a reduction of the hundreds of functional connectivity edges to a greatly-reduced set of 6 latent factors achieves a large noise reduction and signal boost (e.g., for use in heritability estimates, as discussed and presented in Elliott et al., Nature 2018), and so we retained those 6.

However, we suspect that the reviewer is more likely asking for more details about which IDPs were removed during the preprocessing of IDPs prior to the longitudinal modelling. IDPs (both existing and new) were removed automatically as follows:

(a) For all cases and controls, IDPs were removed if their scan-rescan reproducibility, averaged across cases and controls, was less than $r=0.5$. This reduced the number of IDPs from 2,630 to 2,048.

(b) For all cases and controls, IDPs were removed if less than 50 participants (in total) had valid data. This removed just one IDP from the full (“Exploratory”) list, leaving 2,047 IDPs.

(c) For each remaining IDP, for each subject, an IDP was set to “missing” if it was more than 8 times the median absolute deviation around the median (across all subjects). This outlier removal is applied at both the individual-timepoint stage in IDP preprocessing, and also after computing the IDP2-IDP1 longitudinal change.

Missing data for individual subjects and specific IDPs can therefore occur because of step (c) above, or because the IDP was missing in the original data (e.g., because a given modality was not usable from a given participant). The fraction of total non-missing data, averaged across IDPs, is 0.93; all full xls results tables include the number of usable measurements for each IDP and for each statistical test. Importantly, there was no imbalance in amount of missing/outlier data between cases and controls: the number of cases with usable data, normalised by the total number of subjects with usable data, has the following percentiles across IDPs: percentiles [0, 1, 50, 99, 100] = 0.50, 0.50, 0.52, 0.52, 0.60, i.e., the median percentile is 0.52. From this analysis, the only 3 IDPs having this fraction greater than 0.53 were thalamic nuclei diffusion IDPs, which do not appear in any of our main results. These are also the only 3 IDPs with more than 24% missing/outlier data. We would be happy to provide an even finer-grained breakdown of the patterns of missing/outlier data across IDPs and/or subjects, if the reviewers/editor would like additional Supplementary Material, but we believe that the above summaries are sufficient.

Line 14, page 33: “using the olfactory bulb and hypothalamus maps (unthresholded and thresholded at 0.3)” - Why a threshold of 0.3 is adopted? And why two separate volume extraction methods?

Interpolation is unavoidable when transforming masks from one space to another (e.g., standard population average template space into subject-specific native space). This means that even a binary mask needs re-thresholding (noting that even nearest-neighbour interpolation in effect implies a choice on thresholding). It is therefore necessary to choose a threshold (a step often carried out “behind the scenes”, without being explicitly described). To reduce concerns about arbitrariness of threshold selection when re-binarising these ROI masks, we generated these IDPs in two ways (two choices of threshold), at the risk of reducing overall testing sensitivity by increasing the number of tests. This increase in number of tests is of course correctly accounted for by FDR/FWE. The olfactory bulb is very thin, compared with the data resolution, hence the need for the partial volume modelling for this structure. Having said all this, these IDPs did not generate significant results in any case.

Lines 22-24, page 34: “cross-sectional deconfounding... was carried out for head size, scanner table position, and image motion in the diffusion MRI data.” Instead of a separate modeling step that may cause biases, would it be better to add these confounds to your main model (2)?

R4 already asked this question (Point 1d), which we have answered above.

Editorial Note: These responses were sent back to Reviewer 2, who felt that they addressed the remaining concerns.

To settle the scholarly debate between the authors and Reviewer 4, the editors sought advice from an independent expert.

This individual read through Reviewer 4’s concerns and the authors’ responses and ultimately agreed with the authors. Based on this feedback, we considered these issues resolved